# On Theoretical Interpretations of Concept-Based In-Context Learning

## Abstract

In-Context Learning (ICL) has emerged as an important new paradigm in natural language processing and large language model (LLM) applications. However, the theoretical understanding of the ICL mechanism remains limited. This paper aims to investigate this issue by studying a particular ICL approach, called concept-based ICL (CB-ICL). In particular, we propose theoretical analyses on applying CB-ICL to ICL tasks, which explains why and when the CB-ICL performs well for predicting query labels in prompts with only a few demonstrations. In addition, the proposed theory quantifies the knowledge that can be leveraged by the LLMs to the prompt tasks, and leads to a similarity measure between the prompt demonstrations and the query input, which provides important insights and guidance for model pre-training and prompt engineering in ICL. Moreover, the impact of the prompt demonstration size and the dimension of the LLM embeddings in ICL are also explored based on the proposed theory. Finally, several real-data experiments are conducted to validate the practical usefulness of CB-ICL and the corresponding theory.

## 1 Introduction

With the great successes of large language models (LLMs), in-context learning (ICL) has emerged as a new paradigm for natural language processing (NLP) (Brown et al., 2020; Chowdhery et al., 2023; Achiam et al., 2023), where LLMs addresses the requested queries in context prompts with a few demonstrations. In contrast to conventional supervised learning, the ICL can perform well in prediction and inference tasks with very few samples by leveraging the semantic knowledge learned from the LLMs without training or fine-tuning the model parameters (Liu et al., 2022; Lu et al., 2022; Wei et al., 2022; Wu et al., 2023). This enables rapid task on-boarding (Sun et al., 2022), lowers computation and data costs compared with fine-tuning, and underpins current practice in instruction following (Lin et al., 2024), tool use (Schick et al., 2023), and agent memory (Chhikara et al., 2025). Therefore, a systematical understanding of ICL mechanism has appeared to be important in engineering designs in many areas of LLMs and NLP.

Recent researches on understanding the ICL mechanism mainly focused on functional modules (Olsson et al., 2022; Bietti et al., 2023; Wang et al., 2023a; Li et al., 2024a), theoretical interpretation based on Bayesian and gradient descent Views (Xie et al., 2022; Zhou et al., 2023; Dai et al., 2023; Mahankali et al., 2023), and learning and information theoretic perspectives (Garg et al., 2022; Akyürek et al., 2022; Pan et al., 2023; Yang et al., 2024). In particular, most of such researches concentrated on analyzing specific mathematical models such as linear regression for given functional classes, investigating the asymptotic learning behaviors of ICL, or characterizing different kinds of convergent properties of transformers in gradient descent. However, there still lacks theoretical justification of why ICL can performs well with only very few demonstrations, and some important questions for deeply understanding ICL mechanism remained open, such as theoretically characterizing the knowledge leveraged by LLMs, and quantifying the impact of the prompt engineering (Brown et al., 2020) in ICL.

In this paper, we develop a theoretical framework to analyze the performance of the Concept-Based In-Context Learning (CB-ICL) approach to address the aforementioned issues. As illustrated in Figure 1, in CB-ICL, a pre-trained LLM is employed to represent the semantic embeddings of the prompt contexts, where the parameters of the LLM is fixed throughout the learning task without

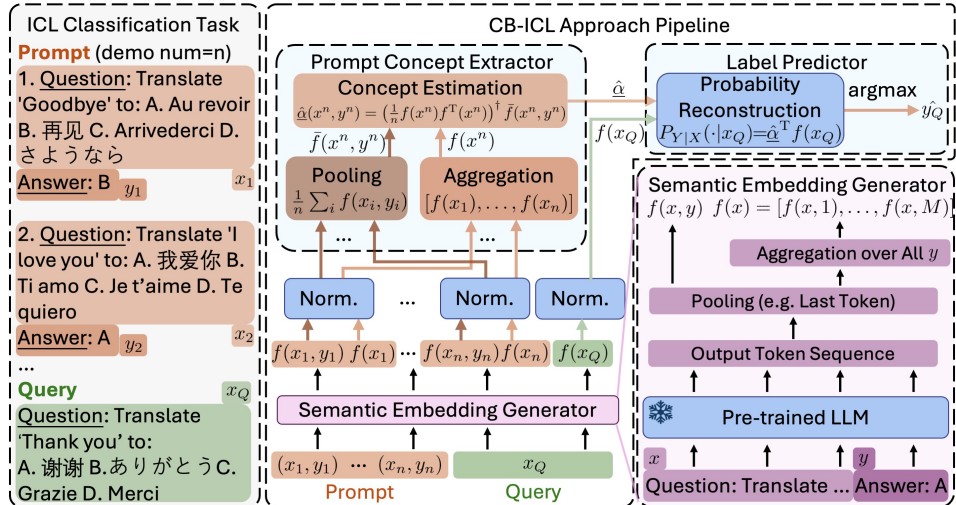

Figure 1: The targeting task and working pipeline of the CB-ICL approach.

fine-tuning. Then, the concept of the prompt contexts, represented by the vector $\underline{\alpha}$, is learned by the a prompt concept extractor. Finally, the concept vector is applied to estimate the posterior distribution of the label given the query context for solving the prediction task. To characterize the performance of CB-ICL, we establish upper bounds for the mean-squared excessive risk between the estimated posterior distribution and the ground truth distribution. The proposed upper bounds and the corresponding analyses lead to the theoretical insights and contributions summarized as follows:

- We model the semantic knowledge leveraged from the pre-trained LLM to the CB-ICL as a projection of the prompt distribution onto the semantic knowledge subspace spanned by the LLM embedding, and show that the CB-ICL achieves theoretically provably good learning performance, if both the semantic concept of the prompt is captured by the pre-trained LLM, and the correlation between the prompt query and the corresponding label is strong (which is true in many practical scenarios). This explains why well-designed LLMs can lead to good performances in many ICL applications.

- In addition, a similarity measure between the prompt demonstrations and queries is defined from the derived upper bound, which characterizes the impact of selecting related prompt demonstrations in CB-ICL. Such a similarity measure suggests how to design theoretically good demonstrations in prompt engineering.

- Moreover, we demonstrate the impacts of the dimension of LLM embedding, number of prompt demonstrations, and the cardinality of labels in ICL. In particular, it is shown that the higher LLM embedding dimension, the more difficult to learn the prompt concept from the semantic knowledge subspace of the LLM embedding, which suggests the importance of constructing parsimony and informative LLM embeddings in ICL.

- Furthermore, we quantify the learning performance degradation when the prompt demonstrations are not sufficient to illustrate the prompt concept, or the LLM embedding cannot fully capture the semantic knowledge of the prompt, which provides theoretical insights of prompt engineering and pre-training.

- Finally, real-data experiments are conducted on several LLMs and datasets to validate the performance of CB-ICL. The results show that the performance of CB-ICL is comparable to the existing ICL methods. Moreover, the aforementioned theoretical insights are also verified, which leads to useful guidance for designing effective ICL approaches.

## 2 RELATED WORKS

### 2.1 IN-CONTEXT LEARNING MECHANISM

Recent studies have proposed multiple explanations for ICL. One view interprets it as implicit meta-learning or Bayesian inference, where the prompt defines a latent task and transformers adapt by approximating Bayesian model averaging over tasks (Xie et al., 2022; Dai et al., 2023; Li et al., 2023; Zhou et al., 2023). Another line emphasizes algorithmic simulation, showing that transformers can reproduce standard estimators and approximate gradient descent updates, with deeper layers corresponding to iterative optimization (Garg et al., 2022; Akyürek et al., 2022; Von Oswald et al., 2023; Pan et al., 2023). A third perspective highlights the role of pre-training diversity: sufficiently broad task coverage enables robust in-context generalization, while narrow pre-training yields biased predictors (Raventós et al., 2023; Yang et al., 2024). Although insightful, these works mostly rely on simplified setups such as linear regression or asymptotic analysis. Extending theory to real-world LLMs in high-dimensional semantic spaces remains a key open challenge.

### 2.2 PROMPT ENGINEERING AND DEMONSTRATION SELECTION

Another research direction studies how prompt design affects ICL. Performance is highly sensitive to format, order, and content (Wang et al., 2023b; Liu et al., 2024). Retrieval-based approaches select demonstrations similar to the query (Su et al., 2022; Qin et al., 2023; Li & Qiu, 2023), but risk redundancy, motivating strategies that balance relevance and diversity (Wang et al., 2023b). More advanced methods fine-tune retrievers (Rubin et al., 2021; Mavromatis et al., 2023), optimize policies with reinforcement learning (Zhang et al., 2022), or leverage chain-of-thought prompting (Wei et al., 2022). Despite practical progress, most strategies are heuristic, lacking principled criteria to explain why certain demonstrations are more effective. Developing such theory remains an active research frontier.

## 3 CB-ICL MODEL FORMULATION

**In Context Prompt Assumptions** The prompt contexts are consisted of a collection of $n$ demonstrations $\{(x_i, y_i)\}_{i=1}^n$, where $x_i$'s are the input texts, and $y_i$'s are the corresponding answers such that $y_i \in \{1, \ldots, M\}, \forall i$, and a query input text $x_Q$. Denote $x^n = (x_1, \ldots, x_n)$ and $y^n = (y_1, \ldots, y_n)$, the goal of ICL is to predict the label $y_Q$ of $x_Q$, given the prompt contexts $x^n, y^n$, and $x_Q$ (Yang et al., 2024). In addition, we assume that $\{(x_i, y_i)\}_{i=1}^n$ and $(x_Q, y_Q)$ are generated and follow the probabilistic relationship:

$$P(x_Q, y_Q, y^n | x^n) = P_{X_Q}(x_Q) P_{Y|X}(y_Q | x_Q) \prod_{i=1}^n P_{Y|X}(y_i | x_i),$$

for some ground truth distributions $P_{Y|X}$, and $P_{X_Q}$, i.e., the labels $y^n$ and $y_Q$ are conditionally independently generated from $x^n$ and $x_Q$ by the same conditional distribution $P_{Y|X}(y|x)$, respectively. Note that we do not make assumptions on the joint distribution of the input texts $x_1, \ldots, x_n$, such as independent and identically distributed (i.i.d.), since such assumptions are often unrealistic in practical applications.

**Semantic Embeddings** Given a pair of input text and label $(x, y)$, we denote the semantic embedding generated by the pre-trained LLM as $f(x, y; \underline{\theta})$, where $\underline{\theta}$ represents the parameters of the LLM. Since there is no parameter fine-tuning in ICL, the parameters of the LLM will be fixed throughout the whole learning tasks, and the semantic embedding will simply be denoted as $f(x, y)$. In addition, before feeding into the prompt concept extractor, the LLM embedding is normalized and a bias term is padded at the end of the embedding vector, i.e., if we denote $f(x, y) = [f_1(x, y), \ldots, f_K(x, y)]$, then $\sum_y f_k(x, y) = 0$, $\sum_y f_k^2(x, y) = 1$, $\forall k < K - 1$, and $f_K(x, y) = \frac{1}{\sqrt{M}}, \forall x, y$.

Moreover, we express the ground truth distributions $P_{Y|X}$ in terms of a projection onto the space spanned by the LLM embedding functions $f_k(x, y)$, for $k = 1, \ldots, K$, as

$$P_{Y|X}(y|x) = \sum_{k=1}^K \alpha_k f_k(x, y) + R(x, y) = \underline{\alpha}^{\mathrm{T}} f(x, y) + R(x, y), \quad \forall x, y, \tag{1}$$

where $\underline{\alpha}$ can be viewed as ground truth concept, and $R(x, y)$ is a residual term that can be interpreted as the knowledge not captured by the LLM, and is orthogonal to the LLM embeddings, i.e.,

$$\sum_{x,y} P_{X_Q}(x) f_k(x, y) R(x, y) = 0, \quad \forall k.$$

In the remaining parts, we call the pre-trained LLM model *complete* if $R(x, y) = 0, \forall x, y$, and *incomplete* otherwise. Note that the complete pre-trained LLM model fully captures the semantic knowledge of the prompt contexts.

**Prompt Concept Extractor and Label Predictor**   Given the prompt contexts and the LLM embedding, the prompt concept extractor is defined as

$$\hat{\underline{\alpha}}(x^n, y^n) \triangleq \mathbf{F}_n^{\dagger}(x^n) \bar{f}_n(x^n, y^n) \tag{2}$$

where

$$\mathbf{F}_n(x^n) = \frac{1}{n} \sum_{i=1}^n \sum_y f(x_i, y) f^{\mathrm{T}}(x_i, y), \quad \bar{f}_n(x^n, y^n) = \frac{1}{n} \sum_{i=1}^n f(x_i, y_i),$$

and where "$\dagger$" denotes the psuedo-inverse. Note that $\hat{\underline{\alpha}}(x^n, y^n)$ can be interpreted as an estimation of the ground truth concept $\underline{\alpha}$ from the prompt contexts. In the following, we call the prompt demonstrations *sufficient* if $\mathbf{F}_n(x^n)$ is invertible, and *insufficient* otherwise.

Moreover, we denote $\hat{P}_{Y|X}(\cdot|x_Q)$ as an estimation of the ground truth distributions $P_{Y|X}(\cdot|x_Q)$ given the query input text $x_Q$ defined as

$$\hat{P}_{Y|X}(y|x_Q) = \sum_{k=1}^K \hat{\alpha}_k(x^n, y^n) f_k(x_Q, y) = \hat{\underline{\alpha}}^{\mathrm{T}}(x^n, y^n) f(x_Q, y).$$

Then, the CB-ICL label predictor is given by $\arg\max_y \hat{P}_{Y|X}(y|x_Q)$.

**Mean-Squared Excessive Risk**   In particular, we apply the mean-squared risk to measure the difference between $\hat{P}_{Y|X}(\cdot|x_Q)$ and $P_{Y|X}(\cdot|x_Q)$, defined as

$$\ell(x^n, y^n; x_Q) \triangleq \sum_y \left( \hat{P}_{Y|X}(y|x_Q) - P_{Y|X}(y|x_Q) \right)^2.$$

The in-context learning capability of CB-ICL is measured by the excessive risk (conditioned on the prompt input texts $x^n$), defined as

$$\mathbb{E}_{P_{Y^n|X^n}} \left[ \ell(x^n, Y^n; x_Q) | X^n = x^n \right]. \tag{3}$$

In the next section, we will draw the connection between the excessive risk and the error probability of label prediction, which shows the usefulness of analyzing the excessive risk in ICL scenarios.

## 4   THEORETICAL ANALYSES OF CB-ICL

### 4.1   COMPLETE AND SUFFICIENT MODELS

In this subsection, we analyze the excessive risk of CB-ICL in the case $R(x, y) = 0$, and $\mathbf{F}_n(x^n)$ is invertible. To delineate the theoretical results, we define the matrices of the LLM embeddings with respect to the prompt input texts as:

$$f(x_Q) = [f(x_Q, 1), \dots, f(x_Q, M)] \in \mathbb{R}^{K \times M}, \quad \mathbf{F}(x_Q) = f(x_Q) f^{\mathrm{T}}(x_Q) \in \mathbb{R}^{K \times K},$$

$$f(x^n) = [f(x_1), \dots, f(x_n)] \in \mathbb{R}^{K \times nM}, \quad \mathbf{F}_n(x^n) = \frac{1}{n} f(x^n) f^{\mathrm{T}}(x^n) \in \mathbb{R}^{K \times K},$$

and the matrices

$$\mathbf{Q}(x_i) = \mathsf{diag}\left\{ P_{Y|X}(1|x_i), \dots, P_{Y|X}(M|x_i) \right\} - \phi_i \phi_i^{\mathrm{T}},$$

$$\mathbf{Q}(x^n) = \mathsf{diag}\left\{ \mathbf{Q}(x_1), \dots, \mathbf{Q}(x_n) \right\} \tag{4}$$

where $\phi_i^{\mathrm{T}} = \left[ P_{Y|X}(1|x_i), \dots, P_{Y|X}(M|x_i), \right]$.

**Lemma 4.1.** *The matrix $\boldsymbol{Q}(x_i)$ is positive semi-definite, for all $i$, and the largest eigenvalue of $\boldsymbol{Q}(x_i)$, denoted as $\lambda_1(\boldsymbol{Q}(x_i))$, satisfies*

$$\lambda_1(\boldsymbol{Q}(x_i)) \leq 2P_Y(y_{\max}|x_i)\left(1 - P_Y(y_{\max}|x_i)\right),$$

*where $P_Y(y_{\max}|x_i) \triangleq \max_y P_Y(y|x_i))$. Moreover, the largest eigenvalue of $\boldsymbol{Q}(x^n)$ satisfies*

$$\lambda_1(\boldsymbol{Q}(x^n)) = \max_i \lambda_1(\boldsymbol{Q}(x_i)).$$

*Proof.* See Appendix F.1. $\qquad\square$

Then, the following Theorem characterizes the excessive risk in the complete and sufficient case.

**Theorem 4.2.** *When $R(x,y) = 0$, and $\mathbf{F}_n(x^n)$ is invertible, the excessive risk defined in (3) can be bounded as*

$$\mathbb{E}_{P_{Y^n|X^n}}\left[\ell\left(x^n, Y^n; x_Q\right)|X^n = x^n\right] \leq \frac{K}{n}\lambda_1\left(\mathbf{F}(x_Q)\mathbf{F}_n^{-1}(x^n)\right)\lambda_1(\boldsymbol{Q}(x^n)), \qquad (5)$$

*where $\lambda_1$ denotes the largest eigenvalue, and $\boldsymbol{Q}(x^n)$ is as defined in (4).*

*Proof.* See Appendix F.2. $\qquad\square$

In particular, the following theoretical insights that can be obtained from Theorem 4.2:

- If the input text $x$ and the label $y$ of the prompt are strongly correlated, i.e., the ground truth distribution satisfies $P_{Y|X}(y_{\max}|x) \simeq 1, \forall x$ (e.g. mathematical reasoning tasks), where $P_{Y|X}(y_{\max}|x)$ is as defined in Lemma 4.1, then from Lemma 4.1, it holds for $\lambda_1(\mathbf{Q}(x_i)) \simeq 0$, and the excessive risk of CB-ICL for learning the ground truth distribution is vanishing. Therefore, the CB-ICL can achieve striking performance in predicting the label of the query input text with only a few prompt demonstrations, if the LLM embedding is complete, and the input text and label are strongly correlated.

- In addition, it is shown in Appendix G.1 that $\lambda_1\left(\mathbf{F}(x_Q)\mathbf{F}_n^{-1}(x^n)\right) \geq 1$ with equality holds when $\mathbf{F}(x_Q) = \mathbf{F}_n(x^n)$. This provides a theoretical explanation that designing semantically correlated demonstrations in the prompt can enhance the ICL performance. Moreover, the quantity $\lambda_1^{-1}\left(\mathbf{F}(x_Q)\mathbf{F}_n^{-1}(x^n)\right)$ can be employed as a measure for selecting semantically correlated demonstrations to improve the ICL performance in prompt engineering. The real-data experiments in the Section 5 shows the applicability of this measure in real problem.

- Finally, it can be observed from (4) that excessive risk is inversely proportional to the number $n$ of demonstrations in the prompt, and proportional to the dimension $K$ of the LLM embedding. Therefore, although designing high dimensional LLM embeddings can capture more semantic knowledge of the prompt, i.e., making $R(x,y)$ smaller (c.f. (Kaplan et al., 2020)), in the meanwhile, it is also more difficulty to learn the prompt concept from the high-dimensional embedding space reflected by the growing excessive risk with respect to the dimension $K$. This tradeoff suggest the importance of learning parsimony and informative semantic embeddings in ICL.

## 4.2 COMPLETE AND INSUFFICIENT MODELS

In this subsection, we investigate the situation when the LLM embedding is complete, but the prompt demonstrations are not sufficient to illustrate the prompt concept, i.e., $\mathbf{F}_n(x^n)$ is not a full-rank matrix. We aim to characterize the impact of insufficient prompt demonstrations in CB-ICL, which can be formalized in the following Theorem.

**Theorem 4.3.** *When $R(x,y) = 0$, and $\mathbf{F}_n(x^n)$ is not invertible, the excessive risk (2) can be bounded as*

$$\mathbb{E}_{P_{Y^n|X^n}}\left[\ell\left(x^n, Y^n; x_Q\right)|X^n = x^n\right] \leq \frac{K}{n}\lambda_1\left(\mathbf{F}(x_Q)\mathbf{F}_n^{\dagger}(x^n)\right)\lambda_1(\boldsymbol{Q}(x^n)) + \underbrace{\left\|f^{\mathrm{T}}(x_Q)\mathbf{F}_n^{\perp}(x^n)\underline{\alpha}\right\|^2}_{(*)},$$

*where $\mathbf{F}_n^{\perp}(x^n) = \boldsymbol{I}_K - \mathbf{F}_n^{\dagger}(x^n)\mathbf{F}_n(x^n)$.*

*Proof.* See Appendix F.3. □

Comparing to (5) in Theorem 4.2, there is a penalty term $(*)$ caused by the insufficiency of the prompt demonstrations to the upper bound of the excessive risk. To interpret this term, note that from the definition of (2), $\underline{\hat{\alpha}}(x^n, y^n)$ is orthogonal to the null space of $\mathbf{F}_n(x^n)$, and hence the CB-ICL cannot learn the prompt concept aligned with the null space of $\mathbf{F}_n(x^n)$. Since $\mathbf{F}_n^\perp(x^n)$ is a projection operation, which projects the vector onto the null space of $\mathbf{F}_n(x^n)$, the penalty term $(*)$ can be interpreted as the information of the query embedding $f(x_Q)$ aligned with the null space of $\mathbf{F}_n(x^n)$, which cannot be learned by CB-ICL due to the insufficiency of the prompt demonstrations.

Moreover, it is readily to show that if $\mathbf{F}(x_Q) = \mathbf{F}_n(x^n)$, then the penalty term $(*)$ is 0. This tells that if the semantic information of the query text is well illustrated by the prompt demonstrations, the columns of the query embedding matrix $f(x_Q)$ is orthogonal to the null space of $\mathbf{F}_n(x^n)$, and there is no information and performance loss caused by the insufficiency of the prompt demonstrations. This again indicates the importance of designing illustrative prompt demonstrations.

### 4.3 INCOMPLETE AND INSUFFICIENT MODELS

In practice, the LLM embeddings often cannot capture the complete knowledge of the prompt contexts, i.e., $R(x, y) \neq 0$, which introduce the learning performance degradation caused by the incomplete knowledge LLM embeddings. To quantify such performance degradation, we analyze the expected excessive risk with respect to $P_{X_Q}$, and define the vectors

$$\mathsf{R}(x_Q) = [R(x_Q, 1), \dots, R(x_Q, M)]^\mathrm{T} \in \mathbb{R}^M, \mathsf{R}(x^n) = [\mathsf{R}(x_1), \dots, \mathsf{R}(x_n)]^\mathrm{T} \in \mathbb{R}^{nM},$$

Then, the excessive risk can be characterized as follows.

**Theorem 4.4.** *When $R(x, y) \neq 0$, and $\mathbf{F}_n(x^n)$ is not invertible, the excessive risk (2) averaged with respect to $P_{X_Q}$ can be bounded as*

$$\mathbb{E}_{P_{X_Q} P_{Y^n | X^n}} \left[ \ell\left(x^n, Y^n; x_Q\right) | X^n = x^n \right]$$

$$\leq \frac{K}{n} \lambda_1 \left( \mathbf{F}_Q \mathbf{F}_n^\dagger(x^n) \right) \lambda_1(\boldsymbol{Q}(x^n))$$

$$+ \frac{1}{n} \lambda_1 \left( \mathbf{F}_Q \mathbf{F}_n^\dagger(x^n) \right) \cdot \|\mathsf{R}(x^n)\|^2 + \sum_{x_Q, y} P_{X_Q}(x_Q) R^2(x_Q, y) \tag{6}$$

$$+ \underline{\alpha}^\mathrm{T} \mathbf{F}_n^\perp(x^n)^\mathrm{T} \mathbf{F}_Q \mathbf{F}_n^\perp(x^n) \underline{\alpha} - \frac{2}{n} \mathsf{R}^\mathrm{T}(x^n) f^\mathrm{T}(x^n) \mathbf{F}_n^\dagger(x^n) \mathbf{F}_Q \mathbf{F}_n^\perp(x^n) \underline{\alpha} \tag{7}$$

*where $\mathbf{F}_Q = \mathbb{E}_{P_{X_Q}}[\mathbf{F}(X_Q)]$.*

*Proof.* See Appendix F.4. □

Similar to the discussions in Section 4.1, the penalty term (7) quantifies the amount of semantic information aligned with the null space of $\mathbf{F}_n(x^n)$ that cannot be learned in the CB-ICL approach due to the insufficiency of the prompt demonstrations. Moreover, the penalty terms (6) quantify the performance degradation of the excessive risk caused by the incompleteness of the LLM embedding. In particular, the first term of (6) can be interpreted as the learning bias in the model (1), due to the incompleteness of the LLM embedding, and the second term of (6) quantifies the amount of semantic information of the query input text that is not captured by the LLM embeddings. Finally, we present some remarks of CB-ICL to draw the connections and comparisons to existing intuitions and techniques of ICL researches:

- It is widely believed in ICL researches that during pre-training, LLM models acquire a broad range of semantic prior knowledge from the training data, which later aids task-specific learning representations (Chan et al., 2022; Shin et al., 2022; Yadlowsky et al., 2023; Yang et al., 2024). In particular, this empirical observation can be theoretically justified by the CB-ICL, which allows the generalizability to a broad class of ICL problems, as long as the ground truth distribution of the prompt contexts is somehow aligned with the semantic knowledge subspace spanned by the LLM embeddings.

- The pre-training/warm-up techniques (Brunet et al., 2023; Shi et al., 2023; Li et al., 2024b) in traditional ICL can also be beneficial in CB-ICL, which reduces the modeling error $R(x, y)$, and improve the learning performance. Moreover, the pre-training loss does not need to be restricted to the MSE loss between $\hat{P}_{Y|X}$ and $P_{Y|X}$, as long as the global minimum of the pre-training loss is achieved at $\hat{P}_{Y|X} = P_{Y|X}$.

- Note that the prompt concept extractor can be interpreted as a wide-sense transformer with the softmax activation function replaced by a quadratic function (c.f. linear attention (Wang et al., 2020a; Shen et al., 2021; Han et al., 2024)). This essentially suggests the application of more general kinds of transformer architectures in theoretical analyses and algorithm designs in ICL and other machine learning fields. Usually, the linear transformer takes the form as (Wang et al., 2020a)

$$\text{Attention}(q, \{k_i, v_i\}) = \phi^{\text{T}}(q) \left( \sum_i \phi(k_i) v_i^{\text{T}} \right),$$

where $\phi$ is a feature map such that $\langle \phi(q), \phi(k) \rangle = \kappa(q, k)$, for some kernel function $\kappa$, allowing the attention operation to be rewritten in a linearized (kernelized) form. In our setting, we define

$$\phi(q) = \left( \mathbf{F}_N^{\dagger}(x^n) \right)^{1/2} f(x_Q), \ \phi(k) = \left( \mathbf{F}_N^{\dagger}(x^n) \right)^{1/2} f(x_i),$$

with $v_i = e_i$ being one-hot vector with the $i$-th element equals 1. Under this construction, the original estimated $\hat{P}_{Y|X}(y|x_Q)$ can be written as a linear attention.

### 4.4 THE LABEL PREDICTING ERROR PROBABILITY

To further justify the practically usefulness of the CB-ICL and the corresponding theoretical analyses, in this subsection we establish the connection between the mean-squared excessive risk and the label predicting error probability that is widely adopted in real applications. To this end, we define

$$\hat{y}_{\max} = \arg\max_y \hat{P}_{Y|X}(y|x_Q), \tag{8}$$

and denote $P_j$ as the $j$th largest probability among $\{P_{Y|X}(y|x_Q)\}_{y=1}^M$. In the following, we assume $P_1 > P_j$, for all $j \geq 2$.

**Theorem 4.5.** *Suppose that for some $j \geq 1$, the excessive risk*

$$\mathbb{E}_{P_{Y^n|X^n}} \left[ \ell\left(x^n, Y^n; x_Q\right) | X^n = x^n \right] = \frac{1}{2}(P_1 - P_j)^2 + \gamma,$$

*where $0 \leq \gamma < \frac{1}{2}(P_1 - P_{j+1})^2 - \frac{1}{2}(P_1 - P_j)^2$, then the label predicting error probability is lower bounded by*

$$\mathbb{E}_{P_{Y^n|X^n}} \left[ P_{Y|X}(\hat{y}_{\max}|x_Q) \mid X^n = x^n \right] \geq P_j - \frac{2\gamma}{2P_1 - P_j - P_{j+1}}. \tag{9}$$

*Proof.* See Appendix F.5. □

Note that the lower bound of error probability actually defines one upper bound of correct probability. Theorem 4.5 shows that designing $\hat{P}_{Y|X}(y|x_Q)$ with small excessive risk also leads to small label predicting error probability from the label predictor Eq. (8), which demonstrates the applicability of the theoretical analyses of the CB-ICL in real scenarios.

## 5 EXPERIMENT

We conduct experiments on four representative benchmarks that collectively measure both general knowledge and complex reasoning ability, including MMLU (Hendrycks et al., 2020), MMLU-Pro(Wang et al., 2024), GPQA and GPQA-Diamond (Rein et al., 2024). Together, these datasets provide a comprehensive evaluation suite, balancing breadth (MMLU, MMLU-Pro) with depth in

Table 1: Performance comparisons on classification task are conducted in both the vanilla ICL (Brown et al., 2020) and CB-ICL setting. We report the accuracy with 5 randomly selected demonstrations from the same task.

| Dataset | | 8B | | | 14B | | 32B | Average |
|---|---|---|---|---|---|---|---|---|
| | | LLaMA3 | Qwen3 | Deepseek-R1 | Qwen3 | Deepseek-R1 | Qwen3 | |
| MMLU | ICL (Brown et al., 2020) | 68.40% | 76.89% | 63.54% | 81.05% | 74.46% | 83.61% | 74.66% |
| | CB-ICL | **71.07%** | **77.77%** | **64.58%** | **81.38%** | **80.32%** | **83.62%** | **76.46%** |
| MMLU-Pro | ICL (Brown et al., 2020) | **35.36%** | **56.73%** | 41.10% | **61.03%** | 57.80% | 65.54% | **52.93%** |
| | CB-ICL | 33.26% | 53.62% | **42.30%** | 60.47% | **59.04%** | **65.89%** | 52.43% |
| GPQA | ICL (Brown et al., 2020) | 34.50% | 44.44% | 45.32% | 47.90% | 46.32% | 49.49% | 44.66% |
| | CB-ICL | **40.77%** | **61.60%** | **50.82%** | **50.65%** | **48.45%** | **54.82%** | **51.19%** |
| GPQA-diamond | ICL (Brown et al., 2020) | 28.29% | 62.00% | **49.10%** | 64.31% | 59.10% | **68.40%** | 55.20% |
| | CB-ICL | **33.56%** | **63.01%** | 48.82% | **64.88%** | **60.13%** | 66.34% | **56.13%** |

complex reasoning (GPQA, GPQA-Diamond). Furthermore, we benchmark three representative families of open-source LLMs at different parameter scales (8B, 14B, and 32B), including LLaMA3 family (Dubey et al., 2024), Qwen3 family (Yang et al., 2025) and Deepseek-R1 distilled (Guo et al., 2025) model family. This selection covers both general-purpose and reasoning-enhanced model families, enabling a systematic comparison across scaling and architectural choices.

## 5.1 PERFORMANCE VALIDATIONS OF CB-ICL

The result in Table 1 provide consistent evidence for the effectiveness of CB-ICL: To begin with, across all model families and datasets, CB-ICL either matches or surpasses vanilla ICL (Brown et al., 2020), denoting the effectiveness of proposed CB-ICL framework. Moreover, improvements are larger on harder datasets. On MMLU (general factual recall), gains are marginal (less than 2%), while on GPQA and GPQA-Diamond the improvements are substantial. Furthermore, scaling laws persist under CB-ICL. For example, within Qwen3 family, performance consistently increases from 8B to 32B, expect Qwen3-8B on GPQA dataset. This shows that CB-ICL is compatible with scaling, implying that this mechanism complements rather than replaces larger model capacity.

## 5.2 PROMPT DEMONSTRATION DESIGNS

As discussed in Section 4.1, the semantic similarity $\lambda_1^{-1}\left(\mathbf{F}(x_Q)\mathbf{F}^\dagger(x^n)\right)$ works as a score function of the quantity of prompt. To demonstrate this, we adopt a simple "translate to Chinese" dataset split into two parts that differ only in the cue language (English to Chinese and Italian to Chinese). We call a demonstration set similar to a query if they come from the same part, otherwise dissimilar. As is illustrated in Figure. 2, when the demonstrations are chosen from the similar prompt, the similarity score $\lambda_1^{-1}\left(\mathbf{F}(x_Q)\mathbf{F}^\dagger(x^n)\right)$ tends to be larger.

To further validate the effectiveness of our proposed similarity measure, we compare CB-ICL under two settings: (i) CB-ICL, where demonstrations are randomly selected from the same task, and (ii) CB-ICL (golden), where demonstrations are selected based on the similarity measure. Specifically, for each candidate demonstration $x_i$ in the demonstration pool, we compute its similarity score $\lambda_1^{-1}(\mathbf{F}(x_Q)\mathbf{F}^\dagger(x_i))$ with the target query $x_Q$, and then the top 5 demonstrations with the highest scores are selected as the golden demonstrations. The results are shown in Table 2. We observe two consistent trends: First, golden selection substantially improves accuracy across datasets. These large margins highlight that the similarity measure successfully identifies high-value demonstrations that guide the model more effectively. Second, the improvement generalizes across model families and scales. Both LLaMA3, Qwen3 and DeepSeek-R1 benefit from golden selection, regardless of parameter size, suggesting that the similarity measure provides complementary guidance beyond model architecture and scaling.

What's more, we compared the performance of CB-ICL (golden) with the cosine similarity baseline across models of different parameter scales and datasets, where the top-5 demonstrations are selected according to embedding cosine similarity. Results show that across most model scales and benchmarks, CB-ICL (golden) matches or outperforms cosine similarity selection by 0.4%–1.5% in

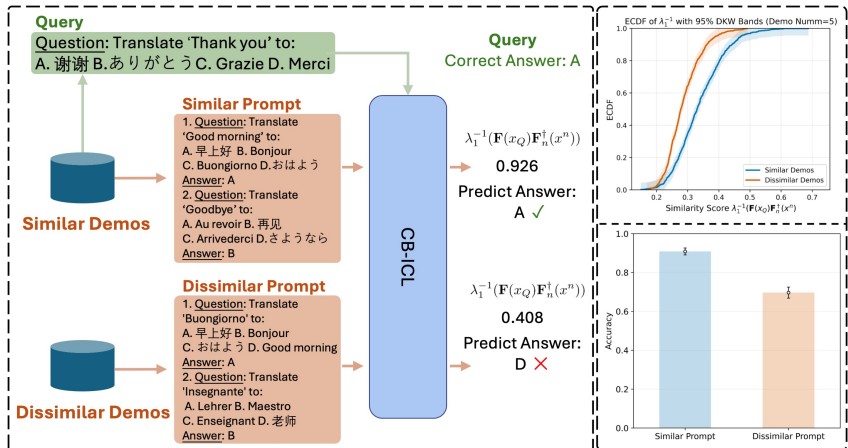

Figure 2: Comparison of similarity score $\lambda_1^{-1}\left(\mathbf{F}(x_Q)\mathbf{F}^\dagger(x^n)\right)$ between similar and dissimilar demonstration sets on the "translate to Chinese" task. We report the result with 5 demonstrations.

Table 2: Performance is compared between randomly selected and golden demonstrations. CB-ICL uses 5 random demonstrations, while ICL (Consine Similarity) and CB-ICL (golden) selects the top-5 demonstrations ranked by proposed similarity metric.

| | Dataset | 8B | | | 14B | | 32B | Average |
| | | LLaMA3 | Qwen3 | Deepseek-R1 | Qwen3 | Deepseek-R1 | Qwen3 | |
|---|---|---|---|---|---|---|---|---|
| MMLU | CB-ICL | 71.07% | 77.77% | 64.58% | 81.38% | 80.32% | 83.62% | 76.46% |
| | ICL (Consine Similarity) | 72.45% | 79.41% | 68.24% | 82.98% | 83.15% | 84.20% | 78.41% |
| | CB-ICL (golden) | **73.74%** | **80.86%** | **68.44%** | **82.64%** | **82.34%** | **84.42%** | **78.74%** |
| MMLU-Pro | CB-ICL | 33.26% | 53.62% | 42.30% | 60.47% | 59.04% | 65.89% | 52.43% |
| | ICL (Consine Similarity) | 36.54% | 59.74% | 42.84% | 60.34% | 61.52% | 66.24% | 54.54% |
| | CB-ICL (golden) | **36.58%** | **58.97%** | **44.24%** | **62.67%** | **61.34%** | **67.29%** | **55.18%** |
| GPQA | CB-ICL | 40.77% | 61.60% | 50.82% | 50.65% | 48.45% | 54.82% | 51.19% |
| | ICL (Consine Similarity) | **49.10%** | **66.67%** | 52.56% | 53.16% | 50.50% | 57.35% | **54.89%** |
| | CB-ICL (golden) | 48.51% | 65.77% | **51.67%** | 55.83% | 52.32% | 58.73% | 55.47% |
| GPQA-diamond | CB-ICL | 33.56% | 63.01% | 48.82% | 64.88% | 60.13% | 66.34% | 56.12% |
| | ICL (Consine Similarity) | 41.35% | 70.56% | **51.78%** | 66.39% | 61.32% | 66.67% | **61.92%** |
| | CB-ICL (golden) | **42.28%** | **73.83%** | 52.47% | **67.32%** | **62.91%** | **67.43%** | 59.68% |

average accuracy, with particularly consistent gains on MMLU, MMLU-Pro, and GPQA-Diamond. These improvements are most pronounced for all-scale models (8B, 14B and 32B). Notably, while cosine similarity sometimes achieves competitive or even slightly higher performance on specific tasks (e.g., GPQA with 8B models), CB-ICL (golden) exhibits more stable improvements across datasets and model families, leading to higher average performance in most settings. Overall, these results suggest that although cosine similarity is a strong and commonly used baseline, incorporating higher-order structure through the proposed similarity measure yields more robust and consistently effective demonstration selection.

## 5.3 IMPACT OF EMBEDDING DIMENSION $K$

In section 4.1, the trade-off of dimension $K$ is discussed. A larger dimension $K$ will improve the representation ability of semantic knowledge, but the associated excessive risk will increase since the larger dimension $K$ adds to the difficulty of estimation $\underline{\alpha}$. In this section, we conducted an experiment to validate this phenomenon. To manipulate the dimension $K$, we trained a lightweight auto-encoder (with a three-layer MLP as encoder) on the raw hidden states of the model. The encoder–decoder is trained with MSE reconstruction loss until the loss is less than $2.5e-6$. During evaluation, we replace the original hidden state with the encoded representation, ensuring that the rest of the model remains unchanged. We report the 5-shot MMLU results of Qwen3-8B and Qwen3-14B as in the Table 3. We find that performance improves as the dimension grows from 256 to

Table 3: Effect of embedding dimension $K$ on CB-ICL performance (accuracy, %). The results are reported on MMLU dataset with the same demonstrations.

| Dimension $K$ | 128 | 256 | 512 | 1024 | 2048 | 4096 | 5120 |
|---|---|---|---|---|---|---|---|
| Qwen3-8B | 79.52 | 80.14 | 80.96 | 81.16 | **81.92** | 80.86 | 79.68 |
| Qwen3-14B | 79.27 | 80.52 | 81.32 | 82.45 | 83.13 | **83.32** | 82.64 |

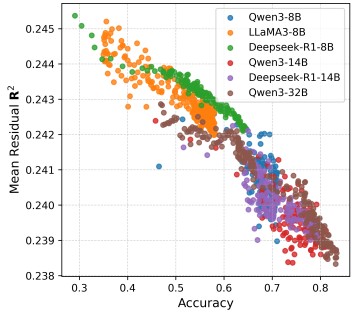

(a) **MMLU**: Accuracy (↑) vs residual risk $\mathbf{R}^2$ (↓).

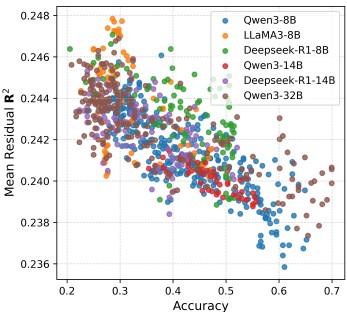

(b) **GPQA**: Accuracy (↑) vs residual risk $\mathbf{R}^2$ (↓).

Figure 3: Performance of incomplete models across datasets. We report the results with 5 golden demonstrations and disturbed last layer of model.

around 2048, but drops again when K becomes excessively large. As for the larger model, this trend still holds, with the dimension of best dimension increases from 2048 to 4096. This matches our theoretical prediction of a non-monotonic risk curve that too small a dimension underfits, whereas too large a dimension amplifies noise and increases excess risk.

### 5.4 IMPACT OF INCOMPLETENESS OF LLMS IN ICL

To further validate the theoretical insights of CB-ICL, we conduct experiments on incomplete models, i.e., $R(x,y) \neq 0$ for some $x$ and $y$. We disturb the last layer of LLMs. For each model and dataset, we measure two metrics: (i) accuracy: calculated across the benchmark with 5 golden demonstrations; (ii) the mean residual $\mathbf{R}^2 = \sum_{x,y} P_X(x) R^2(x,y)$: quantifying the amount of knowledge not captured by the LLM embeddings, with higher $\mathbf{R}^2$ indicating stronger incompleteness. Figure 3 reports the relationship between accuracy and $\mathbf{R}^2$ of different models. A clear negative correlation emerges: (i) $\mathbf{R}^2$ decreases as accuracy increases, implying that their representations better align with the task concepts. (ii) Incomplete models suffer from high $\mathbf{R}^2$ (e.g. LLaMA3-8B achieves the largest $\mathbf{R}^2$), reflecting the incompleteness of the model; (iii) The observed trend is consistent across different architectures, demonstrating that the proposed residual measure is model-agnostic and captures a general phenomenon of incompleteness.

## 6 CONCLUSION

In this paper, we present theoretical analyses of CB-ICL, which reveals the fundamental mechanism of why and how ICL can perform well in prompts with only a few demonstrations. Moreover, our theory quantifies the knowledge leveraged by the pre-trained LLM embeddings, the similarity of the prompt demonstrations and query input text, as well as the impact of the number of prompt demonstrations and the dimensions of LLM embeddings, which provides useful guidance for model pre-training and prompt engineering. Finally, the effectiveness of our theory is validated by several real-data experiments. While our theory offers valuable insights into the performance of CB-ICL, there are several limitations to consider. First, our experiments focus on classification tasks, while the applicability to more complex tasks such as sequence generation or reasoning remains to be fully explored. Second, while we demonstrate improvements on several benchmarks, we acknowledge that the performance gains may not generalize to all datasets, particularly those with substantial reasoning requirements or small sample sizes.

## 7 ETHICS STATEMENT

This work is a theoretical and empirical study of ICL in LLMs. Our analyses are purely mathematical, and our experiments are conducted on publicly available benchmark datasets (e.g., MMLU, GPQA). We do not involve human subjects, personal data, or sensitive attributes. The datasets used are standard in the community and have been widely adopted in prior work. Our contributions are methodological and theoretical in nature, and we do not foresee any direct risks of harm, privacy leakage, or fairness concerns arising from our results. We adhere to the ICLR Code of Ethics throughout the research and submission process.

## 8 REPRODUCIBILITY STATEMENT

We have taken multiple steps to ensure the reproducibility of our work. All theoretical results are stated with explicit assumptions and are accompanied by complete proofs, which are provided in the main paper and the appendix. The experimental evaluations are conducted on publicly available benchmark datasets (MMLU, MMLU-Pro, GPQA, GPQA-diamond). We also release the implementation of our proposed methods and all experiment scripts as anonymous supplementary material to facilitate independent verification. Together, these efforts ensure that both the theoretical analyses and the empirical results can be reproduced by the community.

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

## A  THE USE OF LLM

In this work, we used large language models (LLMs) solely as general-purpose assistive tools. Specifically, LLMs were employed in three ways:

- To assist with the polishing and clarification of mathematical proofs.
- To improve the readability and fluency of the manuscript writing.
- To serve as a coding assistant (e.g., code completion and debugging support).

LLMs were not involved in the research ideation, theoretical development, or experimental design. All conceptual contributions and scientific insights are the work of the authors.

## B  NOTATION TABLE

In this section, we summarize the notation appeared in this paper.

Table 4: Basic Variables

| Symbol | Description |
| --- | --- |
| $x, y$ | Input text and corresponding label |
| $x^n = (x_1, \ldots, x_n)$ | Collection of $n$ demonstration input texts |
| $y^n = (y_1, \ldots, y_n)$ | Collection of $n$ demonstration labels |
| $x_Q$ | Query input text |
| $y_Q$ | Query label to be predicted |
| $M$ | Number of possible classes/labels |
| $K$ | Dimension of LLM embedding space |
| $\theta$ | Parameters of the pre-trained LLM |
| $P_{X_Q}(x_Q)$ | Distribution of query inputs |
| $P_{Y\|X}(y\|x)$ | Ground truth conditional distribution |

Table 5: LLM Embeddings

| Symbol | Description |
| --- | --- |
| $f(x, y) = [f_1(x, y), \ldots, f_K(x, y)] \in \mathbb{R}^K$ | Semantic $K$-dimensional embedding |
| $f(x) = [f(x, 1), \ldots, f(x, M)] \in \mathbb{R}^{K \times M}$ | Embedding matrix for all labels of input $x$ |
| $F(x) = f(x)f^T(x) \in \mathbb{R}^{K \times K}$ | Outer product matrix for input $x$ |
| $F_n(x^n) = \frac{1}{n}\sum_{i=1}^n \sum_{y=1}^M f(x_i, y)f^T(x_i, y)$ | Average embedding matrix for demonstrations |
| $\bar{f}_n(x^n, y^n)$ | Average embedding of demonstrations with their labels |

Table 6: Concept and Prediction

| Symbol | Description |
| --- | --- |
| $\alpha$ | Ground truth concept vector |
| $R(x, y)$ | Residual term (knowledge not captured by LLM) |
| $\hat{\alpha}(x^n, y^n) = F_n^\dagger(x^n)\bar{f}_n(x^n, y^n)$ | Estimated prompt concept |
| $\hat{P}_{Y\|X}(y\|x_Q) = \hat{\alpha}^T(x^n, y^n)f(x_Q, y)$ | Estimated conditional distribution |
| $\hat{y}_{\max}^*$ | Predicted label |

## C  ALGORITHM

In this section, we describe the main pipeline of Algorithm as in Algorithm 1.

Table 7: Risk and Error Measures

| Symbol | Description |
|---|---|
| $\ell(x^n, y^n; x_Q)$ | Mean-squared risk |
| $E_{P_{Y^n\|X^n}}[\ell(x^n, Y^n; x_Q)\|X^n = x^n]$ | Excessive risk (conditioned on prompt inputs) |
| $R^2 = \sum_{x,y} P_X(x) R^2(x, y)$ | Measure of LLM incompleteness |

Table 8: Matrix Operations

| Symbol | Description |
|---|---|
| $A^\dagger$ | Pseudo-inverse of matrix $A$ |
| $F_n^\perp(x^n) = I_K - F_n^\dagger(x^n) F_n(x^n)$ | Projection onto null space of $F_n(x^n)$ |
| $\lambda_1(\cdot)$ | Largest eigenvalue of a matrix |
| $\phi_i = [P_{Y\|X}(1\|x_i), \ldots, P_{Y\|X}(M\|x_i)]^T$ | Probability vector for input $x_i$ |
| $Q(x_i) = \text{diag}\{P_{Y\|X}(1\|x_i), \ldots, P_{Y\|X}(M\|x_i)\} - \phi_i \phi_i^T$ | Covariance-like matrix |
| $Q(x^n) = \text{diag}\{Q(x_1), \ldots, Q(x_n)\}$ | Block diagonal matrix |

# D ABLATION AND EXTRA EXPERIMENTS

## D.1 ABLATION ON TRANSFORMER ARCHITECTURE

In this section, we report the results tested with encoder-decoder and decoder-only transformers. We conducted experiments on encoder-only model (including BERT, MPNet, MiniLM) and encoder-decoder models (including FLAN-T5, BART), covering all architectural types mentioned. The experiments followed the same CB-ICL workflow as the main paper. The model we used is listed as in Table 10.

We extract the output of the last layer of encoder and generate the embedding with mean pooling. We report the resuls on MMLU, MMLU-Pro, GPQA and GPQA-diamond as in Table 11.

For detail information, we repeat all experiment 10 times and report the average value. Results confirm that CB-ICL is effective across these paradigms: on most datasets (e.g., GPQA-Diamond, GPQA), CB-ICL consistently outperforms vanilla ICL. For instance, FLAN-T5-XXL achieves 30.10% in the setting of CB-ICL, compared with 22.46% under ICL on GPQA-Diamond, and BART reaches 33.24% (CB-ICL) compared with 21.50% (ICL) on the same dataset. Even for encoder-only models where vanilla ICL is not applicable, CB-ICL still yields meaningful performance (e.g., 29.18% on GPQA-Diamond for MPNET), demonstrating CB-ICL's broad applicability beyond decoder-only LLMs. What's more, by comparison, decoder-only models had higher ICL baselines (e.g., Qwen3-32B reached 76.46% on MMLU) and required no additional adjustments to work with CB-ICL.

## D.2 ABLATION ON ATTENTION

As is stated in Section 4.3, the proposed estimator can be viewed as a special case of linear attention. To compare the performance with standard attention (Vaswani et al., 2017), we set the standard attention as

$$\hat{P}_{Y|X}(y|x_Q) = \text{softmax}\left(\frac{f^{\mathrm{T}}(x_Q) f(x^n)}{\sqrt{K}}\right) V,$$

where $V = [e_{y_1}; e_{y_2}; \ldots, e_{y_n}] \in \mathbb{R}^{nM}$ and $e_{y_i} \in \mathbb{R}^m$ is one-hot vector with the $y_i$th element being 1. Based on this, we test the performance of LLaMA, Qwen and Deepseek families as in Table 12. The results demonstrate that the proposed approach consistently outperforms traditional attention-based models across different benchmarks, as highlighted in Table 12.

Table 9: Similarity Measure

| Symbol | Description |
|---|---|
| $\lambda_1^{-1}(F(x_Q)F_n^\dagger(x^n))$ | Similarity score between query and demonstrations |

---

**Algorithm 1** CB-ICL Detailed Procedure

---

**Require:** LLM parameters $\theta$, system prompt $s$, demonstrations $\{(x_i, y_i)\}_{i=1}^n$, query $x_Q$
**Ensure:** Predicted label $\hat{y}_Q$

1: **// Semantic embedding generation**
2: **for** $i = 1$ **to** $n$ **do**
3:      **for** $y = 1$ **to** $M$ **do**
4:          $\text{context}_{i,y} \leftarrow [s, x_i, y]$     // Construct full context with system prompt
5:          $h_{i,y} \leftarrow \text{LLM}_\theta(\text{context}_{i,y})$     // Forward pass through LLM
6:          $f(x_i, y) \leftarrow h_{i,y}^{\text{last layer}}[\text{last token}]$     // Extract embedding from last layer, last token
7:      **end for**
8: **end for**
9: **for** $y = 1$ **to** $M$ **do**
10:      $\text{query\_context}_y \leftarrow [s, x_Q, y]$
11:      $h_Q^y \leftarrow \text{LLM}_\theta(\text{query\_context}_y)$
12:      $f(x_Q, y) \leftarrow h_{Q,y}^{\text{last layer}}[\text{last token}]$
13: **end for**
14: **// Embedding normalization (as specified in Section 3)**
15: **for** $k = 1$ **to** $K$ **do**
16:      // Zero-mean and unit variance constraints
17:      $\mu_k \leftarrow \frac{1}{M} \sum_{y=1}^M f_Q^y[k]$
18:      $\sigma_k \leftarrow \sqrt{\frac{1}{M} \sum_{y=1}^M (f_Q^y[k] - \mu_k)^2}$
19:      $f_Q^y[k] \leftarrow \frac{f_Q^y[k] - \mu_k}{\sigma_k}$   for all $y$
20: **end for**
21: **// Add bias dimension**
22:      $f_Q^y[K+1] \leftarrow \frac{1}{\sqrt{M}}$   for all $y$
23: **// Concept estimation**
24: $F_n \leftarrow \frac{1}{n} \sum_{i=1}^n \sum_{y=1}^M f(x_i, y) f^T(x_i, y)$
25: $\bar{f}_n \leftarrow \frac{1}{n} \sum_{i=1}^n f(x_i, y_i)$
26: $\hat{\alpha} \leftarrow F_n^\dagger \bar{f}_n$
27: **Prediction**
28: **for** $y = 1$ **to** $M$ **do**
29:      $\hat{P}(y) \leftarrow \hat{\alpha}^T f_Q^y$
30: **end for**
31: $\hat{y}_Q \leftarrow \arg\max_y \hat{P}(y)$
32: **return** $\hat{y}_Q$

---

### D.3 WALL-CLOCK TIME REPORT

We report the runtime and memory overhead of CB-ICL in Table 13, decomposing the cost into golden demonstration selection and prediction probability extraction. All statistics are measured across the full evaluation set and reported using average, minimum, and maximum values.

The results show that prediction probability extraction incurs negligible overhead, with an average latency of only 3.32 ms and nearly constant memory usage around 73 MB. This stage corresponds to a single forward pass over candidate labels and does not involve matrix inversion or similarity computation, explaining both its low latency and stable memory footprint.

Table 10: The used Transformers architecture and selected models. The embeddings are generated with the of encoder layer with mean pooling.

| Model Structure | Selected Model |
|---|---|
| Encoder-decoder | Flan-T5 (Chung et al., 2024), BART(Lewis et al., 2020) |
| Encoder-only | BERT (Koroteev, 2021), MPNet(Song et al., 2020), MiniLM (Wang et al., 2020b) |

Table 11: Performance comparisons between vanilla ICL and CB-ICL across encoder–decoder and encoder-only models. We report 5-shot accuracy. For encoder-only models, vanilla ICL is not applicable.

| Dataset | | Encoder-Decoder | | | Encoder-Only | | |
|---|---|---|---|---|---|---|---|
| | | FLAN-T5-XXL (%) | FLAN-T5-XL (%) | BART (%) | MiniLM (%) | BERT (%) | MPNet (%) |
| MMLU | ICL | **55.10** | **52.40** | 22.90 | – | 23.02 | – |
| | CB-ICL | 54.67 | 50.53 | **26.12** | 25.75 | **25.65** | 25.85 |
| MMLU-Pro | ICL | **16.54** | 15.37 | **11.24** | – | **10.59** | – |
| | CB-ICL | 15.43 | **15.90** | 10.36 | 10.10 | 9.43 | 9.72 |
| GPQA | ICL | 26.85 | 23.52 | **27.27** | – | 25.44 | – |
| | CB-ICL | **28.00** | **27.80** | 26.93 | 29.08 | **27.31** | 28.63 |
| GPQA-Diamond | ICL | 22.46 | 24.08 | 21.50 | – | **29.19** | – |
| | CB-ICL | **30.10** | **29.03** | **33.24** | 25.98 | 29.03 | 29.18 |

In contrast, golden demonstration selection constitutes the primary computational cost of CB-ICL. The average runtime is 17.38 ms, with higher variance across samples. This variance is expected, as golden selection involves embedding construction and similarity evaluation over the demonstration pool, whose cost depends on both the embedding dimension and the numerical properties of the aggregated matrices. Despite this, the average overhead remains modest and is incurred only once per query, not per token generation step.

From a memory perspective, golden selection requires additional workspace to store embedding matrices and intermediate statistics, leading to higher peak memory usage (up to 507.84 MB in worst-case samples). However, the average memory usage remains well within practical GPU limits, and the prediction stage itself does not introduce additional memory pressure.

Overall, these results indicate that CB-ICL introduces a small, one-off preprocessing overhead for demonstration selection, while leaving inference-time cost almost unchanged. This makes the method practical for real-world deployment, especially in scenarios where multiple queries reuse the same or similar demonstration pools.

# E CHECK OF PROPOSED ESTIMATOR

In this section, we check the sum-to-one property of proposed estimator $\hat{P}_{Y|X}(y|x_Q)$ as

$$\sum_y \hat{P}_{Y|X}(y|x) = \sum_y \hat{\underline{\alpha}}^{\mathrm{T}}(x^n, y^n) f(x, y) = \hat{\underline{\alpha}}^{\mathrm{T}}(x^n, y^n) \left( \sum_y f(x, y) \right)$$

with

$$\sum_y f(x, y) = [0, \dots, 0, \sqrt{M}]^{\mathrm{T}}.$$

Furthermore, since $\mathbf{F}(x) = \sum_y f(x, y) f^{\mathrm{T}}(x, y)$ and $\mathbf{F}(x^n) = \frac{1}{n} \sum_{i=1}^n \mathbf{F}(x)$, Let $(\mathbf{F}(x))_{ij}$ denote the element at $i$-th row and $j$-th column, we have that

$$(\mathbf{F}(x))_{Kj} = \sum_y f_K(x, y) f_j(x, y) = \frac{1}{\sqrt{M}} \sum_y f_j(x, y) = 1(j = K),$$

with $1(j = K) = 1$ if $j = K$ and $1(j = K) = 0$ if $j \neq K$, and

$$(\mathbf{F}(x))_{jK} = \sum_y f_j(x, y) f_K(x, y) = \frac{1}{\sqrt{M}} \sum_y f_j(x, y) = 1(j = K).$$

Table 12: Comparison between CB-ICL and Attention-based example selection across different model scales. We report average 5-shot accuracy with 10 times repeated experiments.

| Dataset | | 8B | | | 14B | | 32B |
|---------|---|------|-------|-------------|-------|-------------|-------|
| | | LLaMA3 | Qwen3 | Deepseek-R1 | Qwen3 | Deepseek-R1 | Qwen3 |
| MMLU | CB-ICL | **71.07** | **77.77** | **64.58** | **81.38** | **80.32** | **83.62** |
| | Attention | 50.24 | 59.02 | 53.19 | 67.43 | 64.53 | 65.12 |
| MMLU-Pro | CB-ICL | **33.26** | **53.62** | **42.30** | **60.47** | **59.04** | **65.89** |
| | Attention | 12.60 | 43.78 | 31.40 | 52.08 | 53.08 | 51.41 |
| GPQA | CB-ICL | **40.77** | **61.60** | **50.82** | **50.65** | **48.45** | **54.82** |
| | Attention | 21.41 | 40.52 | 39.56 | 46.43 | 47.15 | 46.91 |
| GPQA-Diamond | CB-ICL | **33.56** | **63.01** | **48.82** | **64.88** | **60.13** | **66.34** |
| | Attention | 10.18 | 55.21 | 38.47 | 50.43 | 57.96 | 60.32 |

Table 13: Runtime and memory statistics for golden demonstration selection and prediction probability extraction.

| Metric | Statistic | Golden Demonstration Selection | Prediction Probability Extraction |
|--------|-----------|-------------------------------|-----------------------------------|
| Time | Average | 17.38 ms | 3.32 ms |
| | Min | 4.24 ms | 2.16 ms |
| | Max | 257.91 ms | 29.71 ms |
| Memory Usage | Average | 161.97 MB | 73.51 MB |
| | Min | 41.73 MB | 73.47 MB |
| | Max | 507.84 MB | 73.55 MB |

Therefore, we have that

$$\mathbf{F}(x) = \begin{bmatrix} (\mathbf{F}(x))_{1:K-1,1:K-1} & \mathbf{0} \\ \mathbf{0} & 1 \end{bmatrix}$$

and

$$\mathbf{F}_n^\dagger(x) = \begin{bmatrix} (\mathbf{F}_n^\dagger(x))_{1:K-1,1:K-1} & \mathbf{0} \\ \mathbf{0} & 1 \end{bmatrix}$$

Hence, the sum of $\hat{P}_{Y|X}(y|x_Q)$ is

$$\sum_y \hat{P}_{Y|X}(y|x) = \left( \frac{1}{n} \sum_{i=1}^n f(x_i, y_i) \right)^\mathrm{T} \mathbf{F}_n^\dagger(x) \left( \sum_y f(x_Q, y) \right)$$

$$= \left( \frac{1}{n} \sum_{i=1}^n f(x_i, y_i) \right)^\mathrm{T} \begin{bmatrix} \mathbf{0} & \mathbf{0} \\ \mathbf{0} & \sqrt{M} \end{bmatrix}$$

$$= \frac{1}{n} \sqrt{M} \sum_{i=1}^n f_K(x_i, y_i)$$

$$= \frac{\sqrt{M}}{n} \frac{n}{\sqrt{M}}$$

$$= 1.$$

In our formulation, since only the sum-to-one constraint holds, the coefficients should be viewed as signed weights (possibly extrapolative) rather than strictly "share of attention" in the conventional sense. We recognize that this is a limitation of our current linear framework, as we do not yet guarantee that the weights reside in the [0, 1] interval, which would make the analogy to soft-attention tighter.

# F  Proofs of Lemmas and Theorems

## F.1  Proof of Lemma 4.1

In this section, we aim to prove Lemma 4.1. We separate the proof into three steps. First, we analyze the spectrum of $\mathbf{Q}(x_i)$ for a fixed $i$. After that, we derive the upper bound of the largest eigenvalue of $\mathbf{Q}(x_i)$. Finally, we extend the result to $\mathbf{Q}(x^n)$.

**Lemma F.1.** *Let*

$$\boldsymbol{Q}(x_i) = \mathsf{diag}\{P_{Y|X}(1|x_i), \ldots, P_{Y|X}(M|x_i)\} - \phi_i \phi_i^{\mathrm{T}},$$

*where* $\phi_i = [P_{Y|X}(1|x_i), \ldots, P_{Y|X}(M|x_i)]^{\mathrm{T}}$. *Then the eigenvalues of* $\boldsymbol{Q}(x_i)$ *are either some* $P_{Y|X}(j|x_i)$ *or solutions to*

$$1 + \sum_{j=1}^{M} \frac{P_{Y|X}^2(j|x_i)}{\lambda - P_{Y|X}(j|x_i)} = 0, \tag{10}$$

*with all such solutions satisfying* $\lambda \leq \max_j P_{Y|X}(j|x_i)$.

*Proof.* The eigenvalues of $\mathbf{Q}(x_i)$ are the roots of the equation

$$\det(\lambda I - \mathsf{diag}\left\{P_{Y|X=x_i}\right\} + \phi_i \phi_i^{\mathrm{T}}) = 0, \tag{11}$$

where $\mathsf{diag}\left\{P_{Y|X=x_i}\right\} = \mathsf{diag}\{P_{Y|X}(1|x_i), \ldots, P_{Y|X}(M|x_i)\}$. For $\lambda \neq P_{Y|X}(j|x_i), \forall j$, the determinant can be factorized as

$$\begin{aligned}
&\det(\lambda I - \mathsf{diag}\left\{P_{Y|X=x_i}\right\} + \phi_i \phi_i^{\mathrm{T}}) \\
&= \det(\lambda I - \mathsf{diag}\left\{P_{Y|X=x_i}\right\})(1 + \phi_i^{\mathrm{T}}(\lambda I - \mathsf{diag}\left\{P_{Y|X=x_i}\right\})^{-1}\phi_i) \\
&= \prod_{j=1}^{M}(\lambda - P_{Y|X}(j|x_i))\left(1 + \sum_{j=1}^{M} \frac{P_{Y|X}^2(j|x_i)}{\lambda - P_{Y|X}(j|x_i)}\right)
\end{aligned}$$

Thus, eigenvalues are either some $P_{Y|X}(j|x_i)$ or solutions to Eq. (10). Furthermore, we have that all the solutions $\lambda$ to the Eq. (10) must satisfy that $\lambda < \max_j P_{Y|X}(j|x_i)$, which makes all roots of Eq. (11) no larger than $\max_j P_{Y|X}(j|x_i)$. □

**Lemma F.2.** *For each* $i$, *the largest eigenvalue of* $\boldsymbol{Q}(x_i)$ *satisfies*

$$\lambda_1(\boldsymbol{Q}(x_i)) \leq 2 \max_j P_{Y|X}(j|x_i)\left(1 - \max_j P_{Y|X}(j|x_i)\right).$$

*Proof.* To prove this result, we distinguish between two cases: whether the maximum probability is attained by more than one label, or by a unique label.

**Case 1: The size of set** $\arg\max_j P_{Y|X}(j|x_i)$ **is larger than** 1    By continuity, we have

$$\det(P_{Y|X}(j|x_i) \cdot I - \mathsf{diag}\left\{P_{Y|X=x_i}\right\} + \phi_i \phi_i^{\mathrm{T}}) = P_{Y|X}^2(j|x_i) \prod_{k \neq j}(P_{Y|X}(j|x_i) - P_{Y|X}(k|x_i)).$$

Since $\arg\max_y P_{Y|X}(y|x_i)$ is not a single point set, $\max_y P_{Y|X}(y|x_i)$ is one solution to Eq. (11). Then, the largest eigenvalue of $\mathbf{Q}(x_i)$ is $\max_y P_{Y|X}(y|x_i)$. In this case, since the size of set $\arg\max_j P_{Y|X}(j|x_i)$ is larger than 1, we have that $\max_j P_{Y|X}(j|x_i) \leq 1/2$, which gives that $\max_j P_{Y|X}(j|x_i) \leq 2 \max_j P_{Y|X}(j|x_i)(1 - \max_j P_{Y|X}(j|x_i))$.

**Case 2: The size of set** $\arg\max_j P_{Y|X}(j|x_i)$ **equals to** 1  Let $j_0 = \arg\max_j P_{Y|X}(j|x_i)$ and denote the largest root of the Eq. (10) by $\lambda_0$. Further, let arbitrary $j_1 \in \arg\max_{j,j\neq j_0} P_{Y|X}(j|x_i)$. Note that

$$1 + \sum_{j=1}^{M} \frac{P_{Y|X}^2(j|x_i)}{\lambda - P_{Y|X}(j|x_i)} > 1, \forall \lambda > P_{Y|X}(j_0|x_i),$$

$$\lim_{\lambda \to P_{Y|X}(j_0|x_i)^-} 1 + \sum_{j=1}^{M} \frac{P_{Y|X}^2(j|x_i)}{\lambda - P_{Y|X}(j|x_i)} = -\infty,$$

and

$$\lim_{\lambda \to P_{Y|X}(j_1|x_i)^+} 1 + \sum_{j=1}^{M} \frac{P_{Y|X}^2(j|x_i)}{\lambda - P_{Y|X}(j|x_i)} = \infty.$$

We have $P_{Y|X}(j_1|x_i) < \lambda_0 < P_{Y|X}(j_0|x_i)$. Following the analysis in Case 1, the value $P_{Y|X}(j_0|x_i)$ will not be the root of Eq. (11). Hence, we have the largest eigenvalue of $\mathbf{Q}(x_i)$ must be $\lambda_0$.

In this case, we have the

$$\frac{P_{Y|X}^2(j_0|x_i)}{\lambda_0 - P_{Y|X}(j_0|x_i)} + \frac{P_{Y|X}^2(j_1|x_i)}{\lambda_0 - P_{Y|X}(j_1|x_i)} \leq \overbrace{\sum_{j=1}^{M} \frac{P_{Y|X}^2(j|x_i)}{\lambda_0 - P_{Y|X}(j|x_i)}}^{=-1}$$

and

$$\overbrace{\sum_{j=1}^{M} \frac{P_{Y|X}^2(j|x_i)}{\lambda_0 - P_{Y|X}(j|x_i)}}^{=-1} \leq \frac{P_{Y|X}^2(j_0|x_i)}{\lambda_0 - P_{Y|X}(j_0|x_i)} + \frac{\sum_{k\neq j} P_{Y|X}^2(k|x_i)}{\lambda_0 - P_{Y|X}(j_1|x_i)}$$

$$\leq \frac{P_{Y|X}^2(j_0|x_i)}{\lambda_0 - P_{Y|X}(j_0|x_i)} + \frac{(1 - P_{Y|X}(j_0|x_i))^2}{\lambda_0 - (1 - P_{Y|X}(j_0|x_i))}$$

Hence, solving

$$\frac{P_{Y|X}^2(j_0|x_i)}{\lambda_0 - P_{Y|X}(j_0|x_i)} + \frac{(1 - P_{Y|X}(j_0|x_i))^2}{\lambda_0 - (1 - P_{Y|X}(j_0|x_i))} \geq -1$$

gives $\lambda_0 \leq 2P_{Y|X}(j_0|x_i)(1 - P_{Y|X}(j_0|x_i))$, where the equality is achieved by Bernoulli distribution.

In conclusion of the two cases, both cases give that

$$\lambda_1(\mathbf{Q}(x_i)) \leq 2\max_y P_{Y|X}(y|x_i)\left(1 - \max_y P_{Y|X}(y|x_i)\right).$$

This completes the proof. $\qquad\square$

**Lemma F.3.** *The largest eigenvalue of the block matrix $\boldsymbol{Q}(x^n)$ satisfies*

$$\lambda_1(\boldsymbol{Q}(x^n)) = \max_i \lambda_1(\boldsymbol{Q}(x_i)).$$

*Proof.* By the Rayleigh–Ritz characterization,

$$\lambda_1(\mathbf{Q}(x^n)) = \max_{\|\boldsymbol{v}\|=1} \boldsymbol{v}^{\mathrm{T}}\mathbf{Q}(x^n)\boldsymbol{v}.$$

Write $\boldsymbol{v}^{\mathrm{T}} = [\boldsymbol{v}_1^{\mathrm{T}}, \dots, \boldsymbol{v}_n^{\mathrm{T}}]$, with $\boldsymbol{v}_i$ that matches the size of $\mathbf{Q}(x_i)$. Then

$$\boldsymbol{v}^{\mathrm{T}}\mathbf{Q}(x^n)\boldsymbol{v} = \sum_{i=1}^{n} \boldsymbol{v}_i^{\mathrm{T}}\mathbf{Q}(x_i)\boldsymbol{v}_i \leq \sum_{i=1}^{n} \lambda_1(\mathbf{Q}(x_i))\|\boldsymbol{v}_i\|^2 \leq \max_i \lambda_1(\mathbf{Q}(x_i)).$$

Let $j \in \arg\max_i \lambda_1(\mathbf{Q}(x_i))$ and let $\boldsymbol{u}_j$ be a unit eigenvector of $\mathbf{Q}(x_j)$ for $\lambda_1(\mathbf{Q}(x_j))$. Define $\boldsymbol{v}$ by $\boldsymbol{v}_j = \boldsymbol{u}_j$ and $\boldsymbol{v}_i = \mathbf{0}$ for $i \neq j$. Then $\|\boldsymbol{v}\| = 1$ and

$$\boldsymbol{v}^{\mathrm{T}}\mathbf{Q}(x^n)\boldsymbol{v} = \boldsymbol{u}_j^{\mathrm{T}}\mathbf{Q}(x_j)\boldsymbol{u}_j = \lambda_1(\mathbf{Q}(x_j)) = \max_i \lambda_1(\mathbf{Q}(x_i)).$$

Therefore, $\lambda_1(\mathbf{Q}(x^n)) \geq \max_i \lambda_1(\mathbf{Q}(x_i))$. Combining both inequalities yields

$$\lambda_1(\mathbf{Q}(x^n)) = \max_i \lambda_1(\mathbf{Q}(x_i)).$$

This completes the proof. $\square$

By Lemmas F.1, F.2, and F.3, we conclude the proof of Lemma 4.1.

## F.2 PROOF OF THEOREM 4.2

To prove the Theorem 4.2, we first develop the lemma for expectation value of estimated concept $\hat{\underline{\alpha}}$.

**Lemma F.4.** *For sufficient prompt demonstrations, i.e., $\mathbf{F}_n(x^n)$ is invertible, it holds that for all $x^n$,*

$$\mathbb{E}_{P_{Y^n|X^n}}[\hat{\underline{\alpha}}(x^n, Y^n)|X^n = x^n] = \underline{\alpha}.$$

*Proof.* From definition of $\hat{\underline{\alpha}}$ in Eq. (2), we have that

$$\mathbb{E}_{P_{Y^n|X^n}}\left[\hat{\underline{\alpha}}(x^n, Y^n)|X^n = x^n\right]$$
$$= \mathbb{E}_{P_{Y^n|X^n}}\left[\mathbf{F}_n^{-1}(x^n)\bar{f}_n(x^n, Y^n)|X^n = x^n\right]$$
$$= \sum_{y^n}\prod_{i=1}^{n}P_{Y|X}(y_i|x_i)\left(\frac{1}{n}\sum_{i=1}^{n}\sum_{y}f(x_i, y)f^{\mathrm{T}}(x_i, y)\right)^{-1}\left(\frac{1}{n}\sum_{i=1}^{n}f(x_i, y_i)\right)$$
$$= \left(\frac{1}{n}\sum_{i=1}^{n}\sum_{y}f(x_i, y)f^{\mathrm{T}}(x_i, y)\right)^{-1}\frac{1}{n}\sum_{i=1}^{n}\sum_{y}P_{Y|X}(y|x_i)f(x_i, y)$$
$$= \left(\frac{1}{n}\sum_{i=1}^{n}\sum_{y}f(x_i, y)f^{\mathrm{T}}(x_i, y)\right)^{-1}\frac{1}{n}\sum_{i=1}^{n}\sum_{y}f(x_i, y)f^{\mathrm{T}}(x_i, y)\underline{\alpha}$$
$$= \underline{\alpha}.$$

$\square$

With the Lemma F.4, we directly have the written form of excessive risk for a complete model (i.e. $R(x, y) = 0$ for all $x, y$).

**Corollary F.5.** *For sufficient prompt demonstrations and complete model, the excessive risk can be written as*

$$\mathbb{E}_{P_{Y^n|X^n}}\left[\ell(x^n, Y^n; x_Q)|X^n = x^n\right] = \mathrm{tr}\left\{\mathbf{F}_n^{-1}(x^n)\mathbf{F}(x_Q)\mathbf{F}_n^{-1}(x^n)\mathsf{F}(x^n)\right\} - \underline{\alpha}^{\mathrm{T}}\mathbf{F}(x_Q)\underline{\alpha},$$

*where $\mathsf{F}(x^n) = \mathbb{E}_{P_{Y^n|X^n}}\left[\bar{f}_n(x^n, Y^n)\bar{f}_n^{\mathrm{T}}(x^n, Y^n)|X^n = x^n\right]$.*

*Proof.* From the definition of $\ell(x^n, y^n; x_Q)$, we have that (note that $\mathbf{F}_n^{-1}(x^n)$ is symmetric)

$$\mathbb{E}_{P_{Y^n|X^n}}\left[\ell(x^n, Y^n; x_Q) | X^n = x^n\right]$$

$$= \mathbb{E}_{P_{Y^n|X^n}}\left[\sum_y \left(\hat{P}_{Y|X}(y|x_Q) - P_{Y|X}(y|x_Q)\right)^2 | X^n = x^n\right]$$

$$= \mathbb{E}_{P_{Y^n|X^n}}\left[\sum_y \left(\underline{\hat{\alpha}}^{\mathrm{T}} f(x_Q, y) - \underline{\alpha}^{\mathrm{T}} f(x_Q, y)\right)^2 | X^n = x^n\right]$$

$$= \mathbb{E}_{P_{Y^n|X^n}}\left[(\underline{\hat{\alpha}} - \underline{\alpha})^{\mathrm{T}} \mathbf{F}(x_Q)(\underline{\hat{\alpha}} - \underline{\alpha}) | X^n = x^n\right]$$

$$= \mathbb{E}_{P_{Y^n|X^n}}\left[\underline{\hat{\alpha}}^{\mathrm{T}} \mathbf{F}(x_Q)\underline{\hat{\alpha}} | X^n = x^n\right] - 2\mathbb{E}_{P_{Y^n|X^n}}\left[\underline{\hat{\alpha}}^{\mathrm{T}} \mathbf{F}(x_Q)\underline{\alpha} | X^n = x^n\right] + \underline{\alpha}^{\mathrm{T}} \mathbf{F}(x_Q)\underline{\alpha}$$

$$= \mathbb{E}_{P_{Y^n|X^n}}\left[\mathrm{tr}\left\{\mathbf{F}_n^{-1}(x^n)^{\mathrm{T}} \mathbf{F}(x_Q)\mathbf{F}_n^{-1}(x^n)\bar{f}_n(x^n, Y^n)\bar{f}_n^{\mathrm{T}}(x^n, Y^n)\right\} | X^n = x^n\right] - \underline{\alpha}^{\mathrm{T}} \mathbf{F}(x_Q)\underline{\alpha}$$

$$= \mathrm{tr}\left\{\mathbf{F}_n^{-1}(x^n)\mathbf{F}(x_Q)\mathbf{F}_n^{-1}(x^n)\mathsf{F}(x^n)\right\} - \underline{\alpha}^{\mathrm{T}} \mathbf{F}(x_Q)\underline{\alpha}.$$

$\square$

Further, $\mathsf{F}(x^n)$ can be written as

$$\mathsf{F}(x^n) = \frac{1}{n^2} f(x^n)\tilde{\mathbf{Q}}(x^n)f^{\mathrm{T}}(x^n),$$

where

$$\tilde{\mathbf{Q}}(x^n) = \mathsf{diag}\{\mathsf{diag}\{P_{Y|X=x_1}\}, \ldots, \mathsf{diag}\{P_{Y|X=x_n}\}\} + \sum_{i=1}^n \sum_{j=1}^n v_i v_j^{\mathrm{T}} - \sum_i v_i v_i^{\mathrm{T}},$$

with $\mathsf{diag}\{P_{Y|X=x_i}\} = \mathsf{diag}\{P_{Y|X}(1|x_i), \ldots, P_{Y|X}(M|x_i)\}$, and

$$v_i = \left[0, \ldots, 0, P_{Y|X}(1|x_i), \ldots, P_{Y|X}(M|x_i), 0 \ldots, 0\right]^{\mathrm{T}}.$$

In the next step, we show the connection between $v_i$, $f(x^n)$ and $\underline{\alpha}$.

**Lemma F.6.** *It holds that*

$$\frac{1}{n^2}\mathrm{tr}\left\{\mathbf{F}_n^{-1}(x^n)\mathbf{F}(x_Q)\mathbf{F}_n^{-1}(x^n)f(x^n)\left(\sum_{i,j} v_i v_j^{\mathrm{T}}\right)f^{\mathrm{T}}(x^n)\right\} = \underline{\alpha}^{\mathrm{T}} \mathbf{F}(x_Q)\underline{\alpha}.$$

*Proof.* We have that

$$\frac{1}{n^2}\mathrm{tr}\left\{\mathbf{F}_n^{-1}(x^n)\mathbf{F}(x_Q)\mathbf{F}_n^{-1}(x^n)f(x^n)\left(\sum_{i,j} v_i v_j^{\mathrm{T}}\right)f^{\mathrm{T}}(x^n)\right\}$$

$$= \mathrm{tr}\left\{\mathbf{F}_n^{-1}(x^n)\mathbf{F}(x_Q)\mathbf{F}_n^{-1}(x^n)\left(\sum_{i,j}\left(\frac{1}{n}f(x^n)v_i\right)\left(\frac{1}{n}f(x^n)v_j\right)^{\mathrm{T}}\right)\right\}$$

$$= \mathrm{tr}\left\{\mathbf{F}_n^{-1}(x^n)\mathbf{F}(x_Q)\mathbf{F}_n^{-1}(x^n)\left(\frac{1}{n}\sum_{i,y} P_{Y|X}(y|x_i)f(x_i, y)\right)\left(\frac{1}{n}\sum_{j,y} P_{Y|X}(y|x_j)f(x_j, y)\right)^{\mathrm{T}}\right\}$$

From model 1, we have that for a complete model, $P_{Y|X}(y|x) = \underline{\alpha}^{\mathrm{T}} f(x, y)$. Hence, we have that

$$
\mathrm{tr}\left\{\mathbf{F}_n^{-1}(x^n)\mathbf{F}(x_Q)\mathbf{F}_n^{-1}(x^n)\left(\frac{1}{n}\sum_{i,y} P_{Y|X}(y|x_i)f(x_i,y)\right)\left(\frac{1}{n}\sum_{j,y} P_{Y|X}(y|x_j)f(x_j,y)\right)^{\mathrm{T}}\right\}
$$

$$
= \mathrm{tr}\left\{\mathbf{F}_n^{-1}(x^n)\mathbf{F}(x_Q)\mathbf{F}_n^{-1}(x^n)\left(\frac{1}{n}\sum_{i,y} f(x_i,y)f^{\mathrm{T}}(x_i,y)\underline{\alpha}\right)\left(\frac{1}{n}\sum_{j,y} f(x_j,y)f^{\mathrm{T}}(x_j,y)\underline{\alpha}\right)^{\mathrm{T}}\right\}
$$

$$
= \mathrm{tr}\left\{\mathbf{F}_n^{-1}(x^n)\mathbf{F}(x_Q)\mathbf{F}_n^{-1}(x^n)\mathbf{F}_n(x^n)\underline{\alpha}\left(\mathbf{F}_n(x^n)\underline{\alpha}\right)^{\mathrm{T}}\right\}
$$

$$
= \underline{\alpha}^{\mathrm{T}}\mathbf{F}(x_Q)\underline{\alpha}.
$$

This completes the proof. $\qquad\square$

Hence, we have the excessive risk without $\underline{\alpha}$.

**Corollary F.7.** *Let*

$$
\boldsymbol{Q}(x_i) = \mathrm{diag}\{P_{Y|X}(1|x_i), \ldots, P_{Y|X}(M|x_i)\} - \phi_i\phi_i^{\mathrm{T}},
$$

*where $\phi_i = [P_{Y|X}(1|x_i), \ldots, P_{Y|X}(M|x_i)]^{\mathrm{T}}$, and let*

$$
\boldsymbol{Q}(x^n) = \mathrm{diag}\{\boldsymbol{Q}(x_1), \ldots, \boldsymbol{Q}(x_n)\},
$$

*it holds that*

$$
\mathbb{E}_{P_{Y^n|X^n}}\left[\ell\left(x^n, Y^n; x_Q\right)|X^n = x^n\right] = \mathrm{tr}\left\{\frac{1}{n^2}\mathbf{F}_n^{-1}(x^n)\mathbf{F}(x_Q)\mathbf{F}_n^{-1}(x^n)f(x^n)\mathbf{Q}(x^n)f^{\mathrm{T}}(x^n)\right\}.
$$

*Proof.* Based on the Corollary F.5 and Lemma F.6, we have the excessive risk being

$$
\mathbb{E}_{P_{Y^n|X^n}}\left[\ell\left(x^n, Y^n; x_Q\right)|X^n = x^n\right]
$$
$$
= \mathrm{tr}\left\{\mathbf{F}_n^{-1}(x^n)\mathbf{F}(x_Q)\mathbf{F}_n^{-1}(x^n)\mathsf{F}(x^n)\right\} - \underline{\alpha}^{\mathrm{T}}\mathbf{F}(x_Q)\underline{\alpha}
$$
$$
= \mathrm{tr}\left\{\frac{1}{n^2}\mathbf{F}_n^{-1}(x^n)\mathbf{F}(x_Q)\mathbf{F}_n^{-1}(x^n)f(x^n)\tilde{\mathbf{Q}}(x^n)f^{\mathrm{T}}(x^n)\right\} - \underline{\alpha}^{\mathrm{T}}\mathbf{F}(x_Q)\underline{\alpha}
$$
$$
= \mathrm{tr}\left\{\frac{1}{n^2}\mathbf{F}_n^{-1}(x^n)\mathbf{F}(x_Q)\mathbf{F}_n^{-1}(x^n)f(x^n)\left(\tilde{\mathbf{Q}}(x^n) - \sum_{i,j} v_i v_j^{\mathrm{T}}\right)f^{\mathrm{T}}(x^n)\right\}.
$$

Note that $\tilde{\mathbf{Q}}(x^n) = \sum_{i,j} v_i v_j^{\mathrm{T}} + \mathbf{Q}(x^n)$, we have that

$$
\mathbb{E}_{P_{Y^n|X^n}}\left[\ell\left(x^n, Y^n; x_Q\right)|X^n = x^n\right] = \mathrm{tr}\left\{\frac{1}{n^2}\mathbf{F}_n^{-1}(x^n)\mathbf{F}(x_Q)\mathbf{F}_n^{-1}(x^n)f(x^n)\mathbf{Q}(x^n)f^{\mathrm{T}}(x^n)\right\}.
$$

$\qquad\square$

Based on this form, we further find the upper bound.

**Lemma F.8.** *For sufficient prompt demonstrations and complete model, the excessive risk can be bounded by*

$$
\mathbb{E}_{P_{Y^n|X^n}}\left[\ell\left(x^n, Y^n; x_Q\right)|X^n = x^n\right] \leq \frac{1}{n}\lambda_1\left(\mathbf{F}(x_Q)\mathbf{F}_n^{-1}(x^n)\right)S_K(\boldsymbol{Q}),
$$

*where $S_K(\boldsymbol{Q})$ is the sum of top K eigenvalues.*

*Proof.* Denote $\mathbf{F}_n^{-1/2}(x^n)$ as $\mathbf{F}_n^{-1}(x^n) = \mathbf{F}_n^{-1/2}(x^n)\mathbf{F}_n^{-1/2}(x^n)$. Then, by Corollary F.7, the excessive risk can be written as

$$\mathbb{E}_{P_{Y^n|X^n}}\left[\ell\left(x^n, Y^n; x_Q\right)|X^n = x^n\right]$$

$$= \mathrm{tr}\left\{\frac{1}{n^2}\mathbf{F}_n^{-1/2}(x^n)\mathbf{F}(x_Q)\mathbf{F}_n^{-1/2}(x^n)\mathbf{F}_n^{-1/2}(x^n)f(x^n)\mathbf{Q}(x^n)f^{\mathrm{T}}(x^n)\mathbf{F}_n^{-1/2}(x^n)\right\}.$$

Denote the $i$-th large eigenvalue for one matrix as $\lambda_i(\cdot)$. Note that both matrices $\mathbf{F}_n^{-1/2}(x^n)\mathbf{F}(x_Q)\mathbf{F}_n^{-1/2}(x^n)$ and $\mathbf{F}_n^{-1/2}(x^n)f(x^n)\mathbf{Q}(x^n)f^{\mathrm{T}}(x^n)\mathbf{F}_n^{-1/2}(x^n)$ are Hermitian. From Von Neumann's trace inequality (Mirsky, 1975), we have the

$$\mathrm{tr}\left\{\frac{1}{n^2}\mathbf{F}_n^{-1/2}(x^n)\mathbf{F}(x_Q)\mathbf{F}_n^{-1/2}(x^n)\mathbf{F}_n^{-1/2}(x^n)f(x^n)\mathbf{Q}(x^n)f^{\mathrm{T}}(x^n)\mathbf{F}_n^{-1/2}(x^n)\right\}$$

$$\leq \frac{1}{n}\sum_{i=1}^{K}\lambda_i\left(\mathbf{F}_n^{-1/2}(x^n)\mathbf{F}(x_Q)\mathbf{F}_n^{-1/2}(x^n)\right)\lambda_i\left(\frac{1}{\sqrt{n}}\mathbf{F}_n^{-1/2}(x^n)f(x^n)\mathbf{Q}(x^n)f^{\mathrm{T}}(x^n)\mathbf{F}_n^{-1/2}(x^n)\frac{1}{\sqrt{n}}\right).$$

Since that

$$\left(\frac{1}{\sqrt{n}}\mathbf{F}_n^{-1/2}(x^n)f(x^n)\right)\left(\frac{1}{\sqrt{n}}\mathbf{F}_n^{-1/2}(x^n)f(x^n)\right)^{\mathrm{T}}$$

$$= \frac{1}{\sqrt{n}}\mathbf{F}_n^{-1/2}(x^n)f(x^n)f^{\mathrm{T}}(x^n)\mathbf{F}_n^{-1/2}(x^n)\frac{1}{\sqrt{n}}$$

$$= \mathbf{F}_n^{-1/2}(x^n)\mathbf{F}_n(x^n)\mathbf{F}_n^{-1/2}(x^n)$$

$$= I,$$

the matrix $\frac{1}{\sqrt{n}}\mathbf{F}_n^{-1/2}(x^n)f(x^n)$ is an orthogonal matrix. Then, the eigenvalues of matrix $\frac{1}{\sqrt{n}}\mathbf{F}_n^{-1/2}(x^n)f(x^n)\mathbf{Q}(x^n)f^{\mathrm{T}}(x^n)\mathbf{F}_n^{-1/2}(x^n)\frac{1}{\sqrt{n}}$ equal to those of $\mathbf{Q}(x^n)$, i.e.,

$$\lambda_i\left(\frac{1}{\sqrt{n}}\mathbf{F}_n^{-1/2}(x^n)f(x^n)\mathbf{Q}(x^n)f^{\mathrm{T}}(x^n)\mathbf{F}_n^{-1/2}(x^n)\frac{1}{\sqrt{n}}\right) = \lambda_i\left(\mathbf{Q}(x^n)\right).$$

Furthermore, we prove that $\mathbf{Q}(x^n)$ is semi-positive definite. For any vector $v = [v_1^{\mathrm{T}}, \ldots, v_n^{\mathrm{T}}]^{\mathrm{T}}$, we have that

$$v^{\mathrm{T}}\mathbf{Q}(x^n)v = \sum_{i=1}^{n} v_i^{\mathrm{T}}\mathbf{Q}(x_i)v_i = \sum_{i=1}^{n}\sum_{j=1}^{M} P_{Y|X}(j|x_i)v_i^2(j) - \sum_{i=1}^{n}\left(\sum_{j=1}^{M} P_{Y|X}(j|x_i)v_i(j)\right)^2 \geq 0,$$

where the inequality is achieved by Jensen's inequality. Hence, all eigenvalues of $\mathbf{Q}(x^n)$ are non-negative, making that

$$\mathrm{tr}\left\{\frac{1}{n^2}\mathbf{F}_n^{-1/2}(x^n)\mathbf{F}(x_Q)\mathbf{F}_n^{-1/2}(x^n)\mathbf{F}_n^{-1/2}(x^n)f(x^n)\mathbf{Q}(x^n)f^{\mathrm{T}}(x^n)\mathbf{F}_n^{-1/2}(x^n)\right\}$$

$$\leq \frac{1}{n}\sum_{i=1}^{K}\lambda_i\left(\mathbf{F}_n^{-1/2}(x^n)\mathbf{F}(x_Q)\mathbf{F}_n^{-1/2}(x^n)\right)\lambda_i\left(\frac{1}{\sqrt{n}}\mathbf{F}_n^{-1/2}(x^n)f(x^n)\mathbf{Q}(x^n)f^{\mathrm{T}}(x^n)\mathbf{F}_n^{-1/2}(x^n)\frac{1}{\sqrt{n}}\right)$$

$$\leq \frac{1}{n}\lambda_1\left(\mathbf{F}_n^{-1/2}(x^n)\mathbf{F}(x_Q)\mathbf{F}_n^{-1/2}(x^n)\right)\sum_{i=1}^{K}\lambda_i\left(\mathbf{Q}(x^n)\right)$$

$$= \frac{1}{n}\lambda_1\left(\mathbf{F}(x_Q)\mathbf{F}_n^{-1}(x^n)\right)S_K(\mathbf{Q}).$$

$\square$

Based on Lemma F.8, we prove the Theorem 4.2 by

$$S_K(\mathbf{Q}(x^n)) \leq K\lambda_1(\mathbf{Q}(x^n)).$$

F.3   PROOF OF THEOREM 4.3

Similar to Lemma F.4, we have the expectation of $\underline{\alpha}$ under insufficient demonstrations.

**Lemma F.9.** *For insufficient prompt demonstrations, i.e., $\mathbf{F}_n(x^n)$ is not invertible, it holds that for all $x^n$,*

$$\mathbb{E}_{P_{Y^n|X^n}}[\underline{\hat{\alpha}}(x^n, Y^n)|X^n = x^n] = \mathbf{F}_n^\dagger(x^n)\mathbf{F}_n(x^n)\underline{\alpha}.$$

*Proof.* From definition of $\underline{\hat{\alpha}}$ in Eq. (2), we have that

$$\mathbb{E}_{P_{Y^n|X^n}}\left[\underline{\hat{\alpha}}(x^n, Y^n)|X^n = x^n\right]$$

$$= \mathbb{E}_{P_{Y^n|X^n}}\left[\mathbf{F}_n^\dagger(x^n)\bar{f}_n(x^n, Y^n)|X^n = x^n\right]$$

$$= \sum_{y^n}\prod_{i=1}^n P_{Y|X}(y_i|x_i)\mathbf{F}_n^\dagger(x^n)\left(\frac{1}{n}\sum_{i=1}^n f(x_i, y_i)\right)$$

$$= \mathbf{F}_n^\dagger(x^n)\frac{1}{n}\sum_{i=1}^n\sum_y P_{Y|X}(y|x_i)f(x_i, y)$$

$$= \mathbf{F}_n^\dagger(x^n)\frac{1}{n}\sum_{i=1}^n\sum_y f(x_i, y)f^{\mathrm{T}}(x_i, y)\underline{\alpha}$$

$$= \mathbf{F}_n^\dagger(x^n)\mathbf{F}_n(x^n)\underline{\alpha}.$$

$\square$

**Corollary F.10.** *For insufficient prompt demonstrations and complete model, the excessive risk can be written as*

$$\mathbb{E}_{P_{Y^n|X^n}}\left[\ell\left(x^n, Y^n; x_Q\right)|X^n = x^n\right]$$

$$= \mathrm{tr}\left\{\mathbf{F}_n^\dagger(x^n)\mathsf{F}(x_Q)\mathbf{F}_n^\dagger(x^n)\mathsf{F}(x^n)\right\} - 2\underline{\alpha}^{\mathrm{T}}\mathsf{F}(x_Q)\mathbf{F}_n^\dagger(x^n)\mathbf{F}_n(x^n)\underline{\alpha} + \underline{\alpha}^{\mathrm{T}}\mathsf{F}(x_Q)\underline{\alpha},$$

*where $\mathsf{F}(x^n) = \mathbb{E}_{P_{Y^n|X^n}}\left[\bar{f}_n(x^n, Y^n)\bar{f}_n^{\mathrm{T}}(x^n, Y^n)|X^n = x^n\right]$.*

*Proof.* From the definition of $\ell\left(x^n, y^n; x_Q\right)$, we have that (note that $\mathbf{F}_n^{-1}(x^n)$ is symmetric)

$$\mathbb{E}_{P_{Y^n|X^n}}\left[\ell\left(x^n, Y^n; x_Q\right)|X^n = x^n\right]$$

$$= \mathbb{E}_{P_{Y^n|X^n}}\left[\sum_y\left(\hat{P}_{Y|X}(y|x_Q) - P_{Y|X}(y|x_Q)\right)^2|X^n = x^n\right]$$

$$= \mathbb{E}_{P_{Y^n|X^n}}\left[\sum_y\left(\underline{\hat{\alpha}}^{\mathrm{T}}f(x_Q, y) - \underline{\alpha}^{\mathrm{T}}f(x_Q, y)\right)^2|X^n = x^n\right]$$

$$= \mathbb{E}_{P_{Y^n|X^n}}\left[(\underline{\hat{\alpha}} - \underline{\alpha})^{\mathrm{T}}\mathbf{F}(x_Q)(\underline{\hat{\alpha}} - \underline{\alpha})|X^n = x^n\right]$$

$$= \mathbb{E}_{P_{Y^n|X^n}}\left[\underline{\hat{\alpha}}^{\mathrm{T}}\mathbf{F}(x_Q)\underline{\hat{\alpha}}|X^n = x^n\right] - 2\mathbb{E}_{P_{Y^n|X^n}}\left[\underline{\hat{\alpha}}^{\mathrm{T}}\mathbf{F}(x_Q)\underline{\alpha}|X^n = x^n\right] + \underline{\alpha}^{\mathrm{T}}\mathbf{F}(x_Q)\underline{\alpha}$$

$$= \mathrm{tr}\left\{\mathbf{F}_n^\dagger(x^n)\mathbf{F}(x_Q)\mathbf{F}_n^\dagger(x^n)\mathsf{F}(x^n)\right\} - 2\underline{\alpha}^{\mathrm{T}}\mathbf{F}(x_Q)\mathbf{F}_n^\dagger(x^n)\mathbf{F}_n(x^n)\underline{\alpha} + \underline{\alpha}^{\mathrm{T}}\mathbf{F}(x_Q)\underline{\alpha}.$$

$\square$

Write $\mathsf{F}(x^n)$ as

$$\mathsf{F}(x^n) = \frac{1}{n^2}f(x^n)\tilde{\mathbf{Q}}(x^n)f^{\mathrm{T}}(x^n),$$

where

$$\tilde{\mathbf{Q}}(x^n) = \mathsf{diag}\{\mathsf{diag}\{P_{Y|X=x_1}\}, \ldots, \mathsf{diag}\{P_{Y|X=x_n}\}\} + \sum_{i=1}^n\sum_{j=1}^n v_iv_j^{\mathrm{T}} - \sum_i v_iv_i^{\mathrm{T}},$$

with $\mathsf{diag}\{P_{Y|X=x_i}\} = \mathsf{diag}\{P_{Y|X}(1|x_i), \ldots, P_{Y|X}(M|x_i)\}$, and

$$v_i = \left[0, \ldots, 0, P_{Y|X}(1|x_i), \ldots, P_{Y|X}(M|x_i), 0\ldots, 0\right]^{\mathrm{T}}.$$

**Lemma F.11.** *For an insufficient demonstrations and complete model, it holds that*

$$\frac{1}{n^2} \text{tr} \left\{ \mathbf{F}_n^\dagger(x^n) \mathbf{F}(x_Q) \mathbf{F}_n^\dagger(x^n) f(x^n) \left( \sum_{i,j} v_i v_j^\mathsf{T} \right) f^\mathsf{T}(x^n) \right\}$$

$$= \left( \mathbf{F}_n^\dagger(x^n) \mathbf{F}_n(x^n) \underline{\alpha} \right)^\mathsf{T} \mathbf{F}(x_Q) \mathbf{F}_n^\dagger(x^n) \mathbf{F}_n(x^n) \underline{\alpha}.$$

*Proof.* We have that

$$\frac{1}{n^2} \text{tr} \left\{ \mathbf{F}_n^\dagger(x^n) \mathbf{F}(x_Q) \mathbf{F}_n^\dagger(x^n) f(x^n) \left( \sum_{i,j} v_i v_j^\mathsf{T} \right) f^\mathsf{T}(x^n) \right\}$$

$$= \text{tr} \left\{ \mathbf{F}_n^\dagger(x^n) \mathbf{F}(x_Q) \mathbf{F}_n^\dagger(x^n) \left( \frac{1}{n} \sum_{i,y} P_{Y|X}(y|x_i) f(x_i, y) \right) \left( \frac{1}{n} \sum_{j,y} P_{Y|X}(y|x_j) f(x_j, y) \right)^\mathsf{T} \right\}$$

From model 1, we have that for a complete model, $P_{Y|X}(y|x) = \underline{\alpha}^\mathsf{T} f(x, y)$. Hence, we have that

$$\text{tr} \left\{ \mathbf{F}_n^\dagger(x^n) \mathbf{F}(x_Q) \mathbf{F}_n^\dagger(x^n) \left( \frac{1}{n} \sum_{i,y} P_{Y|X}(y|x_i) f(x_i, y) \right) \left( \frac{1}{n} \sum_{j,y} P_{Y|X}(y|x_j) f(x_j, y) \right)^\mathsf{T} \right\}$$

$$= \text{tr} \left\{ \mathbf{F}_n^\dagger(x^n) \mathbf{F}(x_Q) \mathbf{F}_n^\dagger(x^n) \left( \frac{1}{n} \sum_{i,y} f(x_i, y) f^\mathsf{T}(x_i, y) \underline{\alpha} \right) \left( \frac{1}{n} \sum_{j,y} f(x_j, y) f^\mathsf{T}(x_j, y) \underline{\alpha} \right)^\mathsf{T} \right\}$$

$$= \text{tr} \left\{ \mathbf{F}_n^\dagger(x^n) \mathbf{F}(x_Q) \mathbf{F}_n^\dagger(x^n) \mathbf{F}_n(x^n) \underline{\alpha} \left( \mathbf{F}_n(x^n) \underline{\alpha} \right)^\mathsf{T} \right\}$$

$$= \left( \mathbf{F}_n^\dagger(x^n) \mathbf{F}_n(x^n) \underline{\alpha} \right)^\mathsf{T} \mathbf{F}(x_Q) \mathbf{F}_n^\dagger(x^n) \mathbf{F}_n(x^n) \underline{\alpha}.$$

This completes the proof. □

Hence, we have the excessive risk in a simplified form.

**Corollary F.12.** *Let*

$$\boldsymbol{Q}(x_i) = \text{diag}\{P_{Y|X}(1|x_i), \dots, P_{Y|X}(M|x_i)\} - \phi_i \phi_i^\mathsf{T},$$

*where $\phi_i = [P_{Y|X}(1|x_i), \dots, P_{Y|X}(M|x_i)]^\mathsf{T}$, and let*

$$\boldsymbol{Q}(x^n) = \text{diag}\{\boldsymbol{Q}(x_1), \dots, \boldsymbol{Q}(x_n)\},$$

*it holds that*

$$\mathbb{E}_{P_{Y^n|X^n}} \left[ \ell(x^n, Y^n; x_Q) \,|\, X^n = x^n \right]$$

$$= \text{tr} \left\{ \frac{1}{n^2} \mathbf{F}_n^\dagger(x^n) \mathbf{F}(x_Q) \mathbf{F}_n^\dagger(x^n) f(x^n) \mathbf{Q}(x^n) f^\mathsf{T}(x^n) \right\} + \left\| f^\mathsf{T}(x_Q) \mathbf{F}_n^\perp(x^n) \underline{\alpha} \right\|^2,$$

*where $\mathbf{F}_n^\perp(x^n) = \boldsymbol{I}_K - \mathbf{F}_n^\dagger(x^n) \mathbf{F}_n(x^n)$.*

*Proof.* Based on the Corollary F.5 and Lemma F.6, we have the excessive risk being

$$\mathbb{E}_{P_{Y^n|X^n}} \left[ \ell(x^n, Y^n; x_Q) \,|\, X^n = x^n \right]$$

$$= \text{tr} \left\{ \mathbf{F}_n^\dagger(x^n) \mathbf{F}(x_Q) \mathbf{F}_n^\dagger(x^n) \mathsf{F}(x^n) \right\} - 2\underline{\alpha}^\mathsf{T} \mathbf{F}(x_Q) \mathbf{F}_n^\dagger(x^n) \mathbf{F}_n(x^n) \underline{\alpha} + \underline{\alpha}^\mathsf{T} \mathbf{F}(x_Q) \underline{\alpha}.$$

$$= \text{tr} \left\{ \frac{1}{n^2} \mathbf{F}_n^{-1}(x^n) \mathbf{F}(x_Q) \mathbf{F}_n^{-1}(x^n) f(x^n) \left( \tilde{\mathbf{Q}}(x^n) - \sum_{i,j} v_i v_j^\mathsf{T} \right) f^\mathsf{T}(x^n) \right\}$$

$$+ \left( \mathbf{F}_n^\dagger(x^n) \mathbf{F}_n(x^n) \underline{\alpha} \right)^\mathsf{T} \mathbf{F}(x_Q) \mathbf{F}_n^\dagger(x^n) \mathbf{F}_n(x^n) \underline{\alpha} - 2\underline{\alpha}^\mathsf{T} \mathbf{F}(x_Q) \mathbf{F}_n^\dagger(x^n) \mathbf{F}_n(x^n) \underline{\alpha} + \underline{\alpha}^\mathsf{T} \mathbf{F}(x_Q) \underline{\alpha}..$$

Note that $\tilde{\mathbf{Q}}(x^n) = \sum_{i,j} v_i v_j^{\mathrm{T}} + \mathbf{Q}(x^n)$ and $\mathbf{F}(x_Q) = f(x_Q)f^{\mathrm{T}}(x_Q)$, we have that

$$\mathbb{E}_{P_{Y^n|X^n}} \left[ \ell(x^n, Y^n; x_Q) | X^n = x^n \right]$$

$$= \mathrm{tr} \left\{ \frac{1}{n^2} \mathbf{F}_n^{\dagger}(x^n) \mathbf{F}(x_Q) \mathbf{F}_n^{\dagger}(x^n) f(x^n) \mathbf{Q}(x^n) f^{\mathrm{T}}(x^n) \right\} + \left\| f^{\mathrm{T}}(x_Q) \mathbf{F}_n^{\perp}(x^n) \underline{\alpha} \right\|^2.$$

$\square$

**Lemma F.13.** *For sufficient prompt demonstrations and complete model, the excessive risk can is bounded by*

$$\mathbb{E}_{P_{Y^n|X^n}} \left[ \ell(x^n, Y^n; x_Q) | X^n = x^n \right] \leq \frac{1}{n} \lambda_1 \left( \mathbf{F}(x_Q) \mathbf{F}_n^{\dagger}(x^n) \right) S_K(\boldsymbol{Q}(x^n)) + \left\| f^{\mathrm{T}}(x_Q) \mathbf{F}_n^{\perp}(x^n) \underline{\alpha} \right\|^2,$$

*where $S_K(\boldsymbol{Q}(x^n))$ is the sum of top K eigenvalues.*

*Proof.* Denote $\mathbf{F}_n^{\dagger}(x^n)^{1/2}$ as $\mathbf{F}_n^{\dagger}(x^n) = \mathbf{F}_n^{\dagger}(x^n)^{1/2} \mathbf{F}_n^{\dagger}(x^n)^{1/2}$. Then, by Corollary F.12, the excessive risk can be written as

$$\mathbb{E}_{P_{Y^n|X^n}} \left[ \ell(x^n, Y^n; x_Q) | X^n = x^n \right]$$

$$= \mathrm{tr} \left\{ \frac{1}{n^2} \mathbf{F}_n^{\dagger}(x^n)^{1/2}(x^n) \mathbf{F}(x_Q) \mathbf{F}_n^{\dagger}(x^n)^{1/2} \mathbf{F}_n^{\dagger}(x^n)^{1/2} f(x^n) \mathbf{Q}(x^n) f^{\mathrm{T}}(x^n) \mathbf{F}_n^{\dagger}(x^n)^{1/2} \right\}$$

$$+ \left\| f^{\mathrm{T}}(x_Q) \mathbf{F}_n^{\perp}(x^n) \underline{\alpha} \right\|^2.$$

From Von Neumann's trace inequality (Mirsky, 1975), denote the $i$-th large eigenvalue for one matrix as $\lambda_i(\cdot)$, since both matrices $\mathbf{F}_n^{\dagger}(x^n)^{1/2} \mathbf{F}(x_Q) \mathbf{F}_n^{\dagger}(x^n)^{1/2}$ and $\mathbf{F}_n^{\dagger}(x^n)^{1/2}(x^n) f(x^n) \mathbf{Q}(x^n) f^{\mathrm{T}}(x^n) \mathbf{F}_n^{\dagger}(x^n)^{1/2}$ are Hermitian, we have that

$$\mathrm{tr} \left\{ \frac{1}{n^2} \mathbf{F}_n^{\dagger}(x^n)^{1/2} \mathbf{F}(x_Q) \mathbf{F}_n^{\dagger}(x^n)^{1/2} \mathbf{F}_n^{\dagger}(x^n)^{1/2} f(x^n) \mathbf{Q}(x^n) f^{\mathrm{T}}(x^n) \mathbf{F}_n^{\dagger}(x^n)^{1/2} \right\}$$

$$\leq \frac{1}{n} \sum_{i=1}^{K} \lambda_i \left( \mathbf{F}_n^{\dagger}(x^n)^{1/2} \mathbf{F}(x_Q) \mathbf{F}_n^{\dagger}(x^n)^{1/2} \right) \lambda_i \left( \frac{1}{\sqrt{n}} \mathbf{F}_n^{\dagger}(x^n)^{1/2} f(x^n) \mathbf{Q}(x^n) f^{\mathrm{T}}(x^n) \mathbf{F}_n^{\dagger}(x^n)^{1/2} \frac{1}{\sqrt{n}} \right).$$

Since that

$$\left( \frac{1}{\sqrt{n}} \mathbf{F}_n^{\dagger}(x^n)^{1/2} f(x^n) \right) \left( \frac{1}{\sqrt{n}} \mathbf{F}_n^{\dagger}(x^n)^{1/2} f(x^n) \right)^{\mathrm{T}}$$

$$= \frac{1}{\sqrt{n}} \mathbf{F}_n^{\dagger}(x^n)^{1/2} f(x^n) f^{\mathrm{T}}(x^n) \mathbf{F}_n^{\dagger}(x^n)^{1/2} \frac{1}{\sqrt{n}}$$

$$= \mathbf{F}_n^{\dagger}(x^n)^{1/2} \mathbf{F}_n(x^n) \mathbf{F}_n^{\dagger}(x^n)^{1/2}$$

is a projection matrix, the eigenvalues of matrix $\frac{1}{\sqrt{n}} \mathbf{F}_n^{\dagger}(x^n)^{1/2} f(x^n) \mathbf{Q}(x^n) f^{\mathrm{T}}(x^n) \mathbf{F}_n^{\dagger}(x^n)^{1/2} \frac{1}{\sqrt{n}}$ equals to that of $\mathbf{Q}(x^n)$, i.e.,

$$\lambda_i \left( \frac{1}{\sqrt{n}} \mathbf{F}_n^{\dagger}(x^n)^{1/2} f(x^n) \mathbf{Q}(x^n) f^{\mathrm{T}}(x^n) \mathbf{F}_n^{\dagger}(x^n)^{1/2} \frac{1}{\sqrt{n}} \right) = \lambda_i \left( \mathbf{Q}(x^n) \right).$$

As is proved in Lemma F.8, all eigenvalues of $\mathbf{Q}(x^n)$ are non-nagetive, makeing that

$$\mathrm{tr} \left\{ \frac{1}{n^2} \mathbf{F}_n^{\dagger}(x^n)^{1/2} \mathbf{F}(x_Q) \mathbf{F}_n^{\dagger}(x^n)^{1/2} \mathbf{F}_n^{\dagger}(x^n)^{1/2} f(x^n) \mathbf{Q}(x^n) f^{\mathrm{T}}(x^n) \mathbf{F}_n^{\dagger}(x^n)^{1/2} \right\}$$

$$\leq \frac{1}{n} \sum_{i=1}^{K} \lambda_i \left( \mathbf{F}_n^{\dagger}(x^n)^{1/2} \mathbf{F}(x_Q) \mathbf{F}_n^{\dagger}(x^n)^{1/2} \right) \lambda_i \left( \frac{1}{\sqrt{n}} \mathbf{F}_n^{\dagger}(x^n)^{1/2} f(x^n) \mathbf{Q}(x^n) f^{\mathrm{T}}(x^n) \mathbf{F}_n^{\dagger}(x^n)^{1/2} \frac{1}{\sqrt{n}} \right)$$

$$\leq \frac{1}{n} \lambda_1 \left( \mathbf{F}_n^{\dagger}(x^n)^{1/2} \mathbf{F}(x_Q) \mathbf{F}_n^{\dagger}(x^n)^{1/2} \right) \sum_{i=1}^{K} \lambda_i \left( \mathbf{Q}(x^n) \right)$$

$$= \frac{1}{n} \lambda_1 \left( \mathbf{F}(x_Q) \mathbf{F}_n^{\dagger}(x^n) \right) S_K(\mathbf{Q}(x^n)).$$

$\square$

Based on Lemma F.13, we simply prove the Theorem 4.3 by

$$S_K(\mathbf{Q}(x^n)) \le K\lambda_1(\mathbf{Q}(x^n)).$$

### F.4 PROOF OF THEOREM 4.4

Similar to Lemma F.9, we have the expectation of $\underline{\alpha}$ under insufficient demonstrations with incomplete model.

**Lemma F.14.** *For insufficient prompt demonstrations, i.e., $\mathbf{F}_n(x^n)$ is not invertible, and incomplete model, i.e. $R(x,y) \ne 0$, it holds that for all $x^n$,*

$$\mathbb{E}_{P_{Y^n|X^n}}[\hat{\underline{\alpha}}(x^n, Y^n)|X^n = x^n] = \mathbf{F}_n^\dagger(x^n)\mathbf{F}_n(x^n)\underline{\alpha} + \frac{1}{n}\mathbf{F}_n^\dagger(x^n)f(x^n)\mathsf{R}(x^n).$$

*Proof.* From definition of $\hat{\underline{\alpha}}$ in Eq. (2), we have that

$$\mathbb{E}_{P_{Y^n|X^n}}[\hat{\underline{\alpha}}(x^n, Y^n)|X^n = x^n]$$

$$= \mathbb{E}_{P_{Y^n|X^n}}\left[\mathbf{F}_n^\dagger(x^n)\bar{f}_n(x^n, Y^n)|X^n = x^n\right]$$

$$= \mathbf{F}_n^\dagger(x^n)\frac{1}{n}\sum_{i=1}^n\sum_y P_{Y|X}(y|x_i)f(x_i, y)$$

$$= \mathbf{F}_n^\dagger(x^n)\frac{1}{n}\sum_{i=1}^n\sum_y f(x_i, y)f^{\mathrm{T}}(x_i, y)\underline{\alpha} + \mathbf{F}_n^\dagger(x^n)\frac{1}{n}\sum_{i=1}^n\sum_y f(x_i, y)R(x_i, y)$$

$$= \mathbf{F}_n^\dagger(x^n)\mathbf{F}_n(x^n)\underline{\alpha} + \frac{1}{n}\mathbf{F}_n^\dagger(x^n)f(x^n)\mathsf{R}(x^n).$$

$\square$

**Corollary F.15.** *For insufficient prompt demonstrations and incomplete model, the excessive risk can be written as*

$$\mathbb{E}_{P_{Y^n|X^n}}\left[\ell\left(x^n, Y^n; x_Q\right)|X^n = x^n\right]$$

$$= \mathrm{tr}\left\{\mathbf{F}_n^\dagger(x^n)\mathbf{F}(x_Q)\mathbf{F}_n^\dagger(x^n)\mathsf{F}(x^n)\right\} - 2\underline{\alpha}^{\mathrm{T}}\mathbf{F}(x_Q)\mathbf{F}_n^\dagger(x^n)\mathbf{F}_n(x^n)\underline{\alpha} + \underline{\alpha}^{\mathrm{T}}\mathbf{F}(x_Q)\underline{\alpha}$$

$$- \frac{2}{n}\mathsf{R}^{\mathrm{T}}(x^n)f^{\mathrm{T}}(x^n)\mathbf{F}_n^\dagger(x^n)\mathbf{F}(x_Q)\underline{\alpha} - 2\mathsf{R}^{\mathrm{T}}(x_Q)f^{\mathrm{T}}(x_Q)\mathbf{F}_n^\perp(x^n)\underline{\alpha}$$

$$+ \frac{2}{n}\mathsf{R}^{\mathrm{T}}(x_Q)f^{\mathrm{T}}(x_Q)\mathbf{F}_n^\dagger(x^n)f(x^n)\mathsf{R}(x^n) + \sum_y R^2(x_Q, y),$$

*where $\mathsf{F}(x^n) = \mathbb{E}_{P_{Y^n|X^n}}\left[\bar{f}_n(x^n, Y^n)\bar{f}_n^{\mathrm{T}}(x^n, Y^n)|X^n = x^n\right]$.*

*Proof.* From the definition of $\ell\left(x^n, y^n; x_Q\right)$, we have that (note that $\mathbf{F}_n^{-1}(x^n)$ is symmetric)

$$\mathbb{E}_{P_{Y^n|X^n}}\left[\ell\left(x^n, Y^n; x_Q\right)|X^n = x^n\right]$$

$$= \mathbb{E}_{P_{Y^n|X^n}}\left[\sum_y\left(\hat{\underline{\alpha}}^{\mathrm{T}}f(x_Q, y) - \underline{\alpha}^{\mathrm{T}}f(x_Q, y) + R(x_Q, y)\right)^2|X^n = x^n\right]$$

$$= \mathbb{E}_{P_{Y^n|X^n}}\left[(\hat{\underline{\alpha}} - \underline{\alpha})^{\mathrm{T}}\mathbf{F}(x_Q)(\hat{\underline{\alpha}} - \underline{\alpha})|X^n = x^n\right]$$

$$+ 2\mathbb{E}_{P_{Y^n|X^n}}\left[\sum_y(\hat{\underline{\alpha}} - \underline{\alpha})^{\mathrm{T}}f(x_Q, y)R(x_Q, y)|X^n = x^n\right] + \sum_y R^2(x_Q, y)$$

$$= \mathrm{tr}\left\{\mathbf{F}_n^\dagger(x^n)\mathbf{F}(x_Q)\mathbf{F}_n^\dagger(x^n)\mathsf{F}(x^n)\right\} - 2\underline{\alpha}^{\mathrm{T}}\mathbf{F}(x_Q)\mathbf{F}_n^\dagger(x^n)\mathbf{F}_n(x^n)\underline{\alpha} + \underline{\alpha}^{\mathrm{T}}\mathbf{F}(x_Q)\underline{\alpha}$$

$$- \frac{2}{n}\mathsf{R}^{\mathrm{T}}(x^n)f^{\mathrm{T}}(x^n)\mathbf{F}_n^\dagger(x^n)\mathbf{F}(x_Q)\underline{\alpha} - 2\mathsf{R}^{\mathrm{T}}(x_Q)f^{\mathrm{T}}(x_Q)\mathbf{F}_n^\perp(x^n)\underline{\alpha}$$

$$+ \frac{2}{n}\mathsf{R}^{\mathrm{T}}(x_Q)f^{\mathrm{T}}(x_Q)\mathbf{F}_n^\dagger(x^n)f(x^n)\mathsf{R}(x^n) + \sum_y R^2(x_Q, y).$$

$\square$

**Lemma F.16.** *For an insufficient demonstrations and an incomplete model, it holds that*

$$\frac{1}{n^2}\mathrm{tr}\left\{\mathbf{F}_n^\dagger(x^n)\mathbf{F}(x_Q)\mathbf{F}_n^\dagger(x^n)f(x^n)\left(\sum_{i,j}v_iv_j^\mathrm{T}\right)f^\mathrm{T}(x^n)\right\}$$

$$= \left(\mathbf{F}_n^\dagger(x^n)\mathbf{F}_n(x^n)\underline{\alpha}\right)^\mathrm{T}\mathbf{F}(x_Q)\mathbf{F}_n^\dagger(x^n)\mathbf{F}_n(x^n)\underline{\alpha}$$

$$+ \frac{2}{n}\mathsf{R}^\mathrm{T}(x^n)f^\mathrm{T}(x^n)\mathbf{F}_n^\dagger(x^n)\mathbf{F}(x_Q)\mathbf{F}_n^\dagger(x^n)\mathbf{F}_n(x^n)\underline{\alpha}$$

$$+ \frac{1}{n^2}\mathsf{R}^\mathrm{T}(x^n)f^\mathrm{T}(x^n)\mathbf{F}_n^\dagger(x^n)\mathbf{F}(x_Q)\mathbf{F}_n^\dagger(x^n)f(x^n)\mathsf{R}(x^n).$$

*Proof.* We have that

$$\frac{1}{n^2}\mathrm{tr}\left\{\mathbf{F}_n^\dagger(x^n)\mathbf{F}(x_Q)\mathbf{F}_n^\dagger(x^n)f(x^n)\left(\sum_{i,j}v_iv_j^\mathrm{T}\right)f^\mathrm{T}(x^n)\right\}$$

$$= \mathrm{tr}\left\{\mathbf{F}_n^\dagger(x^n)\mathbf{F}(x_Q)\mathbf{F}_n^\dagger(x^n)\left(\frac{1}{n}\sum_{i,y}P_{Y|X}(y|x_i)f(x_i,y)\right)\left(\frac{1}{n}\sum_{j,y}P_{Y|X}(y|x_j)f(x_j,y)\right)^\mathrm{T}\right\}$$

From model 1, we have that for an incomplete model, $P_{Y|X}(y|x) = \underline{\alpha}^\mathrm{T}f(x,y) + R(x,y)$. Hence, we have that

$$\mathrm{tr}\left\{\mathbf{F}_n^\dagger(x^n)\mathbf{F}(x_Q)\mathbf{F}_n^\dagger(x^n)\left(\frac{1}{n}\sum_{i,y}P_{Y|X}(y|x_i)f(x_i,y)\right)\left(\frac{1}{n}\sum_{j,y}P_{Y|X}(y|x_j)f(x_j,y)\right)^\mathrm{T}\right\}$$

$$= \mathrm{tr}\left\{\mathbf{F}_n^\dagger(x^n)\mathbf{F}(x_Q)\mathbf{F}_n^\dagger(x^n)\left(\frac{1}{n}\sum_{i,y}f(x_i,y)(f^\mathrm{T}(x_i,y)\underline{\alpha} + R(x_i,y))\right)\right.$$

$$\left.\left(\frac{1}{n}\sum_{j,y}f(x_j,y)(f^\mathrm{T}(x_j,y)\underline{\alpha} + R(x_i,y))\right)^\mathrm{T}\right\}$$

$$= \mathrm{tr}\left\{\mathbf{F}_n^\dagger(x^n)\mathbf{F}(x_Q)\mathbf{F}_n^\dagger(x^n)\mathbf{F}_n(x^n)\underline{\alpha}\left(\mathbf{F}_n(x^n)\underline{\alpha}\right)^\mathrm{T}\right.$$

$$+ \frac{2}{n}\mathbf{F}_n^\dagger(x^n)\mathbf{F}(x_Q)\mathbf{F}_n^\dagger(x^n)\mathbf{F}_n(x^n)\underline{\alpha}f^\mathrm{T}(x^n)\mathsf{R}^\mathrm{T}(x^n)$$

$$\left.+ \frac{1}{n^2}\mathbf{F}_n^\dagger(x^n)\mathbf{F}(x_Q)\mathbf{F}_n^\dagger(x^n)f(x^n)\mathsf{R}(x^n)f^\mathrm{T}(x^n)\mathsf{R}^\mathrm{T}(x^n)\right\}$$

$$= \left(\mathbf{F}_n^\dagger(x^n)\mathbf{F}_n(x^n)\underline{\alpha}\right)^\mathrm{T}\mathbf{F}(x_Q)\mathbf{F}_n^\dagger(x^n)\mathbf{F}_n(x^n)\underline{\alpha}$$

$$+ \frac{2}{n}\mathsf{R}^\mathrm{T}(x^n)f^\mathrm{T}(x^n)\mathbf{F}_n^\dagger(x^n)\mathbf{F}(x_Q)\mathbf{F}_n^\dagger(x^n)\mathbf{F}_n(x^n)\underline{\alpha}$$

$$+ \frac{1}{n^2}\mathsf{R}^\mathrm{T}(x^n)f^\mathrm{T}(x^n)\mathbf{F}_n^\dagger(x^n)\mathbf{F}(x_Q)\mathbf{F}_n^\dagger(x^n)f(x^n)\mathsf{R}(x^n).$$

This completes the proof. $\square$

Hence, we have the excessive risk in a simplified form.

**Corollary F.17.** *Let*

$$\boldsymbol{Q}(x_i) = \mathsf{diag}\{P_{Y|X}(1|x_i),\ldots,P_{Y|X}(M|x_i)\} - \phi_i\phi_i^\mathrm{T},$$

*where $\phi_i = [P_{Y|X}(1|x_i), \ldots, P_{Y|X}(M|x_i)]^{\mathrm{T}}$, and let*

$$\boldsymbol{Q}(x^n) = \mathsf{diag}\left\{\boldsymbol{Q}(x_1), \ldots, \boldsymbol{Q}(x_n)\right\},$$

*it holds that*

$$\mathbb{E}_{P_{Y^n|X^n}}\left[\ell\left(x^n, Y^n; x_Q\right) | X^n = x^n\right]$$

$$= \mathrm{tr}\left\{\frac{1}{n^2}\mathbf{F}_n^\dagger(x^n)\mathbf{F}(x_Q)\mathbf{F}_n^\dagger(x^n)f(x^n)\mathbf{Q}(x^n)f^{\mathrm{T}}(x^n)\right\}$$

$$+ \left\|f^{\mathrm{T}}(x_Q)\mathbf{F}_n^\perp(x^n)\underline{\alpha} - \mathsf{R}(x_Q) - \frac{1}{n}f^{\mathrm{T}}(x_Q)\mathbf{F}_n^\dagger(x^n)f(x^n)\mathsf{R}(x^n)\right\|^2,$$

*where $\mathbf{F}_n^\perp(x^n) = \boldsymbol{I}_K - \mathbf{F}_n^\dagger(x^n)\mathbf{F}_n(x^n)$.*

*Proof.* Based on the Corollary F.5 and Lemma F.6, we have the excessive risk being

$$\mathbb{E}_{P_{Y^n|X^n}}\left[\ell\left(x^n, Y^n; x_Q\right) | X^n = x^n\right]$$

$$= \mathrm{tr}\left\{\mathbf{F}_n^\dagger(x^n)\mathbf{F}(x_Q)\mathbf{F}_n^\dagger(x^n)\mathsf{F}(x^n)\right\} - 2\underline{\alpha}^{\mathrm{T}}\mathbf{F}(x_Q)\mathbf{F}_n^\dagger(x^n)\mathbf{F}_n(x^n)\underline{\alpha} + \underline{\alpha}^{\mathrm{T}}\mathbf{F}(x_Q)\underline{\alpha}.$$

$$= \mathrm{tr}\left\{\frac{1}{n^2}\mathbf{F}_n^{-1}(x^n)\mathbf{F}(x_Q)\mathbf{F}_n^{-1}(x^n)f(x^n)\left(\tilde{\mathbf{Q}}(x^n) - \sum_{i,j}v_i v_j^{\mathrm{T}}\right)f^{\mathrm{T}}(x^n)\right\}$$

$$+ \left(\mathbf{F}_n^\dagger(x^n)\mathbf{F}_n(x^n)\underline{\alpha}\right)^{\mathrm{T}}\mathbf{F}(x_Q)\mathbf{F}_n^\dagger(x^n)\mathbf{F}_n(x^n)\underline{\alpha} - 2\underline{\alpha}^{\mathrm{T}}\mathbf{F}(x_Q)\mathbf{F}_n^\dagger(x^n)\mathbf{F}_n(x^n)\underline{\alpha} + \underline{\alpha}^{\mathrm{T}}\mathbf{F}(x_Q)\underline{\alpha}$$

$$+ \frac{2}{n}\mathsf{R}^{\mathrm{T}}(x^n)f^{\mathrm{T}}(x^n)\mathbf{F}_n^\dagger(x^n)\mathbf{F}(x_Q)\mathbf{F}_n^\dagger(x^n)\mathbf{F}_n(x^n)\underline{\alpha}$$

$$+ \frac{1}{n^2}\mathsf{R}^{\mathrm{T}}(x^n)f^{\mathrm{T}}(x^n)\mathbf{F}_n^\dagger(x^n)\mathbf{F}(x_Q)\mathbf{F}_n^\dagger(x^n)f(x^n)\mathsf{R}(x^n)$$

$$- \frac{2}{n}\mathsf{R}^{\mathrm{T}}(x^n)f^{\mathrm{T}}(x^n)\mathbf{F}_n^\dagger(x^n)\mathbf{F}(x_Q)\underline{\alpha} - 2\mathsf{R}^{\mathrm{T}}(x_Q)f^{\mathrm{T}}(x_Q)\mathbf{F}_n^\perp(x^n)\underline{\alpha}$$

$$+ \frac{2}{n}\mathsf{R}^{\mathrm{T}}(x_Q)f^{\mathrm{T}}(x_Q)\mathbf{F}_n^\dagger(x^n)f(x^n)\mathsf{R}(x^n) + \sum_y R^2(x_Q, y).$$

Note that $\tilde{\mathbf{Q}}(x^n) = \sum_{i,j}v_i v_j^{\mathrm{T}} + \mathbf{Q}(x^n)$ and $\mathbf{F}(x_Q) = f(x_Q)f^{\mathrm{T}}(x_Q)$, we have that

$$\mathbb{E}_{P_{Y^n|X^n}}\left[\ell\left(x^n, Y^n; x_Q\right) | X^n = x^n\right]$$

$$= \mathrm{tr}\left\{\frac{1}{n^2}\mathbf{F}_n^\dagger(x^n)\mathbf{F}(x_Q)\mathbf{F}_n^\dagger(x^n)f(x^n)\mathbf{Q}(x^n)f^{\mathrm{T}}(x^n)\right\}$$

$$+ \left\|f^{\mathrm{T}}(x_Q)\mathbf{F}_n^\perp(x^n)\underline{\alpha} - \mathsf{R}(x_Q) - \frac{1}{n}f^{\mathrm{T}}(x_Q)\mathbf{F}_n^\dagger(x^n)f(x^n)\mathsf{R}(x^n)\right\|^2.$$

$$\square$$

Following Lemma F.13, we have that

$$\mathbb{E}_{P_{Y^n|X^n}}\left[\ell\left(x^n, Y^n; x_Q\right) | X^n = x^n\right]$$

$$\leq \frac{1}{n}\lambda_1\left(\mathbf{F}(x_Q)\mathbf{F}_n^\dagger(x^n)\right)S_K(\mathbf{Q}(x^n))$$

$$+ \left\|f^{\mathrm{T}}(x_Q)\mathbf{F}_n^\perp(x^n)\underline{\alpha} - \mathsf{R}(x_Q) - \frac{1}{n}f^{\mathrm{T}}(x_Q)\mathbf{F}_n^\dagger(x^n)f(x^n)\mathsf{R}(x^n)\right\|^2$$

$$\leq \frac{K}{n}\lambda_1\left(\mathbf{F}(x_Q)\mathbf{F}_n^\dagger(x^n)\right)\lambda_1(\mathbf{Q}(x^n))$$

$$+ \left\|f^{\mathrm{T}}(x_Q)\mathbf{F}_n^\perp(x^n)\underline{\alpha} - \mathsf{R}(x_Q) - \frac{1}{n}f^{\mathrm{T}}(x_Q)\mathbf{F}_n^\dagger(x^n)f(x^n)\mathsf{R}(x^n)\right\|^2,$$

where $S_K(\mathbf{Q}(x^n))$ is the sum of top K eigenvalues.

Taking expectation over $P_{X_Q}$, we have the result as

$$\mathbb{E}_{P_{X_Q} P_{Y^n|X^n}} \left[ \ell\left(x^n, Y^n; x_Q\right) | X^n = x^n \right]$$

$$\leq \frac{K}{n} \lambda_1 \left( \mathbf{F}_Q \mathbf{F}_n^\dagger(x^n) \right) \lambda_1(\mathbf{Q}(x^n))$$

$$+ \frac{1}{n} \lambda_1 \left( \mathbf{F}_Q \mathbf{F}_n^\dagger(x^n) \right) \cdot \| \mathsf{R}(x^n) \|^2 + \sum_{x_Q, y} P_{X_Q}(x_Q) R^2(x_Q, y)$$

$$+ \underline{\alpha}^{\mathrm{T}} \mathbf{F}_n^\perp(x^n)^{\mathrm{T}} \mathbf{F}_Q \mathbf{F}_n^\perp(x^n) \underline{\alpha} - \frac{2}{n} \mathsf{R}^{\mathrm{T}}(x^n) f^{\mathrm{T}}(x^n) \mathbf{F}_n^\dagger(x^n) \mathbf{F}_Q \mathbf{F}_n^\perp(x^n) \underline{\alpha},$$

where $\mathbf{F}_Q = \mathbb{E}_{P_{X_Q}}[\mathbb{F}(x_Q)]$. This is because $\sum_{x_Q} P_X(x_Q) f^{\mathrm{T}}(x_Q) \mathsf{R}(x_Q) = 0$.

### F.5 PROOF OF THEOREM 4.5

To prove the Theorem 4.5, we aim to find the minium value of $\mathbb{E}_{P_{Y^n|X^n}}[P_{Y|X}(\hat{y}_{\max}|x_Q)|X^n = x^n]$, which is equivalent to solve the following problem:

$$\min_{\hat{P}_{Y|X}(\cdot|x_Q)} \sum_{y^n} P_{Y^n|X^n}(y^n|x^n) P_{Y|X}(\hat{y}_{\max}|x_Q)$$

$$\text{s.t.} \sum_{y^n} P_{Y^n|X^n}(y^n|x^n) \ell(x^n, y^n; x_Q) \leq \gamma,$$

$$\sum_{y^n} P_{Y^n|X^n}(y^n|x^n) = 1,$$

$$\sum_{y} (\hat{P}_{Y|X}(y|x_Q; y^n) - P_{Y|X}(y|x_Q))^2 = \ell(x^n, y^n; x_Q)$$

$$\hat{y}_{\max} = \arg\max_{y} \hat{P}_{Y|X}(y|x_Q; y^n)$$

**Lemma F.18.** *Given demonstrations $x^n, y^n$, and query $x_Q$, if the mean-squared risk $\ell(x^n, y^n; x_Q)$ satisfies $\ell(x^n, y^n; x_Q) < \frac{1}{2}(P_1 - P_{j+1})^2$, for some $j \geq 1$, then*

$$P_{Y|X}(\hat{y}_{\max}|x_Q) \geq P_j. \tag{12}$$

*Proof.* We prove the contrapositive. Fix an index $s \geq j + 1$. We show that if the predicted label equals $s$ (or more weakly, if $\hat{P}_s \geq \hat{P}_1$), then necessarily

$$\ell(x^n, y^n; x_Q) \geq \frac{1}{2}(P_1 - P_s)^2 \geq \frac{1}{2}(P_1 - P_{j+1})^2,$$

which contradicts the hypothesis. Since this holds for every $s \geq j+1$, no such $s$ can be the predicted label, and hence the predicted label must lie in $\{1, \ldots, j\}$. Because for any $r \leq j$ we have $P_r \geq P_j$, the conclusion follows. Thus fix $s \geq j + 1$ and suppose $\hat{P}_s \geq \hat{P}_1$. Consider the optimization

$$\min_{\hat{P}} \sum_{t=1}^{M} (\hat{P}_t - P_t)^2 \quad \text{s.t.} \quad \hat{P}_s - \hat{P}_1 \geq 0,$$

where $\hat{P}$ ranges over $\mathbb{R}^M$ (the feasible set for probability vectors only increases the minimal cost, so this relaxation provides a valid lower bound). To obtain a lower bound it suffices to restrict attention to coordinates 1 and $s$ and leave all other coordinates equal to their true values $P_t$. With this restriction the problem reduces to

$$\min_{u, v \in \mathbb{R}} (u - P_1)^2 + (v - P_s)^2 \quad \text{s.t.} \quad v - u \geq 0.$$

The minimal value of this two-variable problem under the constraint $v - u \geq 0$ is attained when $v = u$ (pushing toward equality is best), hence we set $u = v = \eta$ and minimize

$$(\eta - P_1)^2 + (\eta - P_s)^2.$$

This quadratic in $\eta$ is minimized at $\eta = \frac{P_1 + P_s}{2}$, giving the minimum value

$$\left(\frac{P_1 + P_s}{2} - P_1\right)^2 + \left(\frac{P_1 + P_s}{2} - P_s\right)^2 = 2\left(\frac{P_1 - P_s}{2}\right)^2 = \tfrac{1}{2}(P_1 - P_s)^2.$$

Therefore any $\hat{P}$ with $\hat{P}_s \geq \hat{P}_1$ must satisfy

$$\sum_{t=1}^{M}(\hat{P}_t - P_t)^2 \geq \tfrac{1}{2}(P_1 - P_s)^2.$$

Because $P_s \leq P_{j+1}$ for all $s \geq j+1$, we have $\tfrac{1}{2}(P_1 - P_s)^2 \geq \tfrac{1}{2}(P_1 - P_{j+1})^2$. Hence if

$$\ell(x^n, y^n; x_Q) < \tfrac{1}{2}(P_1 - P_{j+1})^2,$$

no index $s \geq j+1$ can satisfy $\hat{P}_s \geq \hat{P}_1$, i.e. the maximizer $\hat{y}_{\max}$ must belong to $\{1, \ldots, j\}$. This implies

$$P_{Y|X}(\hat{y}_{\max} \mid x_Q) \geq P_j.$$

$\square$

Lemma F.18 provides the theoretical guarantee of the CB-ICL label predictor with respect to different threshold values of the mean-squared risk. Notice that $P_{Y|X}(\hat{y}_{\max}|x_Q) \leq \max_y P_{Y|X}(y|x_Q) = P_1$, and the equality is achieved when $\ell(x^n, y^n; x_Q) < \tfrac{1}{2}(P_1 - P_2)^2$. Therefore, the CB-ICL label predictor is reduced to the Maximum a Posteriori (MAP) decision when the mean-squared risk is small. Moreover, the following Theorem establishes the connection between the excessive risk and the label predicting error probability based on Lemma F.18.

From Lemma F.18, we know that when $\tfrac{1}{2}(P_1 - P_j)^2 \leq \ell(x^n, y^n; x_Q) < \tfrac{1}{2}(P_1 - P_{j+1})^2$, the minimum value $P_{Y|X}(\hat{y}_{\max}|x_Q)$ can take is $P_j$. Hence, the original problem can turn into a combination problem that $\ell$ only takes value in $\{0, \tfrac{1}{2}(P_1 - P_2)^2, \ldots, \tfrac{1}{2}(P_1 - P_M)^2\}$, with the following $P_{Y|X}(\hat{y}_{\max}|x_Q)$ being $P_1, P_2, \ldots, P_M$. Denote $\ell_j$ as the discrete variable taking value in $\{0, \tfrac{1}{2}(P_1 - P_2)^2, \ldots, \tfrac{1}{2}(P_1 - P_M)^2\}$ with corresponding $P_j$ being the discrete variable taking value in $\{P_1, P_2, \ldots, P_M\}$. The original problem becomes that assign each $y^n$ to one $j \in \{1, \ldots, M\}$ such that $\sum_j \sum_{y^n} P_{Y^n|X^n}(y^n|x^n)1_{\sigma(y^n)=j}P_j$ achieves minimum (with $\sigma$ denoting the assign function). Denote the weight assigned to $j$th index as $w_j = \sum_{y^n} P_{Y^n|X^n}(y^n|x^n)1_{\sigma(y^n)=j}$, and we have the original problem being

$$\min_w \sum_{j=1}^{M} w_j P_j \tag{P1}$$

$$\text{s.t.} \sum_{j=1}^{M} w_j \ell_j \leq \gamma,$$

$$w_j = \sum_{y^n} P_{Y^n|X^n}(y^n|x^n)1_{\sigma(y^n)=j}.$$

To solve this problem, we consider an approximation to this problem as

$$\min_w \sum_{j=1}^{M} w_j P_j \tag{P2}$$

$$\text{s.t.} \sum_{j=1}^{M} w_j \ell_j \leq \gamma,$$

$$\sum_{j=1}^{M} w_j = 1.$$

**Lemma F.19.** *Denote the solution to problem (P1) as $\{w_j^{*,1}\}_{j=1}^M$ and solution to problem (P2) as $\{w_j^{*,2}\}_{j=1}^M$. Then, $\sum_{j=1}^{M} w_j^{*,1}P_j \geq \sum_{j=1}^{M} w_j^{*,2}P_j$.*

*Proof.* Let

$$\mathcal{S}_1 = \left\{ w \in \mathbb{R}^M : \exists\, \sigma : [M]^n \to [M] \text{ such that } w_j = \sum_{y^n} P_{Y^n|X^n}(y^n|x^n)\mathbf{1}_{\{\sigma(y^n)=j\}} \right\}$$

be the feasible set of problem (P1), and

$$\mathcal{S}_2 = \left\{ w \in \mathbb{R}_+^M : \sum_{j=1}^M w_j = 1, \ \sum_{j=1}^M w_j \ell_j \le \gamma \right\}$$

be the feasible set of problem (P2). We first show $\mathcal{S}_1 \subseteq \mathcal{S}_2$.

Take any $w \in \mathcal{S}_1$. By definition there exists an assignment $\sigma$ such that

$$w_j = \sum_{y^n} P_{Y^n|X^n}(y^n|x^n)\mathbf{1}_{\{\sigma(y^n)=j\}} \quad (j = 1, \ldots, M).$$

Clearly $w_j \ge 0$ for all $j$ and

$$\sum_{j=1}^M w_j = \sum_{j=1}^M \sum_{y^n} P_{Y^n|X^n}(y^n|x^n)\mathbf{1}_{\{\sigma(y^n)=j\}} = \sum_{y^n} P_{Y^n|X^n}(y^n|x^n) = 1.$$

Moreover

$$\sum_{j=1}^M w_j \ell_j = \sum_{j=1}^M \sum_{y^n} P_{Y^n|X^n}(y^n|x^n)\mathbf{1}_{\{\sigma(y^n)=j\}}\ell_j = \mathbb{E}_{P_{Y^n|X^n}}\left[\ell(x^n, Y^n; x_Q)\right] \le \gamma,$$

where the last inequality is exactly the feasibility condition in (P1). Hence $w \in \mathcal{S}_2$, proving $\mathcal{S}_1 \subseteq \mathcal{S}_2$.

Now let $f(w) = \sum_{j=1}^M w_j P_j$ be the objective. Since $\mathcal{S}_1 \subseteq \mathcal{S}_2$, the minimum of $f$ over the smaller set $\mathcal{S}_1$ cannot be smaller than the minimum over the larger set $\mathcal{S}_2$. Formally,

$$\min_{w \in \mathcal{S}_1} f(w) \ \ge \ \min_{w \in \mathcal{S}_2} f(w).$$

Noting that the left-hand side equals $\sum_j w_j^{*,1} P_j$ and the right-hand side equals $\sum_j w_j^{*,2} P_j$, the claimed inequality follows. $\square$

By Lemma F.19, we transfer solving of problem (P1) to problem (P2).

**Lemma F.20.** *The solution to problem (P2), denoted as $\{w_j^{*,2}\}_{j=1}^M$, has at most two nonzero components. Furthermore, if there exists two nonzero components, the two components are adjacency to each other.*

*Proof.* Using Lagrange, we have the dual problem as

$$\min_w \sum_j w_j P_j + \mu\left(\sum_j w_j - 1\right) + \lambda\left(\sum_j w_j \ell_j - \gamma\right).$$

The Karush-Kuhn-Tucker (KKT) condition (Bertsekas, 1997) gives that

$$P_j + \mu + \lambda \ell_j \begin{cases} = 0, \ w_j > 0 \\ > 0, \ w_j = 0 \end{cases}$$

Therefore for all $j$, we have that $\hat{P}_j + \mu + \lambda \ell_j \ge 0$. In other words, the support of the optimal distribution $P$ is contained in the set of indices that minimize the affine functional

$$P \mapsto P + \lambda \ell.$$

Consequently, the optimal solution must assign positive probability only to those outcomes lying on the lower envelope of the family of affine functions parameterized by $\lambda$.

Define the discrete slopes

$$s_t := \frac{P_t - P_{t+1}}{\ell_t - \ell_{t+1}}, \qquad t = 1, \dots, M - 1.$$

Compute explicitly using $\ell_t = \frac{1}{2}(P_1 - P_t)^2$:

$$s_t = -\frac{2}{2P_1 - P_t - P_{t+1}}.$$

Because $P_t$ is nonincreasing and $t \mapsto (2P_1 - P_t - P_{t+1})$ is nondecreasing, we have $s_1 < s_2 < \cdots < s_{M-1}$ (strict inequality unless ties occur in the $P_t$'s; ties can be handled by tie-breaking but do not affect the argument).

Now suppose there exist positive $w_i > 0$ and $w_k > 0$ with $k \geq i + 2$ (not adjacent). Since $P_t + \lambda \ell_t$ is affine in the pair $(P_t, \ell_t)$ and the intersection equality above holds for $t = i$ and $t = k$, by intermediate value there must exist an index $r$ with $i < r < k$ such that $P_r + \lambda \ell_r$ is *strictly smaller* than the common value (because the sequence of slopes $s_t$ is strictly increasing, the line through $(\ell_i, P_i)$ and $(\ell_k, P_k)$ lies strictly above at some intermediate lattice point). But then $r$ would yield a strictly smaller $P_r + \lambda \ell_r$, contradicting the KKT condition that all positive-weight indices minimize $P_t + \lambda \ell_t$. Therefore no two positive indices can be non-adjacent; positive indices must be adjacent. $\qquad \square$

**Lemma F.21.** *If mass is placed only at $\ell_j$ and $\ell_{j+1}$ with weights $1 - \alpha$ and $\alpha$ and the mean loss equals $\ell_j + \gamma$ (with $0 \leq \gamma < \ell_{j+1} - \ell_j$), then*

$$\alpha = \frac{\gamma}{\ell_{j+1} - \ell_j}.$$

*Proof.* Immediate from $(1 - \alpha)\ell_j + \alpha \ell_{j+1} = \ell_j + \gamma$. $\qquad \square$

By Lemma F.20 any extreme minimizer has at most two adjacent nonzero elements. Therefore the minimizer can be taken with support $\{\ell_j, \ell_{j+1}\}$. Let the mass at $\ell_{j+1}$ be $\alpha$; by Lemma F.21 we have $\alpha = \gamma/(\ell_{j+1} - \ell_j)$. Consequently

$$\mathbb{E}\big[P_{Y|X}(\hat{y}_{\max} \mid x_Q)\big] \geq (1 - \alpha)P_j + \alpha P_{j+1} = P_j - \alpha(P_j - P_{j+1}).$$

For each $j$,

$$\ell_{j+1} - \ell_j = \tfrac{1}{2}\big[(P_1 - P_{j+1})^2 - (P_1 - P_j)^2\big] = (P_j - P_{j+1})\Big(P_1 - \tfrac{P_j + P_{j+1}}{2}\Big),$$

hence

$$\frac{P_j - P_{j+1}}{\ell_{j+1} - \ell_j} = \frac{2}{2P_1 - P_j - P_{j+1}}.$$

Therefore,

$$\alpha(P_j - P_{j+1}) = \gamma \cdot \frac{P_j - P_{j+1}}{\ell_{j+1} - \ell_j} = \gamma \cdot \frac{2}{2P_1 - P_j - P_{j+1}}.$$

Thus

$$\mathbb{E}\big[P_{Y|X}(\hat{y}_{\max} \mid x_Q)\big] \geq P_j - \frac{2\gamma}{2P_1 - P_j - P_{j+1}},$$

completing the proof.

# G   PROOFS OF PROPERTIES

## G.1   LOWER BOUND OF $\lambda_1(\mathbf{F}(x_Q)\mathbf{F}_n^{-1}(x^n))$

**Theorem G.1.** *Let*

$$f(x, y) \in \mathbb{R}^K, \qquad f(x) = [\, f(x, 1), \dots, f(x, M)\,] \in \mathbb{R}^{K \times M},$$

*and define*

$$\mathbf{F}(x_Q) = f(x_Q)f(x_Q)^{\mathrm{T}}, \qquad \mathbf{F}_n(x^n) = \frac{1}{n}f(x^n)f(x^n)^{\mathrm{T}} = \frac{1}{n}\sum_{i=1}^{n}\sum_{y=1}^{M} f(x_i, y)f(x_i, y)^{\mathrm{T}}.$$

*Assume $\mathbf{F}_n(x^n)$ is positive definite. If the last coordinate satisfies $f_K(x, y) = 1/\sqrt{M}$ for every $x, y$, then*

$$\lambda_1\big(\mathbf{F}(x_Q)\mathbf{F}_n^{-1}(x^n)\big) \geq 1,$$

*where $\lambda_1(\cdot)$ denotes the largest eigenvalue.*

*Proof.* Recall the characterization of the largest generalized eigenvalue for symmetric matrices:

$$\lambda_1\big(\mathbf{F}(x_Q)\,\mathbf{F}_n^{-1}(x^n)\big) = \sup_{w \in \mathbb{R}^K \setminus \{0\}} \frac{w^{\mathrm{T}}\mathbf{F}(x_Q)w}{w^{\mathrm{T}}\mathbf{F}_n(x^n)w}.$$

Let $e_K \in \mathbb{R}^K$ be the unit vector with 1 in the $K$-th coordinate and zeros elsewhere. Using $f_K(x, y) = 1/\sqrt{M}$ for every $x, y$, we compute

$$e_K^{\mathrm{T}}\mathbf{F}(x_Q)e_K = \sum_{y=1}^{M} \big(f_K(x_Q, y)\big)^2 = \sum_{y=1}^{M} \frac{1}{M} = 1,$$

and

$$e_K^{\mathrm{T}}\mathbf{F}_n(x^n)e_K = \frac{1}{n}\sum_{i=1}^{n}\sum_{y=1}^{M} \big(f_K(x_i, y)\big)^2 = \frac{1}{n}\sum_{i=1}^{n}\sum_{y=1}^{M} \frac{1}{M} = 1.$$

Hence the Rayleigh quotient at $e_K$ equals 1:

$$\frac{e_K^{\mathrm{T}}\mathbf{F}(x_Q)e_K}{e_K^{\mathrm{T}}\mathbf{F}_n(x^n)e_K} = 1.$$

Since the supremum over all nonzero $w$ is at least the value at $e_K$, we obtain

$$\lambda_1\big(\mathbf{F}(x_Q)\mathbf{F}_n^{-1}(x^n)\big) \geq 1,$$

as required. $\square$

**Proposition G.2** (When equality holds). *With the notation and assumptions of Theorem G.1, we have*

$$\lambda_1\big(\mathbf{F}(x_Q)\mathbf{F}_n^{-1}(x^n)\big) = 1$$

*if and only if*

$$\mathbf{F}(x_Q) \preceq \mathbf{F}_n(x^n),$$

*i.e. $\mathbf{F}_n(x^n) - \mathbf{F}(x_Q)$ is positive semidefinite. In particular, a simple sufficient condition for equality is*

$$\mathbf{F}_n(x^n) = \mathbf{F}(x_Q),$$

*which occurs for example when $n = 1$ and $x_1 = x_Q$, or more generally when every sample equals $x_Q$ (i.e. $x_1 = \cdots = x_n = x_Q$).*

*Proof.* By the Rayleigh characterization,

$$\lambda_1\big(\mathbf{F}(x_Q)\mathbf{F}_n^{-1}(x^n)\big) = \sup_{w \neq 0} \frac{w^{\mathrm{T}}\mathbf{F}(x_Q)w}{w^{\mathrm{T}}\mathbf{F}_n(x^n)w}.$$

Equality $\lambda_1 = 1$ holds iff for every nonzero $w$,

$$\frac{w^{\mathrm{T}}\mathbf{F}(x_Q)w}{w^{\mathrm{T}}\mathbf{F}_n(x^n)w} \leq 1, \quad \text{i.e.} \quad w^{\mathrm{T}}\mathbf{F}(x_Q)w \leq w^{\mathrm{T}}\mathbf{F}_n(x^n)w.$$

The last inequality for all $w$ is exactly the PSD ordering $\mathbf{F}(x_Q) \preceq \mathbf{F}_n(x^n)$. Hence equality is equivalent to $\mathbf{F}(x_Q) \preceq \mathbf{F}_(x^n)n$.

If $\mathbf{F}_n(x^n) = \mathbf{F}(x_Q)$ then trivially $\mathbf{F}(x_Q) \preceq \mathbf{F}_n(x^n)$ and thus $\lambda_1 = 1$. The condition $\mathbf{F}_n(x^n) = \mathbf{F}(x_Q)$ holds when $f(x_i, y) = f(x_Q, y)$ for every $i, y$, i.e. when $x_i = x_Q$ for all $i$; in particular it holds when $n = 1$ and $x_1 = x_Q$. This proves the stated sufficient condition. $\square$

