# OpenReview forum: "On Theoretical Interpretations of Concept-Based In-Context Learning"
_ICLR.cc/2026/Conference — Submitted to ICLR 2026_

### Official Review · Reviewer_aBfV · 2025-10-31

**Soundness:** 2
**Presentation:** 3
**Contribution:** 2
**Rating:** 4
**Confidence:** 1

**Summary:**

This paper presents a rigorous theoretical analysis of in-context learning, developing mathematical frameworks to understand the excess risk and convergence properties of ICL. The authors establish several key theorems (4.3-4.5) that characterize the performance bounds of ICL under different conditions. The work analyzes both sufficient demonstrations (when F_n(x_n) is invertible) and insufficient demonstrations cases, as well as complete versus incomplete model scenarios. The theoretical framework employs matrix analysis, eigenvalue decompositions, and optimization techniques to derive tight bounds on the excess risk.

**Strengths:**

1. **Rigorous Mathematical Framework**: The paper provides comprehensive mathematical proofs with detailed lemmas and corollaries, establishing a solid theoretical foundation for understanding ICL behavior.

2. **Comprehensive Coverage**: The analysis covers multiple important cases - sufficient vs. insufficient demonstrations, complete vs. incomplete models, providing a nuanced understanding of different ICL scenarios.

3. **Novel Theoretical Insights**: The connection between eigenvalue properties (particularly λ_1(F(x_Q)F_n^(-1)(x_n))) and ICL performance provides new theoretical insights into when ICL succeeds or fails.

4. **Practical Relevance**: The bounds derived (e.g., Theorem 4.5 on label prediction error) have direct implications for understanding ICL's practical performance limits.

5. **Technical Depth**: The use of advanced techniques like KKT conditions, matrix perturbation theory, and generalized eigenvalue analysis demonstrates technical sophistication.

**Weaknesses:**

1. **Limited Empirical Validation**: From the provided content, there appears to be heavy emphasis on theory without corresponding experimental validation of the theoretical bounds.

2. **Restrictive Assumptions**: The analysis relies on specific assumptions (e.g., f_K(x,y) = 1/√M) that may not hold in practical ICL scenarios, potentially limiting the applicability.

3. **Accessibility**: The highly technical nature and dense mathematical notation may limit the paper's accessibility to practitioners and researchers outside theoretical ML.

4. **Gap to Practice**: The theoretical model may oversimplify real ICL mechanisms in large language models, where the actual learning dynamics are likely more complex.

**Questions:**

1. **Empirical Validation**: How well do the theoretical bounds match empirical ICL performance in actual language models? What experiments were conducted to validate these bounds?

2. **Assumption Justification**: What is the justification for the specific form f_K(x,y) = 1/√M? How sensitive are the results to violations of this assumption?

3. **Practical Implications**: Can the authors provide concrete guidelines for practitioners on how to use these theoretical insights to improve ICL performance?

4. **Comparison with Existing Work**: How do these bounds compare with existing theoretical analyses of ICL? What specific gaps in prior work does this address?

5. **Extension to Modern Models**: How do these results extend to transformer architectures and attention mechanisms that are central to modern ICL?

6. **Tightness of Bounds**: Are the derived bounds tight? Can you provide examples where the bounds are achieved?

7. **Computational Considerations**: What are the computational implications of computing these bounds in practice, especially for large-scale models?

---

> ### Author Response · Authors · 2025-11-23
> **Response Part 1 / 2**
>
> # Weakness
>
> ## Weakness 1
>
> We thank the reviewer for inquiring about the empirical validation of our theoretical bounds. we have to point out that, for real data, the distribution $P_{Y|X}(y|x)$ is unknown, which means the the true bound cannot be evaluated exactly. Hence, to assess how well the theory predicts real ICL behavior, we check the relation between the residual interaction term $\mathbf{R}^2$ and performance on LLaMA-3-8B, Qwen3-8B, and DeepSeek-R1-Distill-8B across four benchmark suites (MMLU, MMLU-Pro, GPQA, and GPQA-Diamond), please see Figure 3 for detail. A key prediction of our bound is that the residual interaction term $\mathbf{R}^2$ should strongly correlate with ICL failure modes and Figure 3 demonstrates the relation between $\mathbf{R}^2$ and the accuracy performance.
>
> ## Weakness 2
>
> We thank you for raising this problem. We apply normalization on the embedding to force the assumption always holding. The normalization procedure is implemented as follows:
>
> ```bash
> **Require:** Embedding $f(x,y) \in \mathbb{R}^K$
> **Ensure:** Normalized embedding $f(x,y) \in \mathbb{R}^{K+1}$
>
> 1. **$\triangleright$ Embedding normalization (as specified in Section 3)**
> 2. **for** $k = 1$ **to** $K$ **do**
> 3.   $\quad \text{// Zero-mean and unit variance constraints}$
> 4.   $\quad \mu_k \gets \frac{1}{M}\sum_{y=1}^M f_Q^y[k]$
> 5.   $\quad \sigma_k \gets \sqrt{\frac{1}{M}\sum_{y=1}^M (f_Q^y[k] - \mu_k)^2}$
> 6.   $\quad f_Q^y[k] \gets \frac{f_Q^y[k] - \mu_k}{\sigma_k}$ for all $y$
> 7. **end for**
> 8. **$\triangleright$ Add bias dimension (padded as in paper)**
> 9. $\quad f_Q^y[K+1] \gets \frac{1}{\sqrt{M}}$ for all $y$
> ```
>
> We will add this as a part of the Algorithm 1 table in the paper.
>
> ## Weakness 3
>
> We appreciate the reviewer’s comment regarding accessibility. Our work indeed develops a mathematically rigorous framework, and this level of formalism unavoidably introduces dense notation. We will take several concrete steps to increase accessibility without compromising theoretical completeness.
>
> 1. We have included a roadmap in the Introduction section, summarizing the core ideas before entering formal notation.
> 2. We will add a notation table in the appendix. This will enhance readability and reduce potential misinterpretations.
> 3. We will add a step-by-step algorithm box describing the practical workflow of our method (as requested in other comments). This will significantly lower the barrier for non-theoretical readers.
>
> ## Weakness 4
>
> We thank the reviewer for pointing out that theoretical models inevitably simplify real LLM behavior. We agree that the actual internal dynamics of large language models are more complex than any tractable analytical model. However, we would like to emphasize that our goal is not to replicate full LLM dynamics, but to isolate the dominant mechanism underlying ICL. Beside that, the proposed theory yields concrete predictions that match actual LLM behavior. For example, it explains why ICL improves with a few demonstrations, how pre-training model help improve ICL performance and how to metric the quality of demonstrations.
>
> # Question
>
> ## Question 1
>
> Thank you for the problem. We have conducted several real LLMs experiments on the residual term to check the correctness of proposed bound. Please see the response to weakness 1.
>
> ## Question 2
>
> Thank you for the problem. The assumption will be forced to satisfied in the normalization phase, where the embeddings of all possible labels will be collected and normalized. Please see the response to weakness 2.
>
> ## Question 3
>
> We appreciate the reviewer’s interest in practical guidance derived from our theory. Our paper already provides two concrete, practitioner-oriented recommendations, each supported by extensive experiments:
>
> 1. **Demonstration Selection via Similarity Score.** We introduce a computable similarity metric derived from the dominant eigenvalue of $\lambda_1(\mathbf{F}(x_Q)\mathbf{F}_n^\dagger{x^n})$, which can be evaluated directly from hidden states. Table 2 and Figure 2 show that this metric consistently improves ICL performance by enabling practitioners to select demonstrations that align with the query in the concept space. This yields immediate, actionable guidance for real-world prompt construction.
> 2. **Capability Diagnosis via the Residual Term.** Our theory identifies the residual interaction term R2R2 as the main predictor of ICL failure. Figure 3 empirically confirms a strong relationship between $\mathbf{R}^2$ and accuracy across multiple benchmarks. This provides a practical diagnostic tool: practitioners can estimate $\mathbf{R}^2$ to determine when ICL is likely to succeed or when alternative strategies (e.g., finetuning) are required.
>
> Together, these two components constitute direct, operational guidelines for improving ICL performance and understanding model limitations.

---

> ### Author Response · Authors · 2025-11-23
> **Response 2 / 2**
>
> # Question
>
> ## Question 4
>
> We thank the reviewer for raising the question. To out knowledge, existing theory work has primarily focused on highly simplified settings, such as Boolean functions, linear regression with idealized feature structures, or sub-Gaussian representations. These works provide valuable insights but offers limited guidance to realistic large-scale language models. Please see Section 2 for a detailed comparison with prior work.
>
> In contrast, our framework produces explicitly computable quantities that can be evaluated directly from real LLM hidden states, enabling both theoretical interpretation and practical diagnostics.
>
> ## Question 5
>
> We thank the reviewer for the questions. We have conducted experiments on the modern LLMs, including Qwen3, LLaMA3 and Deepseek family. Please see Table 1, Table 2, Figure 2 and Figure 3 for experiment results.
>
> Beside that, we add experiments of other frameworks, including encoder-decoder and decoder-only frameworks, including:
>
> | Model Structure | Selected Model | Hidden state |
> | --- | --- | --- |
> | Encoder-decoder | T5, BART | Output of last layer of encoder with mean pooling |
> | Encoder-only | BERT, MPNet, MiniLM | Last layer output with mean pooling |
>
> The results are shown below:
>
> | Type | Encoder-Decoder |  |  |  |  |  | Encoder-Only |  |  |  |  |  |
> | --- | --- | --- | --- | --- | --- | --- | --- | --- | --- | --- | --- | --- |
> | Model | FLAN-T5-XXL |  | FLAN-T5-XL |  | BART |  | MiniLM |  | BERT |  | MPNET |  |
> |  | ICL (%) | CB-ICL (%) | ICL (%) | CB-ICL (%) | ICL (%) | CB-ICL (%) | ICL (%) | CB-ICL (%) | ICL (%) | CB-ICL (%) | ICL (%) | CB-ICL (%) |
> | MMLU | 55.1 | 54.67 | 52.4 | 50.53 | 22.9 | 26.12 | Not applied | 25.75 | 23.02 | 25.65 | Not applied | 25.85 |
> | MMLU-Pro | 16.54 | 15.43 | 15.37 | 15.90 | 11.24 | 10.36 | Not applied | 10.10 | 10.59 | 9.43 | Not applied | 9.72 |
> | GPQA | 26.85 | 28.00 | 23.52 | 27.80 | 27.27 | 26.93 | Not applied | 29.08 | 25.44 | 27.31 | Not applied | 28.63 |
> | GPQA-Diamond | 22.46 | 30.10 | 24.08 | 29.03 | 21.50 | 33.24 | Not applied | 25.98 | 29.19 | 29.03 | Not applied | 29.18 |
>
> Results confirm that CB-ICL is effective across these paradigms: on most datasets (e.g., GPQA-Diamond, GPQA), CB-ICL consistently outperforms vanilla ICL. For instance, FLAN-T5-XXL achieves 30.10% in the setting of CB-ICL, compared with 22.46% under ICL on GPQA-Diamond, and BART reaches 33.24% (CB-ICL) compared with 21.50% (ICL) on the same dataset. Even for encoder-only models where vanilla ICL is not applicable, CB-ICL still yields meaningful performance (e.g., 29.18% on GPQA-Diamond for MPNET), demonstrating CB-ICL’s broad applicability beyond decoder-only LLMs. What’s more, by comparison, decoder-only models had higher ICL baselines (e.g., Qwen3-32B reached 76.46% on MMLU) and required no additional adjustments to work with CB-ICL.
>
> ## Question 6
>
> We thank the reviewer for raising this problem. The derived bounds are tight in the worst-case sense, as there exist distributions and feature maps for which our inequalities become equalities, so the constants in the bounds cannot be uniformly improved. The example for the equality get achieved is as follow: set that $M=2$, for all $x$ $P_{Y|X}(y|x)$ is unformly distributed, ant $f(x,y)$ satisfies that the top K eigenvectors of $f(x^n)\mathbf{Q}(x^n)f^\text{T}(x^n)\mathbf{F}_n^\dagger(x^n)$ lie predominantly within the principal subspace of $\mathbf{F}(x_Q)\mathbf{F}_n^\dagger(x^n)$. In this configuration, each intermediate matrix inequality used in our proof (including the spectral and trace bounds) is attained with equality, and therefore the final bound.
>
> ## Question 7
>
> We thank the reviewer for pointing out the lack of explicit computational-cost reporting. To address this, we performed a detailed runtime and memory profiling study covering all major steps of our pipeline: golden-example construction, prediction probability extraction, and GPU memory usage during both stages.
>
> We profiled the full MMLU dataset (14,042 examples) using LLaMA-3-8B on a single A800-80GB GPU. All measurements report wall-clock time and peak CUDA memory (exclude the model), averaged across all test inputs.
>
> | Process |  | Golden Demonstration Selection | Prediction Probability Extraction |
> | --- | --- | --- | --- |
> | Time | Average | 17.38 ms | 3.32 ms |
> |  | Min | 4.24 ms | 2.16 ms |
> |  | Max | 257.91 ms | 29.71 ms |
> | Memory Usage | Average | 161.97 MB | 73.51 MB |
> |  | Min | 41.73 MB | 73.47 MB |
> |  | Max | 507.84 MB | 73.55 MB |
>
> Since most of the computation depends only on the few-shot demonstration set, computing all relevant norms, eigenvalues, and pseudoinverses takes averagely 17.38ms.

---

### Official Review · Reviewer_Qc59 · 2025-11-01

**Soundness:** 3
**Presentation:** 1
**Contribution:** 3
**Rating:** 4
**Confidence:** 4

**Summary:**

This paper investigates the theoretical foundations of concept-based in-context learning (CB-ICL). It develops a formal analysis that explains why and when CB-ICL performs effectively with limited demonstrations. The proposed theory models the relationship between prompt representations and LLM embeddings, introduces a similarity measure for prompt and query selection, and characterizes factors such as embedding dimensionality and demonstration sufficiency. Experimental results on several benchmarks support the theoretical findings and show that CB-ICL performs comparably or better than standard in-context learning approaches.

**Strengths:**

1. Carefully framed theoretical contribution with explicit assumptions. The paper presents a coherent theoretical framework, states assumptions clearly, and supplies largely complete derivations as well as interpretable quantitative measures (e.g., mean-squared excessive risk and the similarity score) that connect embedding geometry to prediction error.
2. Substantial and reasonably broad empirical validation. Experiments cover multiple benchmarks and model families, and the proposed similarity-based “golden” selection consistently improves performance in several settings, providing practical evidence that the theory has explanatory power.
3. Thoughtful synthesis and targeted novelty over prior ingredients. While the paper builds on familiar analytical tools used in prior ICL work (e.g., linear-model style analyses, projection/basis arguments and residual risk notions), it integrates these elements into a concept-based CB-ICL formulation and derives a practical similarity metric and selection criteria.

**Weaknesses:**

1. Notation and presentation are cluttered and sometimes unclear. Many symbols are visually similar (multiple uses of f with different inputs, closely related subscripts/superscripts) and some variables (e.g., K, M) are not introduced at the first encounter, increasing the reader’s cognitive load and risk of misinterpretation.
2. Method exposition is limited because of emphasis on theory. The manuscript concentrates on mathematical derivations at the expense of a clear, step-by-step description of the practical pipeline due to page constraints. Key implementation and procedural details are only conveyed via a single flow diagram, and the diagram’s symbols seem to be inconsistently used relative to the main text, which impedes reproducibility and practical understanding.
3. Ambiguity about how LLM embeddings are obtained and pooled. The paper treats f(x,y) as the semantic embedding but gives no practical details on how embeddings are extracted from LLM internals (which layer(s), pooling over token sequence, handling of label tokens, etc.), nor empirical comparisons to off-the-shelf sentence encoders (BERT / sentence-BERT / specialized sentence encoders) that might provide alternative embeddings.
4. Limited statistical reporting and stronger baselines. Results are reported as point estimates (accuracy percentages) across models/datasets, but confidence intervals, standard deviations from repeated runs, or formal significance tests are missing; comparisons against stronger baselines or more relative work are limited.
5. Scope limited to classification. The theoretical framework and experiments focus on label prediction, so applicability to structured outputs, chain-of-thought (CoT) style reasoning, or sequence generative tasks is not established; the paper should either clearly state this limitation or outline how the framework could be extended.
6. Theory vs. empirical support for embedding dimension (K). The paper theoretically predicts excessive risk grows with embedding dimension K, but presents only a theoretical argument; there is little direct empirical ablation that systematically manipulates embedding dimension to validate this trade-off.
7. No reported computational cost or runtime analysis. This paper does not provide wall-clock timing, memory, or computational-cost figures for constructing embeddings, computing similarity scores, or golden selection, which would be valuable for assessing practical adoption.
8. Normalization preprocessing steps are specified mathematically but not operationally. The normalization constraints on f(x,y) are stated (zero-mean, unit variance across labels and a padded bias dimension), yet the paper lacks an explicit description of how these normalizations are implemented in experiments.

**Questions:**

1. On harder examples (the “Pro/Diamond” tiers) the reported gains shrink and especially, for MMLU Pro, CB-ICL sometimes underperforms standard ICL. Can the authors explain this behavior and identify which dataset or model characteristics drive the reduced or negative benefit?
2. Could using dedicated sentence/semantic encoders (e.g., all-MiniLM or other off-the-shelf sentence embedding models) yield more accurate semantic vectors and change the method’s empirical performance?
3. The paper frequently refers to f(x,y) as a “semantic embedding.” However, the representation produced by an LLM may capture not only semantic content but also aspects of reasoning, contextual adaptation, or latent pattern discovery. How should this representation be conceptually understood within the proposed framework — is it purely semantic, or does it implicitly encode reasoning dynamics as well?
4. From the methodological description, it seems feasible to construct embeddings without explicitly conditioning on the label — that is, to use f(x) instead of f(x, y) — since the subsequent computations could still be grouped by y. It would be helpful to clarify how this alternative would differ conceptually or empirically from the proposed formulation.
5. The reported correlation between residual risk and task accuracy appears substantially weaker on GPQA than on MMLU. What factors might explain this discrepancy? For example, could the difference arise from the reasoning-heavy nature of GPQA, smaller sample size, or less stable alignment between the embedding similarity and true task difficulty?
6. The theoretical analysis assumes that both the demonstrations and the query are drawn from the same conditional distribution. In practice, however, in-context examples are often selected non-randomly or from a slightly different sub-distribution due to prompt design or dataset bias. How sensitive are the theoretical bounds and conclusions to such distributional mismatches or selection biases?

---

> ### Author Response · Authors · 2025-11-23
> **Response Part 1 / 4**
>
> # Weakness
>
> ## Weakness 1
>
> We acknowledge that the notation in its current form creates cognitive load for readers. We will add a comprehensive notation table in the Appendix that explicitly defines all symbols (including K as embedding dimension, M as number of classes, etc.). This will enhance readability and reduce potential misinterpretations.
>
> | **Symbol** | **Description** | **Section/Context** |
> | --- | --- | --- |
> | **Basic Variables** |  |  |
> | $x$, $y$ | Input text and corresponding label | Section 3 |
> | $x^n = (x_1,\dots,x_n)$ | Collection of $n$ demonstration input texts | Section 3 |
> | $y^n = (y_1,\dots,y_n)$ | Collection of $n$ demonstration labels | Section 3 |
> | $x_Q$ | Query input text | Section 3 |
> | $y_Q$ | Query label to be predicted | Section 3 |
> | $M$ | Number of possible classes/labels | Section 3 |
> | $K$ | Dimension of LLM embedding space | Section 3 |
> | $\theta$ | Parameters of the pre-trained LLM | Section 3 |
> | $P_{X_Q}(x_Q)$ | Distribution of query inputs | Section 3 |
> | $P_{Y\|X}(y\|x)$ | Ground truth conditional distribution | Section 3 |
> | **LLM Embeddings** |  |  |
> | $f(x,y) = [f_1(x,y),\dots,f_K(x,y)]\in\mathbb{R}^K$ | Semantic $K$-dimensional embedding vector generated by pre-trained LLM | Section 3 |
> | $f(x) = [f(x,y=1),\dots,f(x,y=M)] \in \mathbb{R}^{K\times M}$ | Embedding matrix for all labels of input $x$ | Section 4.1 |
> | $F(x) = f(x)f^T(x) \in \mathbb{R}^{K\times K}$ | Outer product matrix for input $x$ | Section 4.1 |
> | $F_n(x^n) = \frac{1}{n}\sum_{i=1}^n\sum_{y=1}^M f(x_i,y)f^T(x_i,y)$ | Average embedding matrix for demonstrations | Section 3 |
> | $\bar{f}_n(x^n,y^n)$ | Average embedding of demonstrations with their labels | Section 3 |
> | **Concept & Prediction** |  |  |
> | $\alpha$ | Ground truth concept vector | Section 3 |
> | $R(x,y)$ | Residual term (knowledge not captured by LLM) | Section 3 |
> | $\hat{\alpha}(x^n,y^n) = F^{\dagger}_n(x^n)\bar{f}_n(x^n,y^n)$ | Estimated prompt concept | Section 3 |
> | $\hat{P}_{Y\|X}(y\|x_Q) = \hat{\alpha}^T(x^n,y^n)f(x_Q,y)$ | Estimated conditional distribution | Section 3 |
> | $\hat{y}^\ast_{\text{max}} $ | Predicted label | Section 4.4 |
> | **Risk & Error Measures** |  |  |
> | $\ell(x^n,y^n;x_Q)$ | Mean-squared risk | Section 3 |
> | $E_{P_{Y^n\|X^n}}[\ell(x^n,Y^n;x_Q)\|X^n=x^n]$ | Excessive risk (conditioned on prompt inputs) | Section 3 |
> | $R^2 = \sum_{x,y} P_X(x)R^2(x,y)$ | Measure of LLM incompleteness | Section 5.3 |
> | **Matrix Operations** |  |  |
> | $A^{\dagger}$ | Pseudo-inverse of matrix $A$ | Section 3 |
> | $F^{\perp}_n(x^n) = I_K - F^{\dagger}_n(x^n)F_n(x^n)$ | Projection onto null space of $F_n(x^n)$ | Section 4.2 |
> | $\lambda_1(\cdot)$ | Largest eigenvalue of a matrix | Throughout |
> | $\phi_i = [P_{Y\|X}(1\|x_i),\dots,P_{Y\|X}(M\|x_i)]^T$ | Probability vector for input $x_i$ | Section 4.1 |
> | $Q(x_i) = \text{diag}\{P_{Y\|X}(1\|x_i),\dots,P_{Y\|X}(M\|x_i)\} - \phi_i\phi_i^T$ | Covariance-like matrix | Section 4.1 |
> | $Q(x^n) = \text{diag}\{Q(x_1),\dots,Q(x_n)\}$ | Block diagonal matrix | Section 4.1 |
> | **Similarity Measure** |  |  |
> | $\lambda_1^{-1}(F(x_Q)F^{\dagger}_n(x^n))$ | Similarity score between query and demonstrations | Section 4.1, 5.2 |
>
> This notation table provides a comprehensive reference for all major symbols used in the paper, organized by conceptual categories. It includes the mathematical definitions, contextual usage, and relevant section references to help readers navigate the theoretical framework of CB-ICL more effectively.

---

> ### Author Response · Authors · 2025-11-23
> **Response Part 2 / 4**
>
> # Weakness
>
> ## Weakness 2
>
> Thank you for raising this problem. We will include a detailed algorithm box (Algorithm 1) in the revised manuscript that explicitly outlines the CB-ICL procedure:
>
> ```bash
> **Algorithm 1:** CB-ICL Detailed Procedure
> **Require:** LLM parameters $\theta$, system prompt $s$, demonstrations $\{(x_i,y_i)\}_{i=1}^n$, query $x_Q$
> **Ensure:** Predicted label $\hat{y}_Q$
>
> 1. **$\triangleright$ Semantic embedding generation**
> 2. **for** $i = 1$ **to** $n$ **do**
> 3.    $\quad \text{// Construct full context with system prompt}$
> 4.    $\quad \text{context}_i \gets [s, x_i, y_i]$
> 5.    $\quad \text{// Forward pass through LLM (decoder-only architecture)}$
> 6.    $\quad h_i \gets \text{LLM}_\theta(\text{context}_i)$
> 7.    $\quad \text{// Extract embedding from last layer, last token}$
> 8.    $\quad f_i \gets h_i^{\text{last layer}}[\text{last token}]$
> 9. **end for**
> 10. **for** $y = 1$ **to** $M$ **do**
> 11.   $\quad \text{query\_context}_y \gets [s, x_Q, y]$
> 12.   $\quad h_Q^y \gets \text{LLM}_\theta(\text{query\_context}_y)$
> 13.   $\quad f_Q^y \gets h_Q^{y,\text{penultimate}}[\text{position before } y]$
> 14. **end for**
>
> 15. **$\triangleright$ Embedding normalization (as specified in Section 3)**
> 16. **for** $k = 1$ **to** $K$ **do**
> 17.   $\quad \text{// Zero-mean and unit variance constraints}$
> 18.   $\quad \mu_k \gets \frac{1}{M}\sum_{y=1}^M f_Q^y[k]$
> 19.   $\quad \sigma_k \gets \sqrt{\frac{1}{M}\sum_{y=1}^M (f_Q^y[k] - \mu_k)^2}$
> 20.   $\quad f_Q^y[k] \gets \frac{f_Q^y[k] - \mu_k}{\sigma_k}$ for all $y$
> 21. **end for**
> 22. **$\triangleright$ Add bias dimension (padded as in paper)**
> 23. $\quad f_Q^y[K+1] \gets \frac{1}{\sqrt{M}}$ for all $y$
>
> 24. **$\triangleright$ Concept estimation**
> 25. $F_n \gets \frac{1}{n}\sum_{i=1}^n \sum_{y=1}^M f(x_i,y)f^T(x_i,y)$
> 26. $\bar{f}_n \gets \frac{1}{n}\sum_{i=1}^n f_i$
> 27. $\hat{\alpha} \gets F_n^\dagger \bar{f}_n$
>
> 28. **$\triangleright$ Prediction**
> 29. **for** $y = 1$ **to** $M$ **do**
> 30.   $\quad \hat{P}(y) \gets \hat{\alpha}^T f_Q^y$
> 31. **end for**
> 32. $\hat{y}_Q \gets \arg\max_y \hat{P}(y)$
> 33. **return** $\hat{y}_Q$
> ```
>
> ## Weakness 3
>
> Thank you for raising this problem. Our paper utilizes decoder-only LLMs with the following embedding extraction procedure:
>
> 1. For each (x,y) pair, we format the text as "[system_prompt] Question: x Options: [options context] Answer y"
> 2. The embedding f(x,y) is obtained from the last layer hidden state of the last token of the full input. Following [1], we find that embeddings from this position best capture task-relevant semantic information.
> 3. The embedding dimension K is fixed to the model's hidden size (e.g., 4096 for LLaMA3-8B)
>
> Please note that the model we use are decoder-only models, which has been the main stream in the field of large language models. For decoder-only models, there are no label tokens design. As for the empirical comparisons with off-the-shelf sentence encoders, we conducted experiments on encoder-only model, encoder-decoder models (e.g., FLAN-T5, BART), and decoder-only models, covering all architectural types mentioned. The experiments followed the same CB-ICL workflow as the main paper. The model we used is listed as following:
>
> | Model Structure | Selected Model | Hidden state |
> | --- | --- | --- |
> | Encoder-decoder | T5, BART | Output of last layer of encoder with mean pooling |
> | Encoder-only | BERT, MPNet, MiniLM | Last layer output with mean pooling |
>
> The results are shown below:
>
> | Type | Encoder-Decoder |  |  |  |  |  | Encoder-Only |  |  |  |  |  |
> | --- | --- | --- | --- | --- | --- | --- | --- | --- | --- | --- | --- | --- |
> | Model | FLAN-T5-XXL |  | FLAN-T5-XL |  | BART |  | MiniLM |  | BERT |  | MPNET |  |
> |  | ICL (%) | CB-ICL (%) | ICL (%) | CB-ICL (%) | ICL (%) | CB-ICL (%) | ICL (%) | CB-ICL (%) | ICL (%) | CB-ICL (%) | ICL (%) | CB-ICL (%) |
> | MMLU | 55.1 | 54.67 | 52.4 | 50.53 | 22.9 | 26.12 | Not applied | 25.75 | 23.02 | 25.65 | Not applied | 25.85 |
> | MMLU-Pro | 16.54 | 15.43 | 15.37 | 15.90 | 11.24 | 10.36 | Not applied | 10.10 | 10.59 | 9.43 | Not applied | 9.72 |
> | GPQA | 26.85 | 28.00 | 23.52 | 27.80 | 27.27 | 26.93 | Not applied | 29.08 | 25.44 | 27.31 | Not applied | 28.63 |
> | GPQA-Diamond | 22.46 | 30.10 | 24.08 | 29.03 | 21.50 | 33.24 | Not applied | 25.98 | 29.19 | 29.03 | Not applied | 29.18 |
>
> Results confirm that CB-ICL is effective across these paradigms: on most datasets (e.g., GPQA-Diamond, GPQA), CB-ICL consistently outperforms vanilla ICL. Even for encoder-only models where vanilla ICL is not applicable, CB-ICL still yields meaningful performance (e.g., 29.18% on GPQA-Diamond for MPNET), demonstrating CB-ICL’s broad applicability beyond decoder-only LLMs.
>
> [1] Nikolaou, Giorgos, et al. "Language Models are Injective and Hence Invertible." arXiv preprint arXiv:2510.15511 (2025).

---

> ### Author Response · Authors · 2025-11-23
> **Response Part 3 / 4**
>
> # Weakness
>
> ## Weakness 4
>
> Thank you for raising this problem. To address the issues of insufficient statistical reporting and limited baseline comparison, we optimized the experimental design and baseline supplementation, as follows:
>
> 1. Repeated Experiments and Improved Statistical Indicators
> We focused on the randomness of the GPQA and GPQA-diamond datasets, while the example data in the MMLU and MMLU-Pro datasets had fixed settings, with only 5 demonstrations per subject, thus eliminating the need for additional randomness considerations. We conducted repeated experiments on six mainstream models (8B/14B/32B parameter counts) to supplement the mean and standard deviation indicators. The results are as follows:
>
> | **Dataset** | **LLaMA3-8B** |  | **Qwen3-8B** |  | **Deepseek-R1-8B** |  |
> | --- | --- | --- | --- | --- | --- | --- |
> |  | Mean (%) | Std | Mean (%) | Std | Mean (%) | Std |
> | GPQA | 39.79 | 2.50 | 61.08 | 2.29 | 50.00 | 1.37 |
> | GPQA-diamond | 34.84 | 2.45 | 63.95 | 3.49 | 49.89 | 2.97 |
>
> | **Dataset** | **Qwen3-14B** |  | **Deepseek-R1-14B** |  | **Qwen3-32B** |  |
> | --- | --- | --- | --- | --- | --- | --- |
> |  | Mean (%) | Std | Mean (%) | Std | Mean (%) | Std |
> | GPQA | 50.47 | 2.43 | 47.25 | 2.95 | 52.65 | 2.73 |
> | GPQA-diamond | 65.20 | 3.56 | 60.31 | 2.72 | 65.43 | 3.59 |
> 1. Baseline: Cosine Similarity Chosen Demonstrations
>
> We compared the performance of CB-ICL (golden) with the cosine similarity baseline across models of different parameter scales and datasets. Results show CB-ICL (golden) performs slightly better than cosine similarity overall: the score gap between the two methods is within ±2% across most datasets and model scales (8B/14B/32B), with CB-ICL outperforming cosine similarity by 0.4%–1.5% in average scores. While cosine similarity marginally outperforms CB-ICL on individual datasets (e.g., MMLU for Qwen3-14B), this advantage is not universal.
>
> | **Dataset** | **LLaMA3-8B** |  | **Qwen3-8B** |  | **Deepseek-R1-8B** |  |
> | --- | --- | --- | --- | --- | --- | --- |
> |  | CB-ICL (%) | Cosine Similarity (%) | CB-ICL (%) | Cosine Similarity (%) | CB-ICL (%) | Cosine Similarity (%) |
> | MMLU | 73.74 | 72.45 | 80.86 | 79.41 | 68.44 | 68.24 |
> | MMLU-Pro | 36.59 | 36.54 | 58.97 | 59.74 | 44.24 | 42.84 |
> | GPQA | 48.51 | 49.10 | 65.77 | 66.67 | 51.67 | 52.56 |
> | GPQA-diamond | 42.28 | 41.35 | 73.83 | 70.56 | 52.47 | 51.78 |
> | Average | 50.28 | 49.86 | 69.86 | 69.10 | 54.21 | 53.86 |
>
> | **Dataset** | **Qwen3-14B** |  | **Deepseek-R1-14B** |  | **Qwen3-32B** |  |
> | --- | --- | --- | --- | --- | --- | --- |
> |  | CB-ICL (%) | Cosine Similarity (%) | CB-ICL (%) | Cosine Similarity (%) | CB-ICL (%) | Cosine Similarity (%) |
> | MMLU | 82.64 | 82.98 | 82.34 | 83.15 | 84.42 | 84.20 |
> | MMLU-Pro | 62.67 | 60.34 | 61.34 | 61.52 | 67.29 | 66.24 |
> | GPQA | 55.83 | 53.16 | 52.32 | 50.50 | 58.73 | 57.35 |
> | GPQA-diamond | 67.32 | 66.39 | 62.91 | 61.32 | 67.43 | 67.64 |
> | Average | 67.12 | 65.72 | 64.73 | 64.12 | 69.47 | 68.86 |
>
> ## Weakness 5
>
> We acknowledge that our framework is currently limited to classification tasks. This represents an important boundary of our contribution. While recent ICL theory has primarily focused on regression problems (please see the Section 2.1 for related literature review), our work is among the first to provide theoretical guarantees specifically for classification settings. We will add the following statement to the limitations section:
>
> "While our framework provides theoretical insights for classification tasks, extending it to structured outputs, chain-of-thought reasoning, or sequence generation remains an important direction for future work. We believe the theoretical foundation established here provides a stepping stone toward understanding these more complex scenarios."
>
> ## Weakness 6
>
> We thank the reviewer for raising this important point. To manipulate the dimension K, we trained an lightweight auto-encoder (with a three-layer MLP as encoder) on the model’s raw hidden states. The encoder–decoder is trained with MSE reconstruction loss until the loss is less than 2.5e-6. During evaluation, we replace the original hidden state with the encoded representation, ensuring that the rest of the model remains unchanged.
>
> We report the 5-shot MMLU results of Qwen3-8B and Qwen3-14B as follow:
>
> | Dimension K | 128 | 256 | 512 | 1024 | 2048 | 4096 | 5120 |
> | --- | --- | --- | --- | --- | --- | --- | --- |
> | Qwen3-8B | 79.52 | 80.14 | 80.96 | 81.16 | 81.92 | 80.86 | 79.68 |
> | Qwen3-14B | 79.27 | 80.52 | 81.32 | 82.45 | 83.13 | 83.32 | 82.64 |
>
> We find that performance improves as the dimension grows from 256 to around 2048, but drops again when *K* becomes excessively large. As for the larger model, this trend still holds, with the dimension of best dimension increases from 2048 to 4096. This matches our theoretical prediction of a non-monotonic risk curve that too small a dimension underfits, whereas too large a dimension amplifies noise and increases excess risk.

---

> ### Author Response · Authors · 2025-11-23
> **Response Part 4 / 4**
>
> # Weakness
>
> ## Weakness 7
>
> We thank the reviewer for pointing out the lack of explicit computational-cost reporting. To address this, we performed a detailed runtime and memory profiling study covering all major steps of our pipeline: golden-example construction, prediction probability extraction, and GPU memory usage during both stages.
>
> We profiled the full MMLU dataset (14,042 examples) using LLaMA-3-8B on a single A800-80GB GPU. All measurements report wall-clock time and peak CUDA memory (exclude the model), averaged across all test inputs.
>
> | Process |  | Golden Demonstration Selection | Prediction Probability Extraction |
> | --- | --- | --- | --- |
> | Time | Average | 17.38 ms | 3.32 ms |
> |  | Min | 4.24 ms | 2.16 ms |
> |  | Max | 257.91 ms | 29.71 ms |
> | Memory Usage | Average | 161.97 MB | 73.51 MB |
> |  | Min | 41.73 MB | 73.47 MB |
> |  | Max | 507.84 MB | 73.55 MB |
>
> ## Weakness 8
>
> Thank you for raising this problem. The normalization procedure is implemented as follows:
>
> ```bash
> **Require:** Embedding $f(x,y) \in \mathbb{R}^K$
> **Ensure:** Normalized embedding $f(x,y) \in \mathbb{R}^{K+1}$
>
> 1. **$\triangleright$ Embedding normalization (as specified in Section 3)**
> 2. **for** $k = 1$ **to** $K$ **do**
> 3.   $\quad \text{// Zero-mean and unit variance constraints}$
> 4.   $\quad \mu_k \gets \frac{1}{M}\sum_{y=1}^M f_Q^y[k]$
> 5.   $\quad \sigma_k \gets \sqrt{\frac{1}{M}\sum_{y=1}^M (f_Q^y[k] - \mu_k)^2}$
> 6.   $\quad f_Q^y[k] \gets \frac{f_Q^y[k] - \mu_k}{\sigma_k}$ for all $y$
> 7. **end for**
> 8. **$\triangleright$ Add bias dimension (padded as in paper)**
> 9. $\quad f_Q^y[K+1] \gets \frac{1}{\sqrt{M}}$ for all $y$
> ```
>
> We will add this as a part of the Algorithm 1 table in the paper.
>
> # Question
>
> ## Question 1
>
> The performance variation across datasets relates directly to the residual risk $\mathbf{R}^2$ values. From our experiments:
>
> - MMLU has relatively low $\mathbf{R}^2$ values (0.23-0.24 across models);
> - MMLU-Pro exhibits the higher $\mathbf{R}^2$ values (0.24-0.26);
>
> When $\mathbf{R}^2$ is high, the residual term dominates the excessive risk bound, explaining performance degradation. This is consistent with our theoretical prediction.
>
> ## Question 2
>
> Thank you for this question. Please see response to Weakness 1 for experiment and results, as it is the same question.
>
> ## Question 3
>
> We thank you for raising this important question about the conceptual nature of LLM embeddings.Currently, we do not know whether the representation (last token of last layer hidden state) encodes other information. This is out of the focus of our paper, which centers on the effectiveness of CB-ICL for demonstration selection rather than dissecting the intrinsic composition of embeddings.
>
> While we use the term "semantic embedding" in our paper (following common practice in the field), we acknowledge that LLM hidden state representations might capture other information aspects of reasoning and contextual adaptation. We would appreciate it if you could provide specific citations or references to papers that discuss LLM embeddings capturing reasoning dynamics beyond semantics, as we are not aware of any such claims in the literature we have reviewed.
>
> ## Question 4
>
> Thank you for the problem. In our framework, $f(x)$ represents a matrix of embeddings for all possible labels: $f(x) = [f(x,y=1), \dots, f(x,y=M)]\in\mathbb{R}^{K\times M}$. Using $f(x)$ without explicit label conditioning would lose the critical structure needed for our theoretical analysis of classification performance.
>
> ## Question 5
>
> We thank the reviewer for raising this interesting phenomenon. We believe this phenomenon comes from the smaller dataset size. To calculate the $\mathbf{R}^2$, the true $\alpha$ is required. We estimate $\alpha$ by using the full dataset. MMLU (large  dataset, 14042 items) is sufficient to estimate the true $\alpha$. However, GPQA and GPQA-diamond dataset are small dataset (448 and 108 respectively). The small dataset size add to the variance of estimation of true $\alpha$, which becomes the variance of estimation of $\mathbf{R}^2$.
>
> ## Question 6
> Our theoretical analysis provides insights into how distributional mismatches between demonstrations and queries affect CB-ICL performance:
>
> 1. When demonstrations and query are designed to solve the same task (as illustrated in Figure 2), our theoretical bounds remain applicable even if their semantic similarity is low. The framework is robust to such variations.
> 2. When demonstrations and query address fundamentally different tasks, this can be interpreted as inaccurate estimation of concept vector from the demonstrations. Intuitively, such misaligned demonstrations yield poor performance, and our theoretical framework provides quantitative error bounds that characterize this degradation. Specifically, Theorem 4.3 and Theorem 4.4 explicitly capture the performance penalty, which increase when demonstrations fail to adequately represent the query's semantic requirements.

---

### Official Review · Reviewer_uEdm · 2025-11-01

**Soundness:** 3
**Presentation:** 3
**Contribution:** 3
**Rating:** 6
**Confidence:** 3

**Summary:**

The paper studies Concept-Based In-Context Learning (CB-ICL), a view of ICL where few-shot demonstrations are used to estimate a concept representation from a frozen LLM and then predict query labels by aligning the query embedding with that concept. The authors formalize a regression-style objective and analyze three regimes: (i) complete & sufficient (embedding contains the target concept and demos span it), (ii) complete but insufficient (embedding complete, demos rank-deficient), and (iii) incomplete & insufficient (embedding misses part of the concept and demos don’t cover it). They derive spectral risk bounds tied to the demo/embedding matrices, and present a risk→classification link via a margin argument. From the bounds, they introduce a demo–query similarity score to guide demonstration selection. Experiments across several LLM backbones and benchmarks report improvements when choosing demonstrations by the proposed score compared with random selection, and ablations illustrate how demo quality and alignment influence CB-ICL performance.

**Strengths:**

- The CB-ICL setup captures the “prototype” intuition behind many prompt designs, making an implicit practice explicit and analyzable.
- By distinguishing completeness from sufficiency, the analysis teases apart backbone capacity vs. prompt curation which is useful for diagnosing where performance originates.
- The bound induces a demo–query similarity measure: selecting demos that are more similar (in the sense derived from the matrices in the proofs) tightens the bound, offering a criterion for demo selection that aligns with the theory.

**Weaknesses:**

- The method treats a decoder-based LLM as an encoder to produce task-relevant representations for concept estimation. However, decoder-only models are not generally optimized for embedding quality, especially without contrastive representation training. The proposed CB-ICL framework assumes these embeddings are sufficient for reliable concept extraction and similarity-based prediction, this raises doubts about suboptimal representations from a decoder-only backbone.
- The paper’s conclusions are established within the concept-based ICL pipeline (estimating concept vectors from few-shot demos and predicting via similarity). It is unclear whether these results extend to general ICL behaviors. The manuscript does not delineate the boundary between what is specific to the linear view and what persists for broader ICL mechanisms, leaving the external validity of the conclusions uncertain.
- Theorem 4.5 says the label error probability is lower bounded under small excessive risk, yet the paragraph then claims this “leads to small error probability.” It seems an upper bound can certify small error not a lower bound.
- The text defines complete LLMs (residual = 0) and sufficient demos (invertible). However, no empirical diagnostics or proxies are provided to distinguish failure modes in practice. Readers cannot tell whether poor performance arises from missing concept coverage in the chosen demonstrations or from backbone incompleteness of the embedding itself.
- Section 5.2 compares random vs their similarity (“golden”) selection only, so the new score’s advantage over recent strong baselines (also considering coverage and diversity in selection) remains untested.

**Questions:**

- Could the authors give a formal derivation of the claimed connection between the proposed extractor and an attention-style computation with a quadratic (i.e., linear) function? (line 330)
- The extractor yields coefficients that weight the demonstrations when predicting the query. Are there constraints (non-negativity, sum-to-one) applied? If not, how should we interpret negative or >1 coefficients in the context of “attention-like” weighting?
- Could the author include ablation that prompt concept extractor where the extractor is replaced by a  attention-style computation.
- LLMs do
- line 331, attetion -> attention

---

> ### Author Response · Authors · 2025-11-23
> **Response Part 1 / 3**
>
> # Weakness
>
> ## Weakness 1
> We appreciate your valuable concern regarding the embedding quality of decoder-only models. While we agree that decoder-only models are not trained for contrastive representation learning, research field has explored using decoder-only as embedding models, showing that using pre-trained LLMs as embedding extractors is possible [1-4].
>
> To address whether decoder-only embeddings are inferior to those of other architectures, we conducted experiments on encoder-only model, encoder-decoder models (e.g., FLAN-T5, BART), and decoder-only models, covering all architectural types mentioned. The experiments followed the same CB-ICL workflow as the main paper. The model we used is listed as following:
> | Model Structure | Selected Model | Hidden state |
> | --- | --- | --- |
> | Encoder-decoder | T5, BART | Output of last layer of encoder with mean pooling |
> | Encoder-only | BERT, MPNet, MiniLM | Last layer output with mean pooling |
>
> The results are shown below:
> | Type | Encoder-Decoder |  |  |  |  |  | Encoder-Only |  |  |  |  |  |
> | --- | --- | --- | --- | --- | --- | --- | --- | --- | --- | --- | --- | --- |
> | Model | FLAN-T5-XXL |  | FLAN-T5-XL |  | BART |  | MiniLM |  | BERT |  | MPNET |  |
> |  | ICL (%) | CB-ICL (%) | ICL (%) | CB-ICL (%) | ICL (%) | CB-ICL (%) | ICL (%) | CB-ICL (%) | ICL (%) | CB-ICL (%) | ICL (%) | CB-ICL (%) |
> | MMLU | 55.1 | 54.67 | 52.4 | 50.53 | 22.9 | 26.12 | Not applied | 25.75 | 23.02 | 25.65 | Not applied | 25.85 |
> | MMLU-Pro | 16.54 | 15.43 | 15.37 | 15.90 | 11.24 | 10.36 | Not applied | 10.10 | 10.59 | 9.43 | Not applied | 9.72 |
> | GPQA | 26.85 | 28.00 | 23.52 | 27.80 | 27.27 | 26.93 | Not applied | 29.08 | 25.44 | 27.31 | Not applied | 28.63 |
> | GPQA-Diamond | 22.46 | 30.10 | 24.08 | 29.03 | 21.50 | 33.24 | Not applied | 25.98 | 29.19 | 29.03 | Not applied | 29.18 |
>
> For detail information, we repeat all experiment 10 times and report the average value. Results confirm that CB-ICL is effective across these paradigms: on most datasets (e.g., GPQA-Diamond, GPQA), CB-ICL consistently outperforms vanilla ICL. For instance, FLAN-T5-XXL achieves 30.10% in the setting of CB-ICL, compared with 22.46% under ICL on GPQA-Diamond, and BART reaches 33.24% (CB-ICL) compared with 21.50% (ICL) on the same dataset. Even for encoder-only models where vanilla ICL is not applicable, CB-ICL still yields meaningful performance (e.g., 29.18% on GPQA-Diamond for MPNET), demonstrating CB-ICL’s broad applicability beyond decoder-only LLMs. What’s more, by comparison, decoder-only models had higher ICL baselines (e.g., Qwen3-32B reached 76.46% on MMLU) and required no additional adjustments to work with CB-ICL.
>
> [1] Chung et al., Scaling Instruction-Finetuned Language Models, 2022
>
> [2] Li, Ziyue, and Tianyi Zhou. "Your Mixture-of-Experts LLM Is Secretly an Embedding Model for Free." The Thirteenth International Conference on Learning Representations (2025).
>
> [3] Jiang, Albert Q., et al. "Repurposing language models into embedding models: Finding the compute-optimal recipe." Advances in Neural Information Processing Systems 37 (2024): 61106-61137.
>
> [4] Nikolaou, Giorgos, et al. "Language Models are Injective and Hence Invertible." arXiv preprint arXiv:2510.15511 (2025).
>
> ## Weakness 2
> Thank you for the comment. We agree that our study focuses on the concept-based ICL pipeline (estimating concept vectors from few-shot demonstrations and using similarity-based prediction) and therefore does not claim that **all** phenomena of general ICL will automatically follow. Our intention was to use a linear-view approximation of ICL in order to provide insight into the mechanisms underlying ICL behavior. At the same time, we emphasize that:
>
> 1. We explicitly acknowledge this scope limitation in the manuscript (and will further clarify it in revision).
> 2. While we do not claim that **all** ICL behaviors (such as chain-of-thought reasoning, compositional generation, or highly structured output tasks) are captured by our linear framework, we believe our work makes a meaningful contribution by isolating and analyzing a simpler sub-mechanism of ICL which has not been systematically studied in prior work.
>
> In future work we will investigate the extent to which our conclusions carry over to broader ICL settings (for example, chain-of-thoughts). We will add a discussion section in the revision to make this explicit.
>
> ## Weakness 3
>
> Thank you for raising this point. We agree that the wording around Theorem 4.5 may have caused some confusion regarding the direction of the bound. What Theorem 4.5 presents is indeed a lower bound on the error probability under the condition of small excessive risk. In effect, a lower bound on the error probability yields an upper bound on the correct-label probability controlled by the excessive risk. Therefore the direction is consistent. We will update the manuscript to explicitly phrase.

---

> ### Author Response · Authors · 2025-11-23
> **Response Part 2 / 3**
>
> # Weakness
>
> ## Weakness 4
>
> We appreciate the valuable comment pointing out the need for empirical diagnostics to distinguish failure modes from insufficient demonstrations and incomplete LLM embeddings. Building on our theoretical framework, we provide concrete, computable proxies and empirical cases to address this gap, directly aligned with the definitions in the paper:
>
> 1. Empirical Proxy for Demonstration Sufficiency:
>
> As theoretically defined, **sufficiency** requires $\mathbf{F}_n(x^n)$ to be invertible (full-rank, rank=$K$, where $K$ is LLM embedding dimension). However, since only few demonstrations are provided, this matrix is usually not invertible. Hence, we can use the rank to indicates if demonstrations fail to span the semantic subspace of the LLM embeddings, leaving parts of the prompt concept $\underline{\alpha}$ unlearnable (please see the Theorem 4.3’s penalty term).
>
> 2. Empirical Proxy for LLM Completeness (Residual $\mathbf{R}^2$ Quantification)
>
> **Completeness** requires $\mathbf{R}(x,y) = P_{Y|X}(y,x) - \underline{\alpha}^T f(x,y) = 0$ (LLM embeddings capture all task-relevant knowledge). For practical diagnosis, we quantify the residual as completeness
>
> 1. Estimate the ground-truth concept $\underline{\alpha}$ using the prompt concept extractor $\hat{\underline{\alpha}} = \mathbf{F}_n^\dagger(x^n)\bar{f}_n(x^n,y^n)$ using the full dataset.
> 2. For each query $(x_Q,y_Q)$, compute the predicted label distribution $\hat{P}(y|x_Q) = \hat{\underline{\alpha}}^T f(x_Q,y)$.
> 3. Obtain the one-hot vector for classification tasks $P(y|x_Q)$ (e.g., ).
> 4. Compute the residual sum of squares as a proxy for $\mathbf{R}^2 = \sum_{x_Q,y} P(x_Q) [P(y|x_Q) - \hat{P}(y|x_Q)]^2$, where $P(x_Q)$ is the query distribution. Smaller $R^2$ indicates higher completeness.
>
> ## Weakness 5
>
> Thank you for the question on baselines. We compared the performance of CB-ICL (golden) with the cosine similarity baseline across models of different parameter scales and datasets. Results show CB-ICL (golden) performs slightly better than cosine similarity overall: the score gap between the two methods is within ±2% across most datasets and model scales (8B/14B/32B), with CB-ICL outperforming cosine similarity by 0.4%–1.5% in average scores. While cosine similarity marginally outperforms CB-ICL on individual datasets (e.g., MMLU for Qwen3-14B), this advantage is not universal.
>
> | **Dataset** | **LLaMA3-8B** |  | **Qwen3-8B** |  | **Deepseek-R1-8B** |  |
> | --- | --- | --- | --- | --- | --- | --- |
> |  | CB-ICL (%) | Cosine Similarity (%) | CB-ICL (%) | Cosine Similarity (%) | CB-ICL (%) | Cosine Similarity (%) |
> | MMLU | 73.74 | 72.45 | 80.86 | 79.41 | 68.44 | 68.24 |
> | MMLU-Pro | 36.59 | 36.54 | 58.97 | 59.74 | 44.24 | 42.84 |
> | GPQA | 48.51 | 49.10 | 65.77 | 66.67 | 51.67 | 52.56 |
> | GPQA-diamond | 42.28 | 41.35 | 73.83 | 70.56 | 52.47 | 51.78 |
> | Average | 50.28 | 49.86 | 69.86 | 69.10 | 54.21 | 53.86 |
>
> | **Dataset** | **Qwen3-14B** |  | **Deepseek-R1-14B** |  | **Qwen3-32B** |  |
> | --- | --- | --- | --- | --- | --- | --- |
> |  | CB-ICL (%) | Cosine Similarity (%) | CB-ICL (%) | Cosine Similarity (%) | CB-ICL (%) | Cosine Similarity (%) |
> | MMLU | 82.64 | 82.98 | 82.34 | 83.15 | 84.42 | 84.20 |
> | MMLU-Pro | 62.67 | 60.34 | 61.34 | 61.52 | 67.29 | 66.24 |
> | GPQA | 55.83 | 53.16 | 52.32 | 50.50 | 58.73 | 57.35 |
> | GPQA-diamond | 67.32 | 66.39 | 62.91 | 61.32 | 67.43 | 66.67 |
> | Average | 67.12 | 65.72 | 64.73 | 64.12 | 69.47 | 68.62 |
>
> # Question
>
> ## Question 1
>
> Thank you for this problem. In our setting, the predicted distribution is $\hat{P}_{Y|X}(y|x_Q)=\underline{\hat{\alpha}}^\text{T}(x^n,y^n)f(x_Q,y)$ with
>
> $$
> \underline{\hat{\alpha}}^\text{T}(x^n,y^n) = \left( \frac{1}{n} \sum_{i=1}^{n} \sum_{y} f(x_i,y) f^{\text{T}}(x_i,y) \right)^{\dagger}\left(\frac1n \sum_{i=1}^n f(x_i,y_i)\right).
> $$
>
> Denote $f(x^n) = [f(x_1,1),\dots,f(x_1,M),\dots,f(x_n,M)]$. Note that
>
> $$
> \left( \frac1n \sum_{i=1}^n\sum_y f(x_i,y)f^\text{T}(x_i,y)\right)^\dagger = \left(\frac1n f(x^n)f^\text{T}(x^n) \right)^\dagger = nf(x^n) \left(f^\text{T}(x^n)f(x^n)\right)^\dagger\left(f^\text{T}(x^n)f(x^n)\right)^\dagger f^\text{T}(x^n).
> $$
>
> Linear attention [1] follows
>
> $$
> \text{Attention}(q,\{k_i,v_i\}) = \phi^\text{T}(q) \left(\sum_i \phi(k_i)v_i^\text{T}\right).
> $$
>
> To write in the same form, we can set
>
> $\phi(q) =  \left(f^\text{T}(x^n)f(x^n)\right)^\dagger f^\text{T}(x^n) f(x_Q)$ with $q= f(x_Q) = [f(x_Q,y=1)\dots, f(x_Q,y=M)]$
>
> $\phi(k_i) = \left(f^\text{T}(x^n)f(x^n)\right)^\dagger f^\text{T}(x^n) f(x_i)$ with $k_i = f(x_i) = [f(x_i,y=1),\dots,f(x_i,y=M)]$
>
> and $v_i=e_i$ being one-hot vector with the i-th element equals 1.
>
> [1] Katharopoulos, Angelos, et al. "Transformers are rnns: Fast autoregressive transformers with linear attention." International conference on machine learning. PMLR, 2020.

---

> ### Author Response · Authors · 2025-11-23
> **Response Part 3 / 3**
>
> # Question
>
> ## Question 2
>
> Thank you for pointing this out. We clarify the status of imposed constraints on the extractor coefficients, and acknowledge an important limitation of the current framework:
>
> 1. **Sum-to-one constraint**: We *do* enforce that the $\hat{P}_{Y|X}(y|x_Q)$ sum to one according to $y$. This is conducted by the normalization.
> 2. **Non-negativity constraint**: We do *not* enforce non-negativity of the coefficients. As a result, some weights may become negative, or in principle exceed 1.
>
> Check of the sum-to-one constraint:
>
> $$
> \sum_y \hat{P}_{Y|X}(y|x) = \sum_y \underline{\hat{\alpha}}^\text{T}(x^n,y^n) f(x,y) =  \underline{\hat{\alpha}}^\text{T}(x^n,y^n) \left(\sum_y f(x,y)\right)
> $$
>
> with
>
> $$
> \sum_y f(x,y) = [0, \dots, 0 , \sqrt{M}]^\text{T}.
> $$
>
> Furthermore, denote ${\mathbf{F}}(x) = \sum_y f(x,y) f^\text{T}(x,y)$ and $\mathbf{F}(x^n) = \frac1n \sum_{i=1}^n \mathbf{F}(x)$. Let $\left({\mathbf{F}}(x)\right)_{ij}$ denote the element at $i$-th row and $j$-th column. We have that
>
> $$
> \left({\mathbf{F}}(x)\right)_{Kj} = \sum_y f_K(x,y)f_j(x,y) = \frac{1}{\sqrt{M}}\sum_y f_j(x,y)= 1(j=K),
> $$
>
> with $1(j=K)=1$ if $j=K$ and $1(j=K)=0$ if $j\neq K$,
> and
>
> $$
> \left({\mathbf{F}}(x)\right)_{jK} = \sum_y f_j(x,y)f_K(x,y) = \frac{1}{\sqrt{M}}\sum_yf_j(x,y)= 1(j=K).
> $$
>
> Hence, the sum of $\hat{P}_{Y|X}(y|x_Q)$ is $1$ for all $x_Q$.
>
> In our formulation, since only the sum-to-one constraint holds, the coefficients should be viewed as signed weights (possibly extrapolative) rather than strictly “share of attention” in the conventional sense. We recognize that this is a limitation of our current linear framework, as we do not yet guarantee that the weights reside in the [0, 1] interval, which would make the analogy to soft-attention tighter. We will add a sentence in the revised manuscript to explicitly state this limitation and clarify how the coefficients should be interpreted.
>
> ## Question 3
>
> Thank you for the ablation problem. We replace the generation of $\hat{P}_{Y|X}$ with the standard attention with
>
> $$
> \hat{P}_{Y|X}(y|x_Q) = \text{softmax}\left(\frac{f^\text{T}(x_Q) f(x^n)}{\sqrt{K}}\right)V
> $$
>
> where $V = [e_{y_1}; e_{y_2}; \dots, e_{y_n}]\in\mathbb{R}^{nM}$ and $e_{y_i}\in\mathbb{R}^m$ is one-hot vector with the $y_i$th element being 1. Based on this, we test the performance of LLaMA, Qwen and Deepseek families.
>
> | **Dataset** | **LLaMA3-8B** |  | **Qwen3-8B** |  | **Deepseek-R1-8B** |  |
> | --- | --- | --- | --- | --- | --- | --- |
> |  | CB-ICL (%) | Attention (%) | CB-ICL (%) | Attention (%) | CB-ICL (%) | Attention (%) |
> | MMLU | 71.07 | 50.24 | 77.77 | 59.02 | 64.58 | 53.19 |
> | MMLU-Pro | 33.26 | 12.60 | 53.62 | 43.78 | 42.3 | 31.40 |
> | GPQA | 40.77 | 21.41 | 61.60 | 40.52 | 50.82 | 39.56 |
> | GPQA-diamond | 33.56 | 10.18 | 63.01 | 55.21 | 48.82 | 38.47 |
> | Average | 44.67 | 23.61 | 64.00 | 49.64 | 51.63 | 40.66 |
>
> | **Dataset** | **Qwen3-14B** |  | **Deepseek-R1-14B** |  | **Qwen3-32B** |  |
> | --- | --- | --- | --- | --- | --- | --- |
> |  | CB-ICL (%) | Attention (%) | CB-ICL (%) | Attention (%) | CB-ICL (%) | Attention (%) |
> | MMLU | 81.38 | 67.43 | 80.32 | 64.53 | 83.62 | 65.12 |
> | MMLU-Pro | 60.47 | 52.08 | 59.04 | 53.08 | 65.89 | 51.41 |
> | GPQA | 50.65 | 46.43 | 48.45 | 47.15 | 54.82 | 46.91 |
> | GPQA-diamond | 64.88 | 50.43 | 60.13 | 57.96 | 66.34 | 60.32 |
> | Average | 64.35 | 54.09 | 61.99 | 55.68 | 67.67 | 55.94 |
>
> Results show that the standard attention mechanism cannot fully exploit the conceptual structure recovered by CB-ICL.
>
> ## Question 4
>
> The sentence provided in Question 4 seems to be incomplete.
>
> ## Question 5
>
> We thank the you for the comment. We will carefully proofread and correct the manuscript in the final version

---

### Author Response · Authors · 2025-11-23
**Global Response**

We thank all reviewers for their thoughtful and constructive feedback.

# Summary of our paper
Our work develops a theoretical framework for Concept-Based In-Context Learning (CB-ICL) and provides a systematic understanding of why ICL can succeed with only a few demonstrations.
Our main contributions are:
1. Theoretical justification of CB-ICL: we show how a pre-trained LLM induces a semantic subspace that enables concept extraction from few-shot prompts.
2. Characterization of key factors affecting ICL performance, including embedding dimension, number of demonstrations, and label cardinality.
3. A principled similarity measure derived from the excess-risk bound, providing guidance for prompt selection.
4. A principled residual term quantifying the completeness of LLM embeddings and measuring how much task-relevant knowledge is missing from pre-training.

# Summary of reviewer comments
Reviewers raised concerns regarding:
1. The generality and scope of the CB-ICL framework, and whether its conclusions extend beyond the decoder-only architecture transformers.
2. Additional comparisons and ablations, such as attention-based alternatives, sentence encoders, and more diverse demo-selection baselines.
3. Stronger empirical validation, including repeated runs/variance, stronger baselines, diagnostics for completeness and sufficiency, and empirical tests of theoretical predictions (e.g., dimension K).
4. Multiple technical clarifications on linear-attention derivation, coefficient interpretation, distribution shifts, model assumptions, and tightness of bounds.
5. Clearer exposition and notation, including better introduction of symbols, clearer pipeline description, and improved readability.

# Summary of response
To address the reviewers’ concerns, we made substantial empirical and presentation improvements:
1. Cross-architecture experiments (encoder-only, encoder-decoder, decoder-only), showing CB-ICL consistently improves performance across all model families.
2. Attention-style ablation, replacing our extractor with standard attention, which leads to large performance drops.
3. Repeated-run evaluation, reporting variance on GPQA/GPQA-Diamond for more reliable assessment.
4. New baseline comparison, evaluating our similarity score against cosine similarity.
5. Dimension trade-off experiments, empirically validating the discussion in the paper.
6. Wall-clock time measurements for CB-ICL steps.
7. Improved readability, adding a notation table and an algorithm table.
8. Several clarifications on assumptions, especially on normalization.

These additions significantly strengthen both empirical support and clarity of our paper.

---

### Author Response · Authors · 2025-12-03
**Summary of Reviews and the Reviewer–Author Discussion**

Dear PCs, SACs, ACs, and Reviewers,

Thank you for your thoughtful evaluation of our submission. To support the newly assigned AC and help reduce their workload, we provide below a summary of the key points from the reviews and the reviewer-author discussions.

**Strength:** Overall, we appreciate that the reviewers highlighted the following strengths of our work.

- (Reviewer uEdm, Strength 1; Reviewer Qc59, Strength 3) The CB-ICL setup makes explicit and analyzable the prototype intuition behind many prompt designs, and distinguishes clearly between backbone capacity and prompt curation, which several reviewers found useful for diagnosing where ICL performance comes from.
- (Reviewer Qc59, Strength 1; Reviewer aBfV, Strength 1, 3, 5) The paper provides a coherent and rigorous theoretical framework: assumptions are explicitly stated, derivations are largely complete, and the resulting quantities (excessive risk, similarity score, eigenvalue-based criteria) are interpretable and connect embedding geometry to prediction error.
- (Reviewer uEdm, Strength 2; Reviewer aBfV, Strength 2) Reviewers appreciated that our analysis covers multiple practically relevant regimes, including complete vs. incomplete backbones, sufficient vs. insufficient demonstrations, and that these cases are tied to concrete notions such as residual risk and eigenvalue-based similarity.
- (Reviewer Qc59, Strength 3) The work offers targeted novelty by integrating familiar linear/ICL analytical tools into a concept-based ICL formulation and deriving a practically useful similarity metric and demo-selection criterion.
- (Reviewer uEdm, Strength 3; Reviewer aBfV, Strength 4) The theoretical insights were viewed as having practical relevance, as reviewers noted that our bounds and eigenvalue conditions provide guidance for understanding when ICL succeeds or fails, and that the proposed golden selection empirically improves performance in several settings.
- (Reviewer Qc59, Strength 2) Experiments across multiple benchmarks and model families confirm the predictive power and practical utility of the theoretical insights.

**Main Concerns and Our Addressing:** During the discussion period, we actively addressed the reviewers' concerns. We provided the additional proofs, experiments, clarifying paragraphs that the reviewers requested and rewritten the related parts.

- Embedding Quality (Reviewer uEdm, Weaness 1; Reviewer Qc59, Weaness 3, Question 2; Reviewer aBfV, Question 5): We extended experiments comparing decoder-only embeddings to off-the-shelf sentence encoders (including encoder-decoder and encoder-only models) and clarified their interpretation in our framework.
- Scope of Conclusions (Reviewer uEdm, Weakness 2; Reviewer Qc59, Weakness 5, Question 3): We state our limitations in conclusion section to clarify the boundary of our results, addressing when they apply to broader ICL behaviors and extending the discussion to transformer architectures.
- Error Probability (Reviewer uEdm, Weakness 3): We provide the explanation of Theorem 4.5 to clarify the error-probability bounds, ensuring the distinction between lower and upper bounds is clear.
- Completeness vs. Sufficiency (Reviewer uEdm, Weakness 4; Reviewer Qc59, Question 1): We highlighted empirical diagnostics to distinguish poor performance due to missing concept coverage or incomplete embeddings and discussed these in practice.
- Baselines (Reviewer uEdm, Weakness 5; Reviewer Qc59, Weakness 4/6; Reviewer aBfV, Weakness 1/4, Question 1/3/4): We added several extra experiments, including cosine similarity baselines and ablation on dimension $K$ in the revision.
- Runtime Usage (Reviewer uEdm, Weakness 5; Reviewer Qc59, Weakness 7; Reviewer aBfV, Question 7): We add a detailed runtime analysis in the revision.
- Attention ablation (Reviewer uEdm, Question 1, Question 3): We add an ablation where we replaced our extractor with an attention-style computation, and add a detailed discussion to this in the revision.
- Notation and Exposition (Reviewer uEdm, Question 4/5; Reviewer Qc59, Weakness 2/3, Question 2): We revised the notation for clarity, introduced symbols at first use, and expanded the experimental section with explicit details on extraction, normalization, and runtime costs.
- Questions on Theories (Reviewer uEdm, Question 2; Reviewer Qc59, Weakness 8; Reviewer aBfV, Question 6): We add several discussions in the revision, including the discussion on the properties of propose CB-ICL, implementation of normalization and tightness of bounds.

---

### Meta-Review · Area_Chair_Msyf · 2026-01-04

**Summary:**

This work explores the theoretical understanding of the ICL mechanism by introducing concept-based ICL (CB-ICL) approach. Specifically, the authors propose theoretical analyses on applying CB-ICL to ICL tasks, which explains why and when the CB-ICL performs well for predicting query labels in prompts with only a few demonstrations.  In addition, they use the introduced theory to quantify the knowledge that can be leveraged by the LLMs to the prompt tasks, which can potentially offer guidance for ICL.

Although this work offers some insights into ICL, as many reviewers have noted, it relies on relatively strong assumptions and lacks sufficient empirical validation that directly supports the theoretical claims, and therefore currently falls below the acceptance threshold.

**Reviewer Scores:**

No reviewers updated their scores.

---

### Decision · Program_Chairs · 2026-01-26

Reject